# Variable prediction accuracy of polygenic scores within an ancestry group

**Hakhamanesh Mostafavi[1†‡]\*, Arbel Harpak[1†]\*, Ipsita Agarwal[1], Dalton Conley[2,3], Jonathan K Pritchard[4,5,6], Molly Przeworski[1,7]\***

[1]Department of Biological Sciences, Columbia University, New York, United States; [2]Department of Sociology, Princeton University, Princeton, United States; [3]Office of Population Research, Princeton University, Princeton, United States; [4]Department of Genetics, Stanford University, Stanford, United States; [5]Department of Biology, Stanford University, Stanford, United States; [6]Howard Hughes Medical Institute, Stanford University, Stanford, United States; [7]Department of Systems Biology, Columbia University, New York, United States

**\*For correspondence:**
hsm2137@columbia.edu (HM);
ah3586@columbia.edu (AH);
mp3284@columbia.edu (MP)

[†]These authors contributed equally to this work

**Present address:** [‡]Department of Genetics, Stanford University, Stanford, United States

**Abstract** Fields as diverse as human genetics and sociology are increasingly using polygenic scores based on genome-wide association studies (GWAS) for phenotypic prediction. However, recent work has shown that polygenic scores have limited portability across groups of different genetic ancestries, restricting the contexts in which they can be used reliably and potentially creating serious inequities in future clinical applications. Using the UK Biobank data, we demonstrate that even within a single ancestry group (i.e., when there are negligible differences in linkage disequilibrium or in causal alleles frequencies), the prediction accuracy of polygenic scores can depend on characteristics such as the socio-economic status, age or sex of the individuals in which the GWAS and the prediction were conducted, as well as on the GWAS design. Our findings highlight both the complexities of interpreting polygenic scores and underappreciated obstacles to their broad use.

## Introduction

Genome-wide association studies (GWAS) have now been conducted for thousands of human complex traits, revealing that the genetic architecture is almost always highly polygenic, that is that the bulk of the heritable variation is due to thousands of genetic variants, each with tiny marginal effects (*Boyle et al., 2017*; *Bulik-Sullivan et al., 2015*). These findings make it difficult to interpret the molecular basis for variation in a trait, but they lend themselves more immediately to another use: phenotypic prediction. Under the assumption that alleles act additively, a 'polygenic score' (PGS) can be created by summing the effects of the alleles carried by an individual; this score can then be used to predict that individual's phenotype (*Henderson, 1984*; *Meuwissen et al., 2001*; *Kathiresan et al., 2008*; *Lynch and Walsh, 1998*). For highly heritable traits, such scores already provide informative predictions in some contexts: for example, prediction accuracies are 24.4% for height (using $R^2$ as a measure) (*Yengo et al., 2018*) and up to 13% for educational attainment (using incremental $R^2$) (*Lee et al., 2018*).

This genomic approach to phenotypic prediction has been rapidly adopted in three distinct fields. In human genetics, PGS have been shown to help identify individuals that are more likely to be at risk of diseases such as breast cancer and cardiovascular disease (*Khera et al., 2018*; *Inouye et al., 2018*; *Mavaddat et al., 2019*; *Khera et al., 2019*). Based on these findings, a number of papers have advocated that PGS be adopted in designing clinical studies, and by clinicians as additional risk factors to consider in treating patients (*Torkamani et al., 2018*; *Khera et al., 2018*). In human evolutionary genetics, several lines of evidence suggest that adaptation may often take the form of shifts

**eLife digest** Complex diseases like cancer and heart disease are caused by the interplay of many factors: the variants of genes we inherit, the lifestyles we lead and the environments we inhabit, plus the interaction of all these factors. In fact, almost every trait, even how many years we will spend studying, is influenced both by our environment and our genes.

To identify some of the genetic factors at play, scientists perform analyses known as genome-wide association studies, or GWAS for short. In these studies, the genomes from many different people are scanned to look for genetic differences associated with differences in traits. By summing up all the small genetic differences, so-called "polygenic scores" can be calculated. When there is a large genetic component to a trait, polygenic scores can be useful predictive tools.

But there is a catch: polygenic scores make less accurate predictions for individuals of a different ancestry than those involved in the GWAS, which limits the use of these tools around the world. Mostafavi, Harpak et al. set out to understand if there are other factors in addition to ancestry that could influence the performance of polygenic scores.

Using data from the UK Biobank, an international health resource that pairs genomic data and clinical information, Mostafavi, Harpak et al. examined polygenic scores among individuals that share a single, common ancestry. These polygenic scores were used to predict three traits (blood pressure, body mass index and educational attainment) in individuals and the predictions were then compared to the actual trait values to see how accurate they were. The analysis revealed that even within a group of people with similar ancestry, the accuracy of polygenic scores can vary, depending on characteristics such as the sex, age or socioeconomic status of the individuals.

This analysis emphasises how variable GWAS and their predictive value can be even within seemingly similar population groups. It further highlights both the complexities of interpreting polygenic scores and underappreciated obstacles to their broad use in medical and social sciences.

in the optimum of a polygenic phenotype and hence act jointly on the many variants that influence the phenotype (*Pritchard and Di Rienzo, 2010*; *Berg and Coop, 2014*; *Höllinger et al., 2019*; *Sella and Barton, 2019*). In this context, the goal is to test whether the set of variants that influence a trait are rapidly evolving across populations or over time (*Field et al., 2016*; *Berg et al., 2019*; *Uricchio et al., 2019*; *Edge and Coop, 2019*; *Racimo et al., 2018*; *Berg and Coop, 2014*). Finally, in various disciplines of the social sciences, PGS are increasingly used to distinguish environmental from genetic sources of variability (*Conley, 2016*), as well as to understand how genetic variation among individuals may cause heterogeneous treatment effects when studying how an environmental influence (e.g., a schooling reform) affects an outcome (such as BMI) (*Barcellos et al., 2018*; *Davies et al., 2018*). In all these applications, the premise is that PGS will 'port' well across groups—that is that they remain predictive not only in samples very similar to the ones in which the GWAS was conducted, but also in other sets of individuals (henceforth 'prediction sets').

As recent papers have highlighted, however, PGS are not as predictive in individuals whose genetic ancestry differs substantially from the ancestry of individuals in the original GWAS (reviewed in *Martin et al., 2019*). As one illustration, PGS calculated in the UK Biobank predict phenotypes of individuals sampled in the UK Biobank better than those of individuals sampled in the BioBank Japan Project: for instance, the incremental $R^2$ for height is approximately 11% in the UK versus 3% in Japan (*Martin et al., 2019*). Similarly, using PGS based on Europeans and European-Americans, the largest educational attainment GWAS to date ('EA3') reported an incremental $R^2$ of 10.6% for European-Americans but only 1.6% for African-Americans (*Lee et al., 2018*).

To date, such observations have been discussed mainly in terms of population genetic factors that reduce portability (*Martin et al., 2017*; *Kim et al., 2018*; *Duncan et al., 2018*; *De La Vega and Bustamante, 2018*; *Sirugo et al., 2019*; *Martin et al., 2019*). Notably, GWAS does not pinpoint causal variants, but instead implicates a set of possible causal variants that lie in close physical proximity in the genome. The estimated effect of a given SNP depends on the extent of linkage disequilibrium (LD) with the causal sites (*Pritchard and Przeworski, 2001*; *Bulik-Sullivan et al., 2015*). LD differences between populations that arose from their distinct demographic and recombination histories will lead to variation in the estimated effect sizes and hence to variable

phenotypic prediction accuracies (*Rosenberg et al., 2019*). Populations will also differ in the allele frequencies of causal variants. This problem is particularly acute for alleles that are rare in the population in which the GWAS was conducted but common in the population in which the trait is being predicted. Such variants are likely to have noisy effect size estimates in the estimation sample or may not be included in the PGS at all, and yet they contribute substantially to heritability in the target population. Furthermore, causal loci or effect sizes may differ among populations, for instance if the effect of an allele depends on the genetic background on which it arises (e.g., *Adhikari et al., 2019*). For all these reasons, we should expect PGS to be less predictive across ancestries.

In practice, given that most individuals (about 80%) included in current GWAS are of European ancestry (*Popejoy and Fullerton, 2016*; *Martin et al., 2019*), PGS are systematically more predictive in European-ancestry individuals than among other people. As a consequence, the clinical applications and scientific understanding to be gained from PGS will predominantly and unfairly benefit a small subset of humanity. A number of papers have therefore highlighted the importance of expanding GWAS efforts to include more diverse ancestries (*Martin et al., 2018*; *Bien et al., 2019*; *Wojcik et al., 2019*; *Martin et al., 2019*; *Sirugo et al., 2019*).

Importantly, factors other than ancestry could also impact the accuracy and portability of PGS. For example, the educational attainment of an individual depends not only on their own genotype, but on the genotypes of their parents, due to nurturing effects (*Kong et al., 2018*), and of their peers, due to social genetic effects (*Domingue et al., 2018*), and of course on non-genetic factors. Also, traits such as height and educational attainment show strong patterns of assortative mating, which can distort effect size estimates in GWAS (*Domingue et al., 2014*; *Robinson et al., 2017*; *Ruby et al., 2018*). To what extent these effects remain the same across cultures and environments is unknown, but if they differ, so will the prediction accuracy. More generally, while we still know little about genotype-environment interactions (GxE) in humans, they are well-documented in other species—notably in experimental settings—and would further reduce the portability of PGS across environments (*Gibson, 2008*; *Tropf et al., 2017*; *Mills and Rahal, 2019*; *Lynch and Walsh, 1998*). In addition, the extent of environmental variability could differ between GWAS and prediction groups, which would change the proportion of the variance in the trait explained by a PGS (i.e., the prediction accuracy). PGS for some traits may also include a component of environmental or cultural confounding with population structure (*Sohail et al., 2019*; *Haworth et al., 2019*; *Lawson et al., 2020*; *Kerminen et al., 2018*; *Berg et al., 2019*); this source of confounding can increase or decrease prediction accuracy, depending on the structure in the prediction samples.

Given these considerations, it is important to ask to what extent PGS are portable among groups within the same ancestry. To explore this question, we stratified the subset of UK Biobank samples designated as 'White British' (WB) according to some of the standard sample characteristics of GWAS studies: the ages of the individuals, their sex, and socio-economic status. We chose to focus on these particular characteristics because they vary among GWAS samples depending on sample ascertainment procedures. Furthermore, these characteristics have been shown to influence heritability for some traits in a study of a subset of the UK Biobank (*Ge et al., 2017*), raising the possibility that these choices also influence prediction accuracy. Indeed, for three example traits, we show that there exist major differences in the prediction accuracy of the PGS among these groups, even though they share highly similar genetic ancestries. We further demonstrate for a variety of traits that prediction accuracy differs markedly depending on whether the GWAS is conducted in unrelated individuals or in pairs of siblings, even when controlling for the precision of the estimates. This finding is again unexpected under standard GWAS assumptions; it underscores the importance of genetic effects that are included in estimates from some study designs and not others and highlights underappreciated challenges with GWAS-based phenotypic prediction.

At present, it is difficult to determine the reasons why we see such variable prediction accuracy across these strata and study designs. Contributing factors probably include indirect genetic effects from relatives, assortative mating, varying levels of genetic and environmental variance, GxE interaction effects and perhaps undetected confounding. Nonetheless, our results make clear that the prediction accuracy of PGS can be affected in unpredictable ways by known—and presumably unknown—factors in addition to genetic ancestry.

## Results

### Sample characteristics of the GWAS and prediction set can influence prediction accuracy even within a single ancestry

We examined how PGS for a few example traits port across samples that are of similar genetic ancestry but differ in terms of some common study characteristics, such as the male:female ratio (henceforth 'sex ratio'), age distribution, or socio-economic status (SES). To this end, we limited our analysis to the largest subset of individuals in the UKB with a relatively homogeneous ancestry: 337,536 unrelated individuals that were characterized by the UKB, based on self-reported ethnicities as well as genetic analysis, as 'White British' (WB) (*Bycroft et al., 2018*). In all analyses, we further adjusted for the first 20 principal components of the genotype data, to account for population structure within this set of individuals (Materials and methods).

In all analyses, we randomly selected a subset of individuals to be the prediction set; we then conducted GWAS using the remaining individuals and built a PGS model by LD-based clumping of the associations (Materials and methods). To examine the reliability of the prediction, we considered the incremental $R^2$, that is the $R^2$ increment obtained when adding the PGS to a model with other covariates (referred to as 'prediction accuracy' henceforth). Whether this measure is appropriate depends on how PGS are to be used; it is not always the most obvious choice in human genetics, where the goal is often to identify individuals at high risk of developing a particular disease (i.e., in the tail of the polygenic score distribution). Nonetheless, because it has been widely reported in discussions of portability across genetic ancestries (e.g., *Lee et al., 2018*; *Martin et al., 2019*), we also used it here; later, we also present some results on binary traits using incremental area under the receiver operator curve (AUC).

As a first case, we considered the prediction accuracy of a PGS for diastolic blood pressure in prediction sets stratified by sex, motivated by reports that variation in this trait may arise for somewhat distinct reasons in the two sexes (*Reckelhoff, 2001*; *Zhou et al., 2017*). We randomly selected males and females as prediction sets (20K individuals each), and used a subset of the rest of the individuals for GWAS, matching the numbers of females and males in the GWAS set (total sample size 122,774); we refer to this mixed set, somewhat loosely, as the 'diverse GWAS.' Adjusting for mean sex effects and medication use (see Materials and methods), the prediction accuracy is about 1.15-fold higher for females than for males (Mann-Whitney $p = 1.1 \cdot 10^{-5}$; *Figure 1A*). Thus, despite equal representation of males and females in the GWAS set, the prediction accuracy varies depending on the sex ratio of prediction samples. To examine this further, we repeated the same analysis but performed the GWAS in only one sex (which we refer to as 'stratified GWAS' using the same sample size as in the diverse GWAS). [Note that the diverse GWAS sample is not a merge of the stratified GWAS samples but a mixed-sex sample of equal sample size to that used in the women-only and the men-only GWAS, to allow for direct comparison between GWASs. Results for the merged GWAS (with a much larger sample size) are presented in *Appendix 1—figure 1A*.] When the GWAS is conducted only in females, the prediction accuracy is about 1.35-fold higher for females than for males; in turn, when GWAS was done in only males, the prediction accuracy in both sexes is similar, as well as somewhat decreased (*Figure 1A*).

We then considered two other cases, evaluating prediction accuracy in groups stratified by age for BMI—since the UK Biobank participants were enrolled within about a five-year span, differences in age could in principle also be reflective of cohort effects—and by adult SES for years of schooling, using the Townsend deprivation index as a measure; our choices were motivated by prior evidence suggesting that these characteristics of the GWAS influence estimates of SNP-heritability (*Branigan et al., 2013*; *Conley et al., 2015*; *Belsky et al., 2018*; *Elks et al., 2012*; *Ge et al., 2017*). We withheld a random set of 10K individuals in each quartile of age and SES for prediction and performed GWAS using a subset of the remaining individuals, matching the sample sizes across quartiles in the GWAS set (total sample sizes of 72,328 and 73,280 for BMI and years of schooling GWAS, respectively). Similar to our observation for diastolic blood pressure, the prediction accuracy varies across prediction sets: it is 1.4-fold higher for BMI in the youngest quartile compared to the oldest (Mann-Whitney $p = 1.1 \cdot 10^{-5}$; *Figure 1B*), and 2-fold higher for years of schooling in the lowest SES quartile compared to the highest (Mann-Whitney $p = 2.9 \cdot 10^{-6}$; *Figure 1C*). Furthermore, the differences across groups are again sensitive to the choice of the GWAS set: the differences are

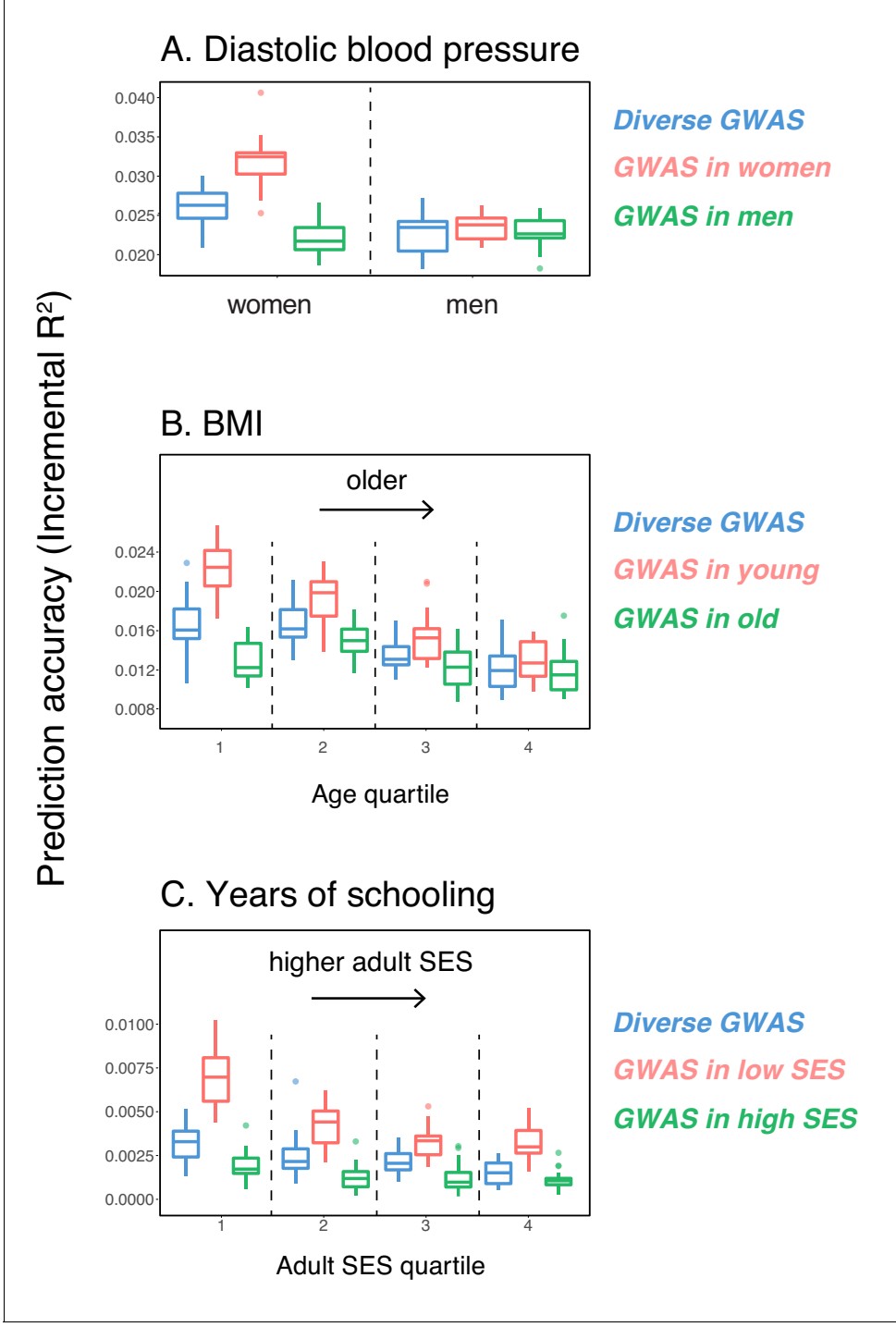

**Figure 1.** Variable prediction accuracy of polygenic scores within an ancestry group. Shown are incremental $R^2$ values (i.e., the increment in $R^2$ obtained by adding a polygenic score predictor to a model with covariates alone) in different prediction sets. Each box and whiskers plot is computed based on 20 iterations of resampling GWAS and prediction sets. Thick horizontal lines denote the medians. The polygenic scores were estimated in samples of unrelated WB individuals. Phenotypes were then predicted in distinct samples of unrelated WB individuals, stratified by sex (**A**), age (**B**) or Townsend deprivation index, a measure of SES (**C**). In red and green cases, polygenic scores are based on a GWAS in a sample limited to one sex, age or SES group (a 'stratum'). In blue, polygenic scores are based on a GWAS in a diverse sample matching the number of individuals in each stratum. GWAS samples sizes are: 122,774 for all three diastolic blood pressure GWAS samples, 72,328 for all three BMI GWAS samples, 73,280 for years of schooling GWAS in the diverse sample and 73,283 for GWAS in the low SES and high SES samples.

marked when GWAS is restricted to the youngest quartile for BMI and the lowest SES quartile for years of schooling, but diminished when the GWAS is performed in the oldest and the highest SES quartiles for BMI and years of schooling, respectively (*Figure 1B, C*). These results remained qualitatively unchanged when we used $R^2$ instead of incremental $R^2$ to measure prediction accuracy (*Appendix 1—figure 2*).

In these analyses, we used a p-value threshold of $10^{-4}$ for inclusion of a SNP in the PGS. The choice of how stringent to make the GWAS p-value threshold is important but somewhat arbitrary, with approaches ranging from requiring genome-wide significance to including all SNPs (*Weedon et al., 2008*; *Pharoah et al., 2008*; *Euesden et al., 2015*; *Vilhjálmsson et al., 2015*; *Ware et al., 2017*; *Mostafavi et al., 2017*; *Speidel et al., 2019*). Often, this threshold is chosen to maximize prediction accuracy in an independent validation set. When the goal is to compare prediction performance across different groups, there is no obvious optimal choice of the p-value threshold. [The optimal p-value in this context will differ across studies, as it depends not only on the genetic architecture and heritability of the trait, but also on the GWAS sample size, that is power (*Dudbridge, 2013*).] As we show, however, the qualitative trends reported in *Figure 1* do not depend on the p-value threshold choice (*Appendix 1—figure 3*); moreover, the qualitative trends remain when LDpred is used (with a prior probability of 1 on loci being causal; *Vilhjálmsson et al., 2015*) instead of pruning approaches (*Appendix 1—figure 3*).

These results pertain to three exemplar traits and do not speak to the prevalence of this phenomenon. Nonetheless, they demonstrate that the prediction accuracy of a polygenic score can vary markedly depending on sample characteristics of both the original GWAS and the prediction set, even within a single ancestry, and that this variation in prediction accuracy can be substantial—on the same order as reported for different continental ancestries within the UK Biobank (*Martin et al., 2019*). As one example, the prediction accuracy in East Asian samples, averaged across a number of traits, is about half of that in European samples when GWAS was European-based; when the GWAS is done in the lowest SES group for years of schooling, prediction accuracy in the highest SES group is less than half of that in the lowest SES (*Figure 1C*). Moreover, whereas for these traits, we had prior information about which characteristics may be relevant, other aspects that vary across sets of individuals are undoubtedly important as well (e.g., smoking behavior and diet may modify genetic effects on lipid traits; *Bentley et al., 2019*; *Telkar et al., 2019*), and for other traits of interest, much less may be known a priori.

## Possible explanations for the variable prediction accuracy

Our goal in this paper is to highlight that prediction accuracies can vary across groups of highly similar ancestry, rather than to investigate the likely causes for any particular phenotype. Nonetheless, we provide some observations that may cast light on these results. We first note that in these three examples, the prediction accuracies track SNP heritability differences across strata (*Figure 2A,B,C*). This relationship should be expected, given that the estimation noise decreases with heritability (Appendix 1), and potentially underlies the observation that prediction accuracies using the diverse GWAS sample are often intermediate between those obtained from stratified GWAS samples of equal sample size (*Figure 1*).

Perhaps the simplest explanation for these findings would be that heritabilities, and hence prediction accuracies, vary only because of differences in the extent of environmental variance across strata, while the genetic variance is the same. We can test this hypothesis by examining whether the heritability decreases with increasing phenotypic variance (more precisely whether it is inversely proportional to it), as expected if the genetic variance is fixed across strata. What we find instead is that the estimated SNP heritabilities for all three traits increase or remain the same with increasing phenotypic variance (*Figure 2D,E,F*). Thus, for these traits at least, the variable prediction accuracy is not simply the result of differences in the extent of environmental heterogeneity across strata.

Another possibility is that there is an interaction between genetic effects and sample characteristics, for instance that different sets of genetic variants contribute to blood pressure levels in males and females or to BMI across different stages of life. [Although such interactions could in some contexts be thought of as reflecting GxE, we use the term 'sample characteristic' rather than 'environment', as environment has different meaning across disciplines, referring in some contexts only to factors that are exogenous to genetics. Viewed in this lens, SES in adulthood cannot be interpreted as exogenous, because it is in part determined by educational achievement, which is itself influenced

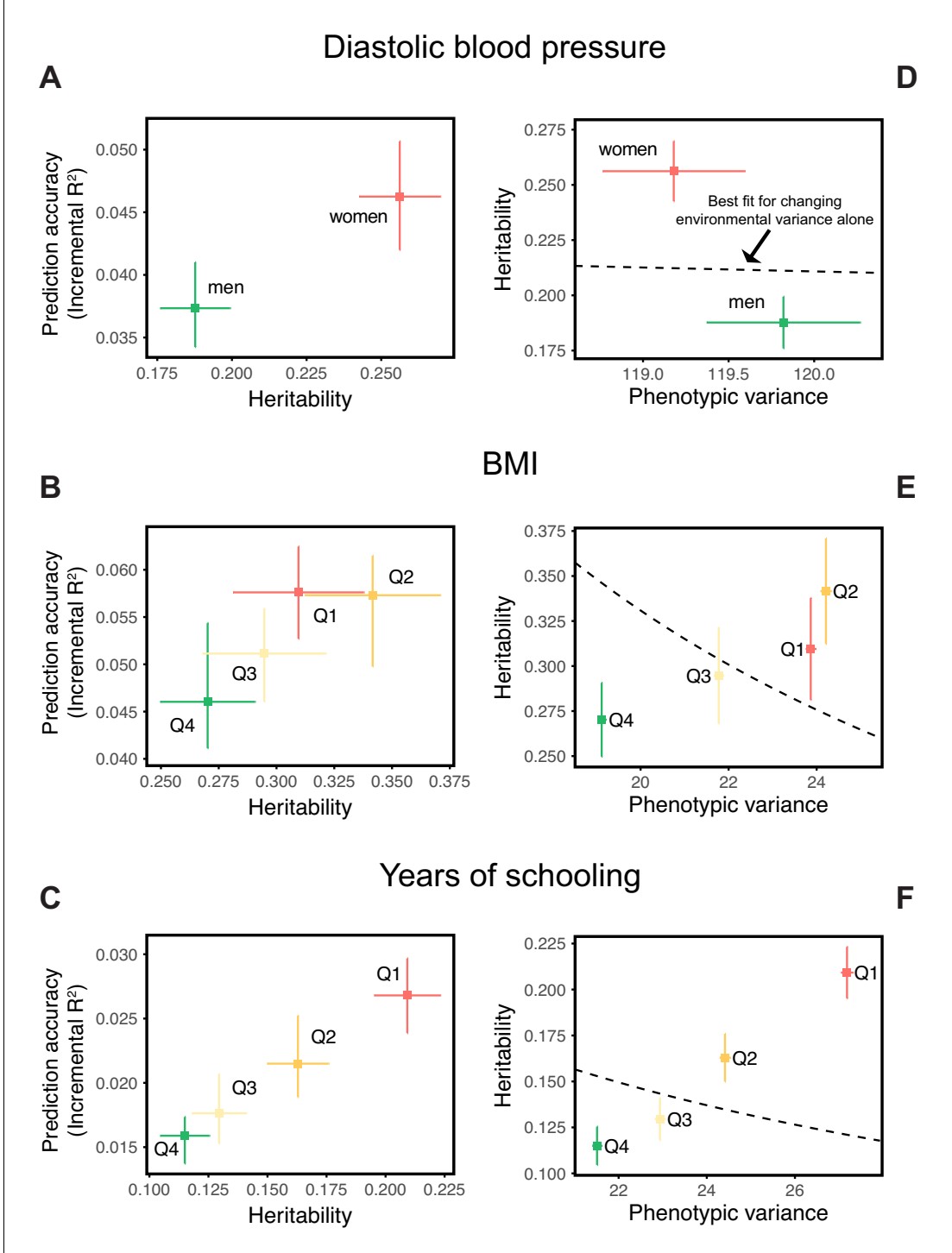

**Figure 2.** Differences in environmental variance alone do not explain the variable prediction accuracy. (A,B,C) The x-axes show heritability estimates (± SE) based on LD score regression in each set. The y-axes show incremental $R^2$ values obtained using the procedure described in *Figure 1*, with GWAS performed in a pooled sample of all strata and testing in stratified prediction sets (see Materials and methods); points and bars show mean and central 80% range computed based on 20 iterations of resampling GWAS and prediction sets. 'Q' denotes quartile of age and SES in (B,E) and (C,F), respectively. (D,E,F) The x-axes show phenotypic variance estimates (± SE) across strata after adjusting for covariates (sex, age and 20 PCs). If the heritability differences across strata are due to differences in environmental variance alone, with genetic variance constant, then heritability should be inversely proportional to phenotypic variance. The best-fitting model for this inverse proportionality (dashed line, simple linear regression) provides a poor fit to the observations.

by genetic factors, and similarly it is questionable whether age or sex are environments.] This explanation is not supported by bivariate LD score regression, which indicates that the genetic correlations across strata are close to 1 (*Appendix 1—table 2*; Materials and methods). Yet when we re-estimate individual SNP effects in the prediction sets for SNPs ascertained in the original GWAS, the estimated effects of trait-increasing alleles are larger in the groups with higher prediction accuracy (*Appendix 1—figure 4*; Materials and methods).

One simple model that could reconcile these findings is if effect sizes are highly correlated across the groups, but systematically larger in those groups with higher prediction accuracy. This explanation is reminiscent of the 'amplification' model of genetic influences on cognition during development (*Briley and Tucker-Drob, 2013*).

Other factors complicate interpretation, however, and may also contribute to our observations. In particular, for the case of years of schooling, conditioning on adult SES induces a form of range restriction, which could contribute to variable prediction accuracy across strata. We note, however, that we see highly variable prediction accuracies across SES strata even when the GWAS is conducted in a diverse sample (i.e., including individuals from all strata) (*Figure 1C*); in that regard, our approach mimics what happens in practice when polygenic scores are used to predict phenotypes in a sample with a smaller range of SES (e.g., *Rimfeld et al., 2018*). More generally, although this type of range restriction is artificially amplified in our example, SES differences may often be a problem for GWAS in which the sample is not representative of the population; for instance, the most recent major GWAS of educational attainment (*Lee et al., 2018*) included numerous medical data sets and the 23andMe data set, which are not representative of the national population.

Another potentially important factor is that the adjustment for PCs may not be a sufficient control for the different ways in which population structure can confound GWAS results (*Vilhjálmsson and Nordborg, 2013*), leading to variable prediction accuracy across strata if they differ in their population structure. To examine this possibility, we repeated the analysis in *Figure 1* but using a linear mixed model (LMM) approach (including PCs among other covariates; see Materials and methods), and obtained qualitatively similar results (*Appendix 1—figure 5*). Although not a perfect fix (*Listgarten et al., 2013*; *Mathieson and McVean, 2013*), the fact that we obtain similar results using PCs and LMM suggests that confounding due to population stratification in the UK Biobank alone does not explain the variable prediction accuracies across strata.

## Obstacles to portability explored through a comparison of standard and family-based GWAS

Beyond sample characteristics such as age or sex, a number of other factors may shape the portability of scores across groups of similar ancestry. Standard GWAS is done in samples of individuals that deliberately exclude close relatives; as implemented, it detects direct effects of the genetic variants, but also any indirect genetic effects of parents, siblings, or peers, effects of assortative mating among parents, and potentially environmental differences associated with fine-scale population structure (*Young et al., 2018*; *Trejo and Benjamin, 2019*; *Kong et al., 2018*; *Lee et al., 2018*; *Berg et al., 2019*). Given that many of these effects are likely to be culturally mediated (*Stulp et al., 2017*; *Selzam et al., 2019*), it seems plausible that they may vary within as well as across groups of individuals with different ancestries. If culturally-contingent effects contribute to GWAS estimates (and hence to PGS), they may lead to differences in the prediction accuracy in samples unlike the original GWAS.

To demonstrate that these considerations are not just hypothetical, we compared the prediction accuracy when the PGS is trained on 'unrelated' individuals such as those used in a standard GWAS to one obtained from a sibling-based (or 'sib-based') GWAS (Materials and methods). In the latter, genotype differences between sibs, a result of random Mendelian segregation in the parents, are tested for association with the phenotypic differences between them. Because the tests depend on phenotypic differences between siblings who, of course, have the same parents, these tests are conditioned on the parental genotypes and hence exclude many of the indirect effects signals that may be picked up in standard GWAS (Appendix 1). Differences between standard and sib-based GWAS are thus informative about the presence of factors other than direct genetic effects (*Wood et al., 2014*; *Trejo and Benjamin, 2019*; *Lee et al., 2018*; *Berg et al., 2019*; *Selzam et al., 2019*).

A challenge in this comparison is that the UKB contains only ~22K sibling pairs, ~19K of whom are labeled as 'White British' (WB). The siblings are similar to the unrelated individuals in terms of ages,

SES distributions and genetic ancestries (*Appendix 1—figures 6* and *7*) but include a higher proportion of females; this difference is unlikely to influence our analyses (see below). While a large number, 19K pairs is still too few to have adequate power to discover trait-associated SNPs, when compared to a standard GWAS using the much larger sample of unrelated WB individuals (~340K).

To increase power and enable a direct comparison between the two designs, we split the SNP ascertainment and effect estimation steps as follows (*Figure 3A*): we identified SNPs using a standard GWAS with a large sample size (median ~270K across the traits considered) (see Materials and methods). We then estimated the effect of each significant SNP using (i) a sib-based association test and (ii) a standard association test. We chose the size of the estimation set in (ii) such that the median standard error of effect estimates in (i) and (ii) is approximately equal. We then compared the prediction accuracy of the two PGS obtained in this way ('standard PGS' and 'sib-based PGS') in an independent prediction set of unrelated individuals; as we show in Appendix 1, our approach leads to highly similar prediction accuracies of the two approaches under a model with direct effects only (see Materials and methods for details). A further advantage is that the two scores are compared for the same set of SNPs, such that LD patterns and allele frequency differences do not come into play.

We applied the approach to 20 traits, focusing on traits with relatively high heritability estimates as well as social and behavioral traits that have been the focus of recent attention in social sciences. For the majority of the traits, such as diastolic blood pressure, BMI, and hair color, the prediction accuracies of standard and sib-based PGS were similar (*Figure 3B*), as expected under standard GWAS assumptions and as observed for traits simulated under these assumptions (*Appendix 1—figure 8*). However, for height and for a range of social and behavioral traits, such as years of schooling, pack years of smoking and household income, the prediction accuracy of the sib-based PGS was substantially lower than that of the standard PGS (*Figure 3B*). [We caution that, because the first step of our study design is to identify SNPs that are associated with the trait in a large set of unrelated individuals and we subsequently match the sampling variances of sib- and standard GWAS, rather than identify distinct sets of SNPs separately in the two designs, the ratio of prediction accuracies that we obtain cannot be directly compared to those reported in other studies.]

A number of factors could contribute to the differences between prediction accuracies for PGS based on sibs versus unrelated individuals, including confounding effects of population stratification, indirect genetic effects from parents and assortative mating. The relative importance of each factor will vary across traits (*Rosenberg et al., 2019*; *Kong et al., 2018*; *Haworth et al., 2019*; *Ruby et al., 2018*; *Selzam et al., 2019*). For educational attainment, this gap is likely to reflect at least in part the documented contribution of indirect genetic effects to the standard PGS (*Lee et al., 2018*; *Kong et al., 2018*; *Young et al., 2018*). We show in Appendix 1 that in the presence of indirect genetic effects mediated through parents, standard PGS outperforms sib-based PGS unless direct and indirect effects are strongly anticorrelated (*Appendix 1—figure 9*), which seems unlikely to be the case for years of schooling. The difference in the performance of sib-based and standard PGS observed for other social and behavioral outcomes, such as household income and age at first sexual intercourse (*Figure 3B*), may reflect a similar phenomenon. An additional contribution to divergent prediction accuracies could come from indirect effects among siblings, which would also contribute differentially to standard and sibling-based PGS. For height, there may be an important contribution of assortative mating to the difference in prediction accuracies (*Wood et al., 2014*; *Robinson et al., 2017*; *Lee et al., 2018*). In Appendix 1, we show that under a simple model of positive assortative mating, the prediction accuracy based on a standard PGS is higher than that of a sib-based PGS (*Appendix 1—figure 10*). We further confirmed that the difference in the sex ratio of the siblings and unrelated individuals, mentioned earlier, has a negligible effect on these differences, though it may underlie the slightly lower prediction accuracy of the standard PGS for pulse rate (*Appendix 1—figure 11*).

The lower prediction accuracies for PGS based on sib-based GWAS indicate that complications such as assortative mating or indirect effects contribute to the standard GWAS estimates. In the absence of these complications, we ensure that prediction accuracies are comparable by matching the sampling errors of the two approaches (*Figure 3A*). In the presence of these complications, the magnitude of the ratio of prediction accuracies should reflect the strength of assortative mating, the relative contribution of indirect genetic effects compared to direct effects, and so forth. However, interpreting the magnitude of the deviation from 1 is far from straightforward: as we show in

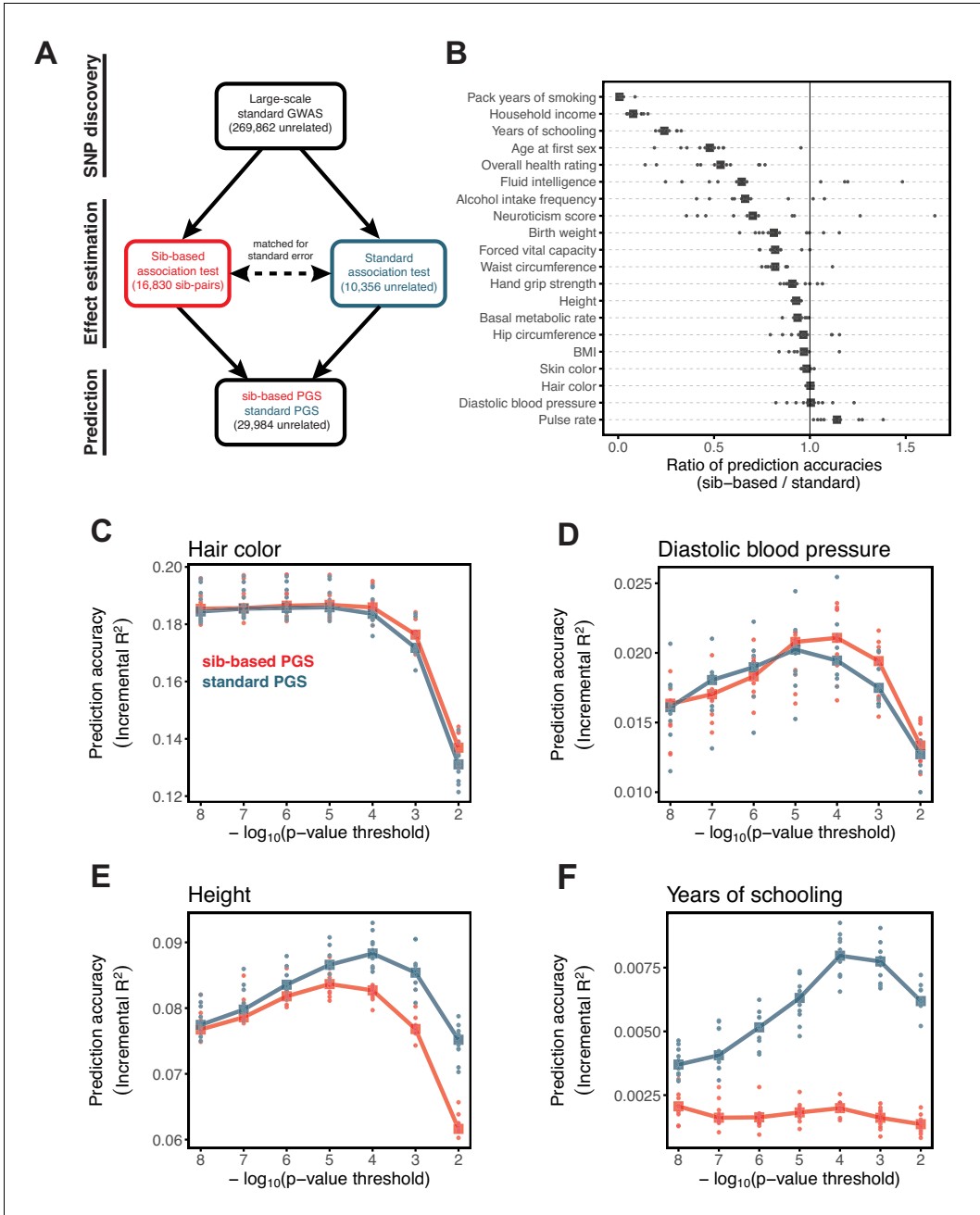

**Figure 3.** Comparison of prediction accuracy of standard and sib-based polygenic scores. (**A**) After ascertaining SNPs in a large sample of unrelated individuals, we estimated the effects of these SNPs with a standard regression using unrelated individuals and, independently, using sib-regression. We then used the polygenic scores for prediction in a third sample of unrelated individuals. We chose the sample size of the standard PGS estimation set such that median effect estimate SEs are equal in the two designs, thereby ensuring equal prediction accuracy under a vanilla model with no indirect effects or assortative mating. Numbers in parentheses are median sample size in each set across 20 traits (see Materials and methods and *Appendix 1—table 1* for the definition of each trait, and *Appendix 1—table 3* for sample sizes for each trait). (**B**) Ratio of prediction accuracy in the two designs across 20 traits. For each trait, we performed 10 resampling iterations of unrelated individuals into three sets for discovery, estimation and prediction (small points). Large points show median values. (**C-F**) We repeated this procedure with different discovery-set p-value thresholds for including a SNP in the polygenic score. The higher the p-value threshold is, the more SNPs are included. For each p-value threshold, points show 10 iterations as described and large points show median values. Shown are a subset of traits, with traits appearing in (**B**) but not shown here presented in *Appendix 1—figure 12*.

Appendix 1, the relative difference in prediction accuracies between the two approaches stems in part from the noise-to-signal ratio for the effect estimates in sib-based versus standard GWAS (Appendix 1, *Appendix 1—figures 9* and *10*), and as a result also depends on features of the comparison like the sample sizes used and the PGS model.

Motivated by these considerations, we examined how the prediction accuracy varies when progressively relaxing the GWAS p-value threshold for inclusion of SNPs, that is when including more weakly associated SNPs in the PGS. [In *Figure 3B*, results are shown for the p-value threshold that maximizes the prediction accuracy of the standard PGS, replicating the practice when comparing populations of different ancestry; *Martin et al., 2019*.] For hair color and diastolic blood pressure, there is little to no difference in prediction accuracy between the two estimation methods, regardless of the number of SNPs included in the score (*Figure 3C,D*). In contrast, for height, standard and sib-based PGS perform similarly when based on the most significantly associated SNPs, but standard PGS progressively outperforms sib-based PGS when more SNPs are included (*Figure 3E*). Similarly, the difference in prediction accuracy between sib-based and standard PGS changes markedly for years of schooling, household income and other social and behavioral traits (*Figure 3F* and *Appendix 1—figure 12*). The growing gap in performance with increasing p-value threshold likely reflects a combination of an increasing noise-to-signal ratio for the effect estimates in sib-based versus standard GWAS (see Appendix 1) and changes in the relative importance of direct effects versus other factors such as indirect parental effects and assortative mating.

In summary, the differences between the prediction accuracies of standard and sib-based PGS seen for a number of traits (*Figure 3B*), notably social and behavioral ones, demonstrate that standard GWAS estimates often include a substantial contribution of factors other than direct effects. In these cases, even if the power to detect direct effects were comparable, standard GWAS would lead to higher prediction accuracy than sib-GWAS. In some contexts that may be a sufficient reason to rely on PGS derived from standard GWAS. However, that gain stems from the inclusion of factors such as indirect effects and assortative mating that are likely to be modulated by SES, environment and culture (e.g., *Selzam et al., 2019*; *Stulp et al., 2017*). Thus, the increased prediction accuracy likely comes at a cost of not always porting well across groups, even of the same ancestry, in ways that may be difficult to anticipate.

## Discussion

Although the conversation around the portability of PGS has largely focused on genetic ancestries, our results show that prediction accuracy can also differ, in some cases substantially, across groups of similar ancestry—even due to basic study design differences such as age, sex or SES composition. When due only to increased environmental variance, such decreased accuracy may not pose a problem, at least for certain applications. But as we have shown, differences in the degree of environmental variance are not the primary explanation for the patterns we report (*Figure 2*), and other factors, including differences in the magnitude of genetic effects among groups, indirect effects and assortative mating, also lead to differences in the prediction accuracy of PGS, in ways that may make applications of phenotypic prediction less reliable, even within a single ancestry group. For some traits, there is prior information about which factors are likely to be important, but not always, and even for well-studied traits, it may be difficult to enumerate all the influential factors. As an example, we considered the accuracy of the polygenic score for years of schooling and found that it also varies somewhat depending on whether individuals have no sibling or one sibling in the prediction sets (Materials and methods; *Appendix 1—figure 13*).

Following the discussion of portability across ancestries, we have focused on incremental $R^2$ as a measure of portability. This measure is less directly informative when the goal is to use PGS to reliably identify individuals in the tails of the distribution, that is those at elevated risk of developing a disease—the main application of PGS in human genetics, as distinct from social science or evolutionary biology. Nonetheless, the same concerns raised here are likely to apply. To illustrate that point, we considered binary outcomes of the traits considered in *Figure 1*, 'hypertension' (defined as diastolic blood pressure > 110 mmHG), 'obesity' (defined as BMI > 35 kg/m$^2$), and 'college completion', and evaluated the prediction accuracy as measured by incremental AUC (*Appendix 1—figure 14*).The qualitative results are the same as in *Figure 1*. We also examined how incremental AUC varies by sex for five binary disease traits that we chose because they have relatively high heritability.

For three of them, hypothyroidism and two cardiovascular outcomes, prediction accuracy varies depending on both the GWAS and prediction sets (*Appendix 1—figure 15*).

Thus, for both quantitative and binary traits, the question of the domain over which a PGS applies is not just about LD patterns, allele frequencies or GxG effects but also about the extent of environmental and genetic variance, GxE, as well as the contribution of direct effects versus indirect effects, assortative mating and environmental confounding. An important implication is that differences in prediction accuracies among groups with distinct ancestries cannot be interpreted exclusively or even primarily in terms of population genetic parameters when these groups differ dramatically in their SES (*Chetty and Hendren, 2018*; *Conley, 2010*; *Nuru-Jeter et al., 2018*; *Reich, 2017*) and other factors that may affect portability—especially when the relative contribution of these factors to GWAS signals remains unknown (*Young et al., 2019*; *Mills and Rahal, 2019*). Thus, efforts to conduct GWAS in groups that vary in ancestry and geographic locations will need to be accompanied by a careful examination of variation in portability along other dimensions.

While these results raise the question of how to best construct a PGS, the answer is not obvious, and likely depends on the specific trait and samples. For example, for the three cases shown in *Figure 1*, considering a fixed GWAS sample size, the highest prediction accuracy is attained with a GWAS sample limited to some stratum (e.g., women for diastolic blood pressure). Yet a much larger merged data set containing the union of strata generates the most predictive PGS (*Appendix 1—figure 1*). Together, these observations suggest a trade-off between the factors that are shared among strata and lead to increased power with sample size and those that differ across strata and underlie the variable prediction accuracy. In principle then, if influential factors were known, the composition of the GWAS sample could be optimized to yield the highest accuracy in a given prediction set, but how much each stratum should be weighted will depend on a number of factors such as the genetic and environmental variance in each stratum, genetic correlation across strata, and sample sizes. Moreover, factors such as assortative mating and indirect effects are soaked up into the GWAS estimates—and critically also into the SNP heritability estimates. Thus, the choice of a GWAS sample is about more than power; it is implicitly making a choice about all sorts of sample characteristics that may or may not hold true of the prediction set.

In that regard, it is worth noting that while classical twin studies were often constituted to be representative of a reference population (often national in nature) (*Polderman et al., 2015*; *Branigan et al., 2013*), the same is not true of most contemporary human genetic datasets, which are skewed towards medical case-control studies, biobanks that are opt-in (and thus tend to include individuals who are wealthier and better educated than the population average) or direct-to-consumer proprietary genetic databases (which are even more skewed along these dimensions) (*Lee et al., 2018*). For instance, individuals in UK Biobank have higher SES than the rest of the British population (*Fry et al., 2017*) and are presumably self-selected for a certain level of interest in biomedical research. These factors alone raise challenges as to the broad portability of PGS derived from them. More generally, it seems plausible that individuals included in a GWAS differ from those that, for myriad reasons, do not end up participating (*Taylor et al., 2018*), in ways that make it difficult to predict the domain over which GWAS-based estimates can be reliably generalized.

One fruitful way forward may be to study data from related individuals, in which it should be possible to decompose the components of the signals identified in GWAS into direct and indirect effects, the degree of assortative mating and the contribution of residual stratification (*Zhang et al., 2015*; *Young et al., 2018*; *Kong et al., 2018*). Not only will this decomposition help us to better interpret the results of GWAS and the resulting PGS, it will make it possible to examine under which circumstances, and for which phenotypes, components port more reliably to other sets of individuals, both unrelated and related. Ultimately, we envisage that in order to be broadly applicable, GWAS-based phenotypic prediction models will need to include not only a PGS but some study characteristics, other social and environmental measures and, perhaps crucially, their interactions.

## Materials and methods

### UK biobank

The UK Biobank (UKB) is a large study of about half a million United Kingdom residents, recruited between years 2006 to 2010 (*Bycroft et al., 2018*). In addition to genetic data, hundreds of

phenotypes were collected through measurements and questionnaires at assessment centers, and by accessing medical records of the participants.

## Inclusion criteria

In this study, we focused on 408,434 participants who passed quality control (QC) measures provided by UKB; specifically, for whom the reported sex (QC parameter 'Submitted.Gender') matched their inferred sex from genotype data (QC parameter 'Inferred.Gender'); who were not identified as outliers based on heterozygosity and missing rate (QC parameter 'het.missing.outliers'==0); and did not have an excessive number of relatives in the database (QC parameter 'excess.relatives'==0). We further selected individuals identified by UKB to be of 'White British' (WB) ancestry (QC parameter 'in.white.British.ancestry.subset'==1), which is a label that refers to those who, when given a set of choices, self-reported to be of 'White' and 'British' ethnic backgrounds and, in addition, were tightly clustered in a principal component analysis of the genotype data, as detailed in *Bycroft et al. (2018)*. We excluded individuals that had withdrawn from the UK Biobank by the time of the analyses here. For a given trait, we further conditioned on individuals for whom the trait value was reported.

## Phenotype data

We focused on 25 traits, including traits with relatively high heritability estimates as well as social and behavioral traits that have been the focus of recent attention in social sciences (see *Appendix 1—table 1* for a complete list of phenotype data used in this work, and their corresponding numeric field codes in the UKB data showcase). We calculated the phenotype 'years of schooling' by converting the maximal educational qualification of the participants to years following *Okbay et al. (2016)* (*Appendix 1—table 4*). For diastolic blood pressure, pulse rate, and forced vital capacity, we took the average of the first two rounds of measurement taken during the same examination at UKB assessment centers. We adjusted the diastolic blood pressure levels for blood pressure lowering medication following *Evangelou et al. (2018)* by shifting the values upward by 10 mmHg for individuals taking medication. For hand grip strength, we took the average of the measurements for the two hands. For categorical phenotypes, we assigned integer values to each category (*Appendix 1—table 1*). For hair color, individuals who reported hair color variable 'Other' were excluded from the analyses. We considered binary traits, 'hypertension' defined as diastolic blood pressure >110 mmHG, 'obesity' defined as BMI >35 kg/m$^2$, and 'college completion' defined based on attainment of a college or a university degree. Disease outcomes were ascertained using self-reported information and/or using the hospital inpatient main and secondary diagnoses coded according to the International Classification of Diseases (ICD-9 and ICD-10). Hypothyroidism, type 2 diabetes, and rheumatoid arthritis were ascertained based on ICD-10 codes of E03.X, E11.X and M06.X, respectively. Myocardial infarction was ascertained based on ICD-9 codes of 410.9, 411.9, 412.9, or ICD-10 codes of I21.X, I22.X, I23.X, I24.1, I25.2 following *Khera et al. (2018)*, or participants with myocardial infarction outcome data among the UK Biobank's algorithmically-defined outcomes. We also considered the binary outcome of ever being diagnosed to have had a heart attack, angina or stroke. For a subset of individuals, multiple measurements of a phenotype were provided, corresponding to multiple visits to UKB assessment centers; in those cases, we used the measurements during the first visit.

## Genotype data

UKB participants were genotyped on either of two similar genotyping arrays, UK Biobank Axiom and UK BiLEVE arrays, at a total of ~850K markers. We focused on autosomal bi-allelic SNPs shared between both arrays, and used *plink v. 1.90b5* (*Chang et al., 2015*) to filter SNPs with calling rate >0.95, minor allele frequency >$10^{-3}$, and Hardy-Weinberg equilibrium test p-val >$10^{-10}$ among the WB samples, resulting in 616,323 SNPs.

## GWAS and trait prediction methods

### GWAS by sample characteristics

We focused on a set of 337,488 WB samples that were identified by the UKB to be 'unrelated' (sample QC parameter 'used.in.pca.calculation'==1 as provided by UKB), defined such that no pairs of

individuals are inferred to be 3rd degree relatives or closer. We split the sample into non-overlapping sets of individuals by one of the following factors: age at recruitment (in years), sex, and Townsend deprivation index at recruitment (used as a proxy for socio-economic status or SES, specifically we take the negative of the Townsend deprivation index as a measure of SES). For SES and age, we divided the sample into four sets: Q1 [minimum value, first quartile], Q2 (first quartile, second quartile], Q3 (second quartile, third quartile], and Q4 (third quartile, maximum value]. We randomly selected 10K samples in each SES and age group, and 20K of males and 20K of females as held-out prediction sets, and performed GWAS using the remaining samples, matching sample sizes across groups in the GWAS set. We performed nine GWASs: for years of schooling in SES Q1 and SES Q4 (sample size 73,283 for each), and in a diverse sample with equal number of individuals from all four groups (sample size 73,280); for body mass index (BMI) in Q1, Q4, and in a diverse sample with equal number of individuals from all four groups (sample size 72,328 for each); and for diastolic blood pressure in males, females, and in a diverse sample with equal number of males and females (sample size 122,774 for each). We performed all GWASs using *plink v. 2.0* (with the flag --linear), adjusting for sex, age (at recruitment) and first 20 PCs as covariates. PCs are principal components of the genotype data, as provided by UKB, calculated using the entire cohort (not just WB individuals). For a subset of cases (where GWAS was performed in samples restricted by characteristics described above), we additionally performed association tests using a linear mixed model (LMM) as implemented in *BOLT-LMM v. 2.3.2* (*Loh et al., 2015*), using LD scores computed from 1000 Genomes European-ancestry samples, with sex, age and first 20 PCs as covariates. The GWAS summary statistics were used to construct PGS for the samples in the prediction sets.

To better understand the performance of PGS across the strata (see 'Possible explanations for the variable prediction accuracy'), we estimated the mean effect sizes of significant SNPs in each of the strata. To avoid overfitting, we first performed an association test in the pooled sample of all strata excluding individuals in the prediction sets and matching the number of individuals per stratum; sample size 293,132 for years of schooling, 272,456 for BMI, and 245,548 for diastolic blood pressure. Then for significantly associated SNPs (LD pruned as described in 'Polygenic score construction and trait prediction'), we re-estimated the effect sizes in each of the strata in the prediction sets (see *Appendix 1—figure 4*). We also used these pooled GWASs to explore the relationship between prediction accuracy and SNP heritability (as shown in *Figure 2*) and with GWAS sample size (*Appendix 1—figure 1*). We performed 20 iterations of all above steps.

In addition to above examples, we explored the prediction accuracy for years of schooling when GWAS and prediction sets are stratified based the participants' number of full siblings. Specifically, we performed GWAS using individuals who had exactly one sibling (sample size 90,417), and evaluated prediction in two independent samples of individuals who reported having no siblings or having one sibling (sample size 20K for each) (see *Appendix 1—figure 13*).

We also considered five binary disease outcomes stratified by sex. Specifically, we performed GWAS in equally sized samples of males and females for hypothyroidism (sample size 135,526), type 2 diabetes (sample size 136,061), rheumatoid arthritis (sample size 136,039), myocardial infarction (sample size 136,061) and having been diagnosed with a heart attack or angina or stroke (sample size 135,833), leaving out 20K samples of males and females for prediction (see *Appendix 1—figures 14* and *15*). For these traits we used a logistic regression model for GWAS (using *plink v. 2.0* with the flag --logistic). An important caveat to analyses of disease outcomes recorded during multiple follow-ups is that for 'age', we could only consider the age at recruitment in the GWAS; that approach is not ideal, considering that a fraction of individuals died during the course of the study (about 20K individuals in the full cohort).

## Standard versus sibling-based polygenic score

We used the genetic relatedness information provided by UKB to infer sibling pairs among the WB samples. Following *Bycroft et al. (2018)*, we marked pairs with $\frac{1}{2^{5/2}} < \phi < \frac{1}{2^{3/2}}$ and IBS0 > 0.0012 as siblings, where $\phi$ is the estimated kinship coefficient and IBS0 is the fraction of loci at which individuals share no alleles. By this approach, we identified 19,329 sibling pairs including 35,634 individuals across 17,328 families. For a given trait, we included pairs with the property that trait values for both individuals were reported. We then formed two sets of individuals: 'Siblings' set, including the sibling pairs randomly sampled to include only one pair per family, and an 'Unrelateds' set, including

the unrelated individuals identified by the UKB (see section 'GWAS by sample characteristics' above), but excluding the Siblings and 6,911 individuals that were related to the Siblings (3rd degree or closer).

We focused on 20 quantitative traits (see *Figure 3B* for the list of traits considered in this analysis) and a number of simulated traits (see below). For each trait, we first downsampled the Unrelateds set to a sample size $n^*$ such that the median standard error of effect estimates roughly matched the median standard error in the sibling-based regression (see '*Estimating* $n^*$' below). We then divided the Unrelateds set into three non-overlapping sets: after sampling $n^*$ individuals (Unrelateds-$n^*$ set), we randomly split the rest of the Unrelateds set into an Unrelateds-prediction set (10% of the samples) to be used as a sample for trait prediction ('prediction set'), and an Unrelateds-discovery set (90% of the samples) to be used for the discovery of trait associated variants (see *Appendix 1—figure 3* for sample sizes in each set). For each trait, we performed standard GWAS in the Unrelateds-discovery set, and ascertained SNPs by thresholding on association p-values. We then estimated the effect sizes for these ascertained SNPs in two ways: by a sibling-based association test in the Siblings set (using *plink v. 1.90b5*'s QFAM procedure with the flag --qfam), and by a standard association test in the Unrelateds-$n^*$ set (using *plink v. 2.0*). Subsequently, for each set of ascertained SNPs in the Unrelateds-discovery set, two PGS were constructed for the samples in the Unrelateds-prediction set (see *Figure 3A* for overview of the pipeline). We performed 10 iterations of the above sampling, ascertainment and estimation steps, except for simulated traits where we performed 30 iterations.

## Estimating $n^*$

In order to compare the performance of sibling-based and standard GWAS designs, we wanted to match both analyses to have similar prediction accuracy under a vanilla model of no assortative mating, population structure stratification or indirect effects. In Appendix 1, we show that this could be achieved by matching median effect estimate standard errors. For each trait, we therefore calculated $n^*$, the sample size of a standard GWAS that yields roughly equal standard errors in the standard and sibling-based regressions. Specifically, for each trait, we first performed sibling-based GWAS in the Siblings using plink's QFAM procedure (with the flag --qfam mperm=100000 emp-se). We then randomly sampled a range of sample sizes from the set of Unrelateds, from 5K to 20K in 1K increments. Following *Wood et al. (2014)*, for each sample size, we performed a standard GWAS, and investigated the linear relationship between the square root of the sample size and the inverse of the median standard error of the effect size estimates. We then used this linear relationship to estimate the sample size of a standard GWAS that corresponds to the inverse of the median standard error of the effect sizes estimate in the sibling-based GWAS.

All standard association tests were performed using *plink v. 2.0* (with the flag --linear), adjusting for sex, age and first 20 PCs as covariates. For sibling-based association tests we first residualized the phenotypic values on age and sex, and then regressed the sibling differences in residuals on sibling genotypic differences using plink's QFAM procedure as described above.

We also considered a version of the analysis described above, in which we first residualized the phenotypes on covariates in the pooled sample of all WB individuals, and then ran the pipeline on the residuals without further adjustment for covariates in the GWAS or prediction evaluation. As shown in *Appendix 1—figure 16*, this approach produced results that are qualitatively the same to what we present in *Figure 3*.

## Simulated traits

We wanted to check that given the study design described above, sibling-based and standard PGS perform similarly with respect to trait prediction, under the vanilla model of no population stratification, assortative mating or indirect genetic effects (*Figure 3*). To this end, we simulated traits with heritability $h^2 = 0.1$ or $0.5$ and either 10K or 100K causal SNPs. For each set of parameters, we simulated three replicates giving a total of 12 simulated traits.

We randomly selected the causal SNPs from a set of 10,879,183 imputed SNPs, considering that most causal variants are plausibly not directly genotyped on SNP arrays. We used a set of SNPs that passed quality control procedures by the Neale lab (http://www.nealelab.is/uk-biobank), namely autosomal SNPs, imputed using the haplotype reference consortium (HRC) panel, which have INFO score > 0.8 and have minor allele frequency > $10^{-4}$; we further limited the SNP set to ones that

were bi-allelic in the WB sample. As in *Martin et al. (2017)*, we randomly assigned effect sizes to these causal SNPs as $\beta \sim N\left(0, \frac{h^2}{m}\right)$, and zero for non-causal SNPs. We then calculated genetic component of the trait, $g$, for all WB samples under an additive model by summing the allelic counts weighted by their effect sizes using plink (with the flag --score). Allelic counts were determined by converting imputation dosages to genotype calls with no hard calling threshold. We also assigned environmental contributions as $\varepsilon \sim N(0, 1 - h^2)$, and then constructed the PGS for each individual,

$$g = \sum_{i=1}^{m} \beta_i X_i,$$

where $X_i$ is the number of minor alleles at SNP $i$ carried by the individual, and the trait value for the individual is calculated as the sum of genetic and environmental contributions:

$$y = \sqrt{h^2}\left(\frac{g - \bar{g}}{\sigma_g}\right) + \sqrt{1 - h^2}\left(\frac{\varepsilon - \bar{\varepsilon}}{\sigma_\varepsilon}\right)$$

where bars represent averages, $\sigma_g$ is the standard deviation of PGS across individuals and $\sigma_\varepsilon$ is the standard deviation of environmental contributions across individuals. These simulated traits were then analyzed using the same pipelines as the other traits (e.g., adjusting for covariates etc.). Importantly, SNP discovery and effect size estimations in GWAS were performed without knowledge of the causal SNPs.

## Polygenic score construction and trait prediction

For all GWAS designs described above, we used p-value thresholding followed by clumping to choose sets of roughly independent SNPs to build PGS. We considered a logarithmically-spaced range of p-values: $10^{-8}$, $10^{-7}$, $10^{-6}$, $10^{-5}$, $10^{-4}$, $10^{-3}$, and $10^{-2}$ (or a subset if no SNP reached that significance level). We then used plink's clumping procedure (with the flag --clump) with LD threshold $r^2 < 0.1$ (using 10,000 randomly selected unrelated WB samples as a reference for LD structure) and physical distance threshold of >1MB. The selected SNPs were then used to calculate PGS for individuals in the prediction sets, by summing the allelic counts weighted by their estimated effect sizes (log of the odds ratios in the case of binary traits) using plink (with the flag --score). In a subset of cases, we also calculated polygenic scores using LDpred assuming all loci are causal (*Vilhjálmsson et al., 2015*). To evaluate prediction accuracy, we calculated the incremental $R^2$: we first determined $R^2$ in a regression of the phenotype to the covariates, and then calculated the change in $R^2$ when including the PGS as a predictor. For binary traits, we calculated the incremental area under the receiver operator curve (AUC).

## Estimating heritability and genetic correlation

We calculated SNP heritability across sex, age and SES groups for diastolic blood pressure, BMI and years of schooling, respectively (as described in the section 'GWAS by sample characteristics') as well as genetic correlations across pairs of groups: we first performed GWAS using all unrelated WB individuals in each group. We then used the GWAS summary statistics to perform LD score regression with LD scores computed from the 1000 Genomes European-ancestry samples (*Bulik-Sullivan et al., 2015*).

## Acknowledgements

This study has been conducted using the UK Biobank resource under application Number 11138, as approved by Columbia University Institutional Review Board, protocol AAAS2914. We are grateful to Daniel Belsky, Jeremy Berg, Graham Coop, Peter Donnelly, Doc Edge, Iain Mathieson, Augustine Kong, Magnus Nordborg, Vincent Plagnol, Guy Sella, Alex Young and members of the Przeworski and Sella labs for valuable discussions and to Doc Edge, Guy Sella, Graham Coop and Magnus Nordborg for comments on a draft of the manuscript. This work was funded by NIH GM121372 to MP, NIH HG008140 to JKP, a Robert Wood Johnson Foundation Pioneer Award (grant number

84337817) to DC and a Junior Fellowship from the Simons Society of Fellows (number 633313) to AH.

# Additional information

## Competing interests

Molly Przeworski: Reviewing editor, *eLife*. The other authors declare that no competing interests exist.

## Funding

| Funder | Grant reference number | Author |
|---|---|---|
| National Institute of General Medical Sciences | GM121372 | Molly Przeworski |
| National Human Genome Research Institute | HG008140 | Jonathan K Pritchard |
| Robert Wood Johnson Foundation | 84337817 | Dalton Conley |
| Simons Foundation | 633313 | Arbel Harpak |

The funders had no role in study design, data collection and interpretation, or the decision to submit the work for publication.

## Author contributions

Hakhamanesh Mostafavi, Arbel Harpak, Conceptualization, Data curation, Software, Formal analysis, Investigation, Visualization, Methodology; Ipsita Agarwal, Formal analysis, Investigation, Visualization; Dalton Conley, Jonathan K Pritchard, Conceptualization; Molly Przeworski, Conceptualization, Resources, Supervision, Methodology, Project administration

## Author ORCIDs

Hakhamanesh Mostafavi (iD) https://orcid.org/0000-0002-1060-2844
Arbel Harpak (iD) https://orcid.org/0000-0002-3655-748X
Ipsita Agarwal (iD) https://orcid.org/0000-0001-8537-0008
Dalton Conley (iD) https://orcid.org/0000-0002-5174-7222
Jonathan K Pritchard (iD) http://orcid.org/0000-0002-8828-5236
Molly Przeworski (iD) https://orcid.org/0000-0002-5369-9009

## Ethics

Human subjects: This study has been conducted using the UK Biobank resource under application Number 11138, as approved by Columbia University Institutional Review Board, protocol AAAS2914.

## Decision letter and Author response

Decision letter https://doi.org/10.7554/eLife.48376.sa1
Author response https://doi.org/10.7554/eLife.48376.sa2

# Additional files

## Supplementary files

• Transparent reporting form

## Data availability

The GWAS summary statistics generated in this study have been uploaded to Dryad.

The following dataset was generated:

| Author(s) | Year | Dataset title | Dataset URL | Database and Identifier |
|---|---|---|---|---|
| Mostafavi H, Harpak A, Agarwal I, Conley D, Pritchard JK, Przeworski M | 2019 | Variable prediction accuracy of polygenic scores within an ancestry group | https://doi.org/10.5061/dryad.66t1g1jxs | Dryad Digital Repository, 10.5061/dryad.66t1g1jxs |

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

## Appendix 1

# 1 Prediction accuracies of polygenic scores based on standard and sib-GWAS

## 1.1 Overview of derived results

In the main text, we compare the prediction accuracies of polygenic scores (PGS) based on a standard GWAS of unrelated individuals and a GWAS based on sibling differences, for a number of traits. Here, we describe how this comparison is implemented, and how indirect effects and assortative mating manifest in this comparison.

### Matching standard and sib-based prediction accuracies

Current standard GWAS are based on huge sample sizes, leading to less noisy estimates than are afforded by family association studies such as those based on sib-differences, which are typically much smaller. This difference in precision needs to be taken into account in making comparisons between the prediction accuracy of scores derived from the two approaches. We show that under a vanilla additive model with no assortative mating, indirect effects, population structure (or other complications), and if the standard GWAS is subsampled to a sample size

$$n^* \approx \frac{1}{1+(1-h^2)(1-2\rho_{sib})}n^{pairs},$$

where $n^{pairs}$ is the number of sib pairs, $h^2$ is the heritability and $\rho_{sib}$ is the correlation in environmental effects experienced by siblings, the two study designs are expected to have the same (out-of-sample) prediction accuracy (see Section 1.2). This analytic result is not that useful in practice, however; in particular, it requires prior knowledge about the extent to which environmental effects correlate among siblings. Instead, we took an empirical approach to match the prediction accuracies in the two approaches: following *Wood et al. (2014)*, we subsampled the regular GWAS to match the median standard errors of the sib-GWAS. As we show in Section 1.2.3, under our vanilla model, we then expect equal out-of-sample prediction accuracies for polygenic scores derived from the two study designs.

### Indirect parental effects

In the presence of indirect parental effects, the out-of-sample prediction accuracy takes a simple form. For a polygenic score based on a standard GWAS, we obtain

$$E[R^2_{ur}] = \tau^2 \frac{1}{1+c},$$

where $\tau^2$ is the ratio of the variance in the trait due to both direct effects and indirect effects of transmitted parental alleles over the total phenotypic variance; and $c$ is a term representing the noise-to-signal ratio in a standard GWAS. For the polygenic score based on sib-GWAS, we obtain

$$E[R^2_{sib}] = (1+\rho\frac{\sigma_\eta}{\sigma_\beta})^2 h^2_\beta \frac{1}{1+c\tau^2/h^2_\beta}.$$

where $\sigma^2_\beta$ and $\sigma^2_\eta$ are the variances of random direct and indirect effects, respectively, $\rho$ is the correlation between direct and indirect effects, and $h^2_\beta$ is the proportion of the phenotypic variance explained by direct effects. Our results suggest that under plausible conditions, the presence of indirect effects would lead to higher prediction accuracy in a standard GWAS. This result holds whether direct and indirect effects are positively correlated, uncorrelated or even somewhat negatively correlated (*Appendix 1—figure 9*).

## Assortative mating

We investigated several models of assortative mating by simulation. Standard GWAS-based polygenic scores have greater prediction accuracies than those based on sib-GWAS when the parental phenotypes are positively correlated, and the reverse is true if they are negatively correlated (*Appendix 1—figure 10 A,B*). The relative difference in prediction accuracies of the two study designs grows with the inclusion of more SNPs in the polygenic score model (*Appendix 1—figure 10 D,F*).

In our analytic model, we ignored the ascertainment step of our study design, in which it is decided which SNPs to include in the polygenic score. We assumed that SNPs are pre-ascertained and that the set of ascertained SNPs includes all causal ones. In a subset of simulations, we implemented the ascertainment step based on an independent simulated GWAS (see below). In both settings, we refer (somewhat loosely therefore) to the regression on ascertained SNPs in a sample of unrelated individuals as 'standard GWAS' and the regression of the difference in phenotypes on the difference in sib genotypes as 'sib-GWAS.'

## 1.2 Picking the sample size of the standard GWAS to match the prediction accuracy of the score based on the sib-GWAS

We look for the sample size $n^*$ of a standard GWAS performed on sample of unrelated individuals such that, under our vanilla model, the resulting polygenic score has the same (out-of-sample) prediction accuracy as the polygenic score obtained from a sib-GWAS with sample size $n_{pairs}$. We begin by assuming that all causal sites $i$ are known; that they are unlinked; that they have only additive, direct effects on the phenotype; and that there is no population stratification or assortative mating. We first find the sampling variance of the effect size estimate for a single site obtained from each of the two study designs. We then examine (and ultimately match) the prediction accuracy of the polygenic scores obtained from effect sizes estimated in the estimation sets, $\hat{\beta}_{ur}, \hat{\beta}_{sib}$, on a new, independent prediction sample of unrelated individuals $\{(x', y')\}$.

## 1.2.1 Sampling error of the estimated effect size at a single site

Our model for the phenotypic value $y$ is

$$y = g + e$$

where $e$ is a Normally distributed environmental effect (which includes all sources of random noise) and

$$g = \beta_0^{ur} + \sum_i \beta_i x_i$$

where $x_i \in \{0, 1, 2\}$ are random genotypes. The genotype is coded as the the number of alleles with effect $\beta_i$ carried by the individual at site $i$. Effect sizes $\beta = \{\beta_i\}$ are treated as fixed parameters throughout (except when noted otherwise in the very last step leading to *Equation 23*). We can rewrite our model to focus on the effect size at a single site $i$:

$$y = \beta_0 + \beta_i x_i + \epsilon_i, \tag{1}$$

where

$$\epsilon_i = g - \beta_i x_i + e,$$

with variance

$$Var[\epsilon_i] = Var[g - \beta_i x_i] + Var[e] = Var[y] - \beta_i^2 Var[x_i]$$

In an OLS regression, the standard error for the effect of an allele at site $i$ is

$$Var[\hat{\beta}_i^{ur}] = \frac{Var[\epsilon_i]}{(n-1)Var[x_i]} = \frac{Var[y] - \beta_i^2 Var[x_i]}{(n-1)Var[x_i]}, \quad (2)$$

where $n$ is the sample size and $\hat{\beta}^{ur}$ denotes that the estimate was obtained using a sample of unrelated individuals. In sib-GWAS, our model for site $i$ is

$$\Delta y = \beta_0^{sib} + \beta_i \Delta x_i + \Delta \epsilon_i,$$

with variance

$$Var[\Delta \epsilon_i] = Var[\Delta g - \beta_i \Delta x_i] + Var[\Delta e] =$$

$$Var[\Delta g] + \beta_i^2 Var[\Delta x_i] - 2\beta_i^2 Var[\Delta x_i] + Var[\Delta e].$$

Recall that for siblings (denoted with subscripts $A$ and $B$), we expect

$$Cov[x_{i,A}, x_{i,B}] = \frac{1}{2} Var[x_i],$$

$$Cov[g_A, g_B] = \frac{1}{2} Var[g].$$

Plugging these back in, we obtain

$$Var[\Delta \epsilon_i] = Var[g] - \beta_i^2 Var[x_i] + 2Var[e](1 - \rho_{sib})$$

where $\rho_{sib} = Cor[e_A, e_B]$ is the correlation in environmental effects between sibs. The variance of the estimated effect size in sib-GWAS is therefore

$$Var[\hat{\beta}_i^{sib}] = \frac{Var[\Delta \epsilon_i]}{(n_{pairs} - 1)Var[\Delta x_i]} = \frac{Var[y] - \beta_i^2 Var[x_i] + Var[e](1 - 2\rho_{sib})}{(n_{pairs} - 1)Var[x_i]}. \quad (3)$$

## 1.2.2 Sample size required for equal prediction accuracy

We measure prediction accuracy as the expected squared correlation between polygenic scores $\hat{g}$ and phenotypic values in an independent prediction set of unrelated individuals, denoted $\{(x', y')\}$,

$$E_{\{(x',y')\}}[R^2] = \frac{Cov^2[\hat{g}(x'), y']}{Var[y']Var[\hat{g}(x')]},$$

To incorporate randomness both in the estimation set (summarized by the Multivariate Normal distribution of $\hat{\beta}$) and the prediction set $\{(x', y')\}$, we will require

$$E_{\hat{\beta}^{ur}(n^*)}[E_{\{(x',y')\}}[R^2]] \overset{!}{=} E_{\hat{\beta}^{sib}(n^{pairs})}[E_{\{(x',y')\}}[R^2]]$$

where $\hat{\beta}(n)$ is a set $\{\hat{\beta}_i\}$ estimated in a GWAS with sample size $n$. Equivalently,

$$E_{\hat{\beta}^{sib}}\left[\frac{Cov^2[\hat{g}_{sib}(x'), y']}{Var[\hat{g}_{sib}(x')]}\right] \overset{!}{=} E_{\hat{\beta}^{ur}}\left[\frac{Cov^2[\hat{g}_{ur}(x'), y']}{Var[\hat{g}_{ur}(x')]}\right], \quad (4)$$

where we left out the sample sizes for brevity, and $Var[y']$ was cancelled out. Finally, we can replace *Equation 4* by its first order Taylor approximation to get the requirement

$$\frac{E_{\hat{\beta}}[Cov_{\{(x',y')\}}[\hat{g}_{sib}(x'), y']]^2}{E_{\hat{\beta}}[Var_{\{(x',y')\}}[\hat{g}_{sib}(x')]]} \overset{!}{=} \frac{E_{\hat{\beta}}[Cov_{\{(x',y')\}}[\hat{g}_{ur}(x'), y']]^2}{E_{\hat{\beta}}[Var_{\{(x',y')\}}[\hat{g}_{ur}(x')]]}. \quad (5)$$

We solve *Equation 4* for a sample size $n^*$ to be used for estimation of the polygenic score

in a standard GWAS that satisfies **Equation 4**. We note that if the vector of estimates $\hat{\beta}$ is given, then

$$
Cov_{\{(x',y')\}}[y', \hat{g}(x')|\hat{\beta}] = Cov_{\{(x',y')\}}[g(x'), g(x') + \sum_i^m x_i'(\hat{\beta}_i - \beta_i)|\hat{\beta}] =
$$
$$
Var_{\{(x',y')\}}[g(x')|\hat{\beta}] + \sum_i^m Cov_{\{(x',y')\}}[\beta_i x_i', (\hat{\beta}_i - \beta_i)x_i'|\hat{\beta}] = \sum_i^m Var[x_i']\beta_i\hat{\beta}_i.
$$

(6)

Since for every $i$, we have

$$
E[\hat{\beta}_i^{ur}] = E[\hat{\beta}_i^{sib}] = \beta_i,
$$

we obtain

$$
E_{\hat{\beta}^{sib}}[Cov[y', \hat{g}_{sib}(x')|\hat{\beta}^{sib}]] = \sum_i^m Var[x_i']\beta_i^2 = E_{\hat{\beta}^{ur}}[Cov[y', \hat{g}_{ur}(x')|\hat{\beta}^{ur}]],
$$

which turns the requirement of **Equation 5** into

$$
E_{\hat{\beta}^{sib}}[Var_{\{(x',y')\}}[\hat{g}_{sib}(x')]] \overset{!}{=} E_{\hat{\beta}^{ur}}[Var_{\{(x',y')\}}[\hat{g}_{ur}(x')]],
$$

or simply

$$
\sum_i^m Var[x_i]Var[\hat{\beta}_i^{ur}] \overset{!}{=} \sum_i^m Var[x_i]Var[\hat{\beta}_i^{sib}].
$$

(7)

Plugging the sampling variance results from **Equation 2** and **Equation 3** into **Equation 7** and reordering, we obtain

$$
\frac{n^* - 1}{n_{pairs} - 1} = \frac{\sum_i^m Var[y] - \beta_i^2 Var[x_i]}{\sum_i^m Var[y] - \beta_i^2 Var[x_i] + Var[e](1 - 2\rho_{sib})},
$$

or, assuming that the trait is polygenic such that $m \gg 1$,

$$
\frac{n^*}{n^{pairs}} \approx \frac{1}{1 + (1 - h^2)(1 - 2\rho_{sib})}.
$$

(8)

**Equation 8** can in principle be applied to the estimation of $\rho_{sib}$ for a given trait, under our model assumptions, and given an independent estimate of $h^2$.

### 1.2.3 Empirical matching of standard errors

The result of **Equation 8** is the same as we would obtain if we required

$$
\forall i \; Var[\hat{\beta}_i^{sib}(x_i)] \overset{!}{=} Var[\hat{\beta}_i^{ur}(x_i^{sib})]
$$

(9)

without taking into account randomness in the prediction set. In practice (and in the results shown in the main text), we have no prior knowledge about $\rho_{sib}$ and instead we find a sample size $n^*$ for the standard GWAS such that

$$
median_{\{sites \; i\}}(Var[\hat{\beta}_i^{sib}(x)]) \overset{!}{=} median_{\{sites \; i\}}(Var[\hat{\beta}_i^{ur}(x)])
$$

(10)

We note that the condition in **Equation 9** is approximately met because, if we assume that $y$ is a highly polygenic trait where

$$
\forall i \; \beta_i^2 Var[x_i] \ll Var[y],
$$

then, if for one site $j$, $n^*$ satisfies

$$
Var[\hat{\beta}_j^{sib}(x)] = Var[\hat{\beta}_j^{ur}(x)] = \frac{D(n^*)}{Var[x_j]}
$$

such that $D(n^*)$ is the same for sib-GWAS and standard GWAS, then for all sites $D(n^*) = \frac{Var[y]}{n^* - 1}$ is the same, namely,

$$\forall i \; Var[\hat{\beta}_i^{sib}(x)] = Var[\hat{\beta}_i^{ur}(x)] = \frac{D(n^*)}{Var[x_i]}$$

*Equation 10* can therefore be thought of as using a weighted-median to estimate $n^*$ where each site $i$ is weighted by $\frac{1}{Var[x_i]}$. In conclusion, the requirement of *Equation 10* leads to equal prediction accuracy of standard and sib-GWAS under the vanilla model assumptions. We note further that in the main text (*Figure 3*), to follow common practice, we use incremental $R^2$ throughout rather than $R^2$. However, as we show in *Appendix 1—figure 16*, using $R^2$ instead gives highly similar qualitative results.

## 1.3 Indirect parental effects

### 1.3.1 Distribution of the effect size estimate at a single site

We consider an additive model with direct effects as well as indirect parental effects, assuming no interaction between the parents and the polygenic score of the children and ignoring possible indirect effects of siblings on each other. The other assumptions from the previous section— for example independent segregation of alleles across sites—remain. We start by considering the model

$$y = \beta_0 + g + n + e$$

where $g$ is the sum of direct effects in an individual with genotype (effect-allele count) $x_i$ at each site $i$,

$$g = \sum_i^m \beta_i x_i,$$

and

$$n = \sum_i^m \eta_i (x_i + \tilde{x}_i^m + \tilde{x}_i^p)$$

is the sum of parental indirect effects, with overall parental effect allele count $x_i + \tilde{x}_i^p + \tilde{x}_i^m$ at each site, where $\tilde{x}_i^m$ is the untransmitted maternal effect allele count, and $\tilde{x}_i^p$ the untransmitted paternal effect allele count, with $\tilde{x}_i^m, \tilde{x}_i^p \in \{0, 1\}$. As we show, when we choose the standard GWAS sample size $n^*$ such that the sampling error of the effect size estimates matches that of the sib-GWAS, the prediction accuracies of the two polygenic scores differ in an independent sample: unless there is a large, negative correlation between indirect and direct effects, the polygenic score from standard GWAS is expected to outperform the one based on sib-GWAS.

We first examine the distribution of an estimated effect size of $x_i$ on the phenotype. The OLS regression for a single site in a standard GWAS follows *Equation 1* and can be rewritten as

$$y = \beta_0 + (\beta_i + \eta_i)x_i + \eta_i(\tilde{x}_i^p + \tilde{x}_i^m) + \epsilon_i \tag{11}$$

with

$$\epsilon_i = g + n + e - (\beta_i + \eta_i)x_i - \eta_i(\tilde{x}_i^p + \tilde{x}_i^m).$$

By the assumption of no assortative mating or other population structure,

$$Cov[\tilde{x}_i^p, \tilde{x}_i^m] = Cov[x_i, \tilde{x}_i^m] = Cov[x_i, \tilde{x}_i^p] = 0. \tag{12}$$

It directly follows that under the generative model specified by *Equation 11*, the OLS regression of $y$ to $x_i$ and $\tilde{x}_i^p + \tilde{x}_i^m$ is a regression involving two independent variables. Therefore, $\hat{\beta}_i^{ur}$ is Normally distributed with expectation

$$E[\hat{\beta}_i^{ur}] = \beta_i + \eta_i.$$

We next calculate the variance of $\hat{\beta}_i^{ur}$. From **Equation 12** and

$$Var[\tilde{x}_i^m + \tilde{x}_i^p] = Var[x_i],$$

we obtain

$$Var[\epsilon_i] = Var[y] + (\beta_i + \eta_i)^2 Var[x_i] + \eta_i^2 Var[x_i] - 2Cov[g + n, (\beta_i + \eta_i)x_i] - 2Cov[n, \eta_i(\tilde{x}_i^m + \tilde{x}_i^p)] =$$

$$= Var[y] - Var[x_i](\beta_i^2 + 2\beta_i\eta_i + 2\eta_i^2).$$

Finally,

$$Var[\hat{\beta}_i^{ur}] = \frac{Var[\epsilon_i]}{(n-1)Var[x_i]} = \frac{Var[y] - Var[x_i](\beta_i^2 + 2\beta_i\eta_i + 2\eta_i^2)}{(n-1)Var[x_i]}. \tag{13}$$

In sib regression, we have

$$\Delta y = \Delta g + \Delta e$$

since indirect parental effects cancel out when taking the difference between siblings (as siblings have the same parental effect allele count). Thus, the expected estimate is the same as it was in the absence of indirect effects. Using the same considerations as in Section 1.2 for the variance in sib differences, we obtain

$$\hat{\beta}_i^{sib} \sim N(\beta_i, \frac{Var[g] - \beta_i^2 Var[x_i] + Var[e](1 - 2\rho_{sib})}{(n_{pairs} - 1)Var[x_i]}),$$

where $\rho_{sib}$ is again the correlation in environmental effects between siblings.

### 1.3.2 Polygenic score prediction accuracy

We now examine the difference in prediction accuracies of $\hat{g}^{ur}$ and $\hat{g}^{sib}$ after matching

$$Var[\hat{\beta}_i^{ur}] \overset{!}{=} Var[\hat{\beta}_i^{sib}] \tag{14}$$

by choosing a standard GWAS sample size $n^*$ that empirically satisfies the condition, as we do in the main text (see also Section 1.2.3).

We can derive the expected prediction accuracy by averaging over both the estimation set (which we again shorthand as the distribution of $\hat{\beta}$) and the prediction set $\{(x', y')\}$. By the law of total expectation,

$$E[R^2] = E_{\hat{\beta}}[E_{\{(x',y')\}}[R^2]] = E_{\hat{\beta}}[\frac{Cov_{\{(x',y')\}}^2[\hat{g}(x'), y'|\hat{\beta}]}{Var_{\{(x',y')\}}[y'|\hat{\beta}]Var_{\{(x',y')\}}[\hat{g}(x')|\hat{\beta}]}] \approx$$

$$\approx \frac{E_{\hat{\beta}}[Cov_{\{(x',y')\}}[\hat{g}(x'), y'|\hat{\beta}]]^2}{Var_{\{(x',y')\}}[y'|\hat{\beta}]E_{\hat{\beta}}[Var_{\{(x',y')\}}[\hat{g}(x')|\hat{\beta}]]}, \tag{15}$$

where the last step is an approximation of the expectation of ratio by its first-order Taylor expansion, a ratio of expectations. The numerator of **Equation 15** is

$$E_{\hat{\beta}}[Cov_{\{(x',y')\}}[\hat{g}(x'), y'|\hat{\beta}]]^2 = E_{\hat{\beta}}[\sum_i^m (\beta_i + \eta_i)\hat{\beta}_i Cov_{\{(x',y')\}}[x_i', x_j'|\hat{\beta}]]^2 =$$

$$= E_{\hat{\beta}}[\sum_i^m Var[x_i](\beta_i + \eta_i)\hat{\beta}_i]^2 =$$

$$= \left( \sum_i^m Var[x_i](\beta_i + \eta_i) E[\hat{\beta}_i] \right)^2. \tag{16}$$

The terms in the denominator of *Equation 15* are

$$Var_{\{(x',y')\}}[y'|\hat{\beta}] = Var[y] \tag{17}$$

and

$$E_{\hat{\beta}}[Var_{\{(x',y')\}}[\hat{g}(x')|\hat{\beta}] = E_{\hat{\beta}}[\sum_i^m Var[x_i]\hat{\beta}_i^2] = \sum_i^m Var[x_i](E[\hat{\beta}_i]^2 + Var[\hat{\beta}_i]). \tag{18}$$

Plugging *Equations 16,17,18* back into *Equation 15*, we obtain

$$E[R^2] \approx \frac{\left( \sum_i^m Var[x_i](\beta_i + \eta_i) E[\hat{\beta}_i] \right)^2}{Var[y] \left( \sum_i^m Var[x_i] Var[\hat{\beta}_i] + \sum_i^m Var[x_i] E[\hat{\beta}_i]^2 \right)}. \tag{19}$$

We note that

$$\tilde{C} := Var[y] \sum_i^m Var[x_i] Var[\hat{\beta}_i]$$

is the same for sib-GWAS and standard GWAS under the requirement of *Equation 14*. We therefore have

$$E[R_{ur}^2] \approx \frac{\left( \sum_i^m Var[x_i](\beta_i + \eta_i)^2 \right)^2}{\tilde{C} + Var[y] \sum_i^m Var[x_i](\beta_i + \eta_i)^2}, \tag{20}$$

and

$$E[R_{sib}^2] \approx \frac{\left( \sum_i^m Var[x_i](\beta_i + \eta_i)\beta_i \right)^2}{\tilde{C} + Var[y] \sum_i^m Var[x_i]\beta_i^2}. \tag{21}$$

If we denote the proportion of the phenotypic variance explained by direct effects by

$$h_\beta^2 := \frac{\sum_i^m Var[x_i]\beta_i^2}{Var[y]},$$

the proportion of the phenotypic variance explained by indirect effects of transmitted parental alleles by

$$\tau_\eta^2 := \frac{\sum_i^m Var[x_i]\eta_i^2}{Var[y]},$$

and the proportion of phenotypic variance explained by both direct and indirect effects of transmitted alleles by

$$\tau^2 := \frac{\sum_i^m Var[x_i](\beta_i + \eta_i)^2}{Var[y]},$$

then *Equation 20* can be written as

$$E[R_{ur}^2] \approx \tau^2 \frac{1}{1+c}, \tag{22}$$

where we defined

$$c := \frac{\sum_i^m Var[x_i] Var[\hat{\beta}_i]}{\sum_i^m Var[x_i](\beta_i + \eta_i)^2}.$$

Here, $c$ can be thought of as a summary of the noise-to-signal ratio, with respect to the signal coming from both direct and indirect effects of transmitted alleles. If we consider effects $\beta$ and $\eta$ as random, treating results obtained thus far as conditional on $\beta$ and $\eta$, and further assume that effects are i.i.d. across sites (implying, in particular, that effect sizes and allele frequencies are independent),

$$\begin{pmatrix} \beta_i \\ \eta_i \end{pmatrix} \sim \left( \begin{pmatrix} 0 \\ 0 \end{pmatrix}, \begin{pmatrix} \sigma_\beta^2 & \rho\sigma_\beta\sigma_\eta \\ \rho\sigma_\beta\sigma_\eta & \sigma_\eta^2 \end{pmatrix} \right),$$

the expectation of the numerator of *Equation 21* is

$$E_{\beta,\eta}[\sum_i^m Var[x_i]\beta_i(\beta_i+\eta_i)|\beta,\eta] = \sum_i^m Var[x_i]E_{\beta_i,\eta_i}[\beta_i^2+\beta_i\eta_i] = \sum_i^m Var[x_i](\sigma_\beta^2+\rho\sigma_\beta\sigma_\eta)$$

and thus *Equation 21*, in expectation, is:

$$E[R_{sib}^2] \approx E_{\beta,\eta}[E[R_{sib}^2|\beta,\eta]] = (1+\rho\frac{\sigma_\eta}{\sigma_\beta})^2 h_\beta^2 \frac{1}{1+c/\alpha}. \tag{23}$$

where

$$\alpha := h_\beta^2/\tau^2 = \frac{\sum_i^m Var[x_i]\beta_i^2}{\sum_i^m Var[x_i](\beta_i+\eta_i)^2}.$$

We examined the fit of this prediction to simulated data. Specifically, we ran simulations to estimate effect sizes in a sib-GWAS and in a standard GWAS, after choosing $n^*$ to match their sampling variances. Finally, we used the polygenic scores to predict phenotypic values in a sample of unrelated individuals (see Section 1.3.3 for further detail).

*Appendix 1—figure 9 A,C,D* show the analytic result alongside simulation results, for different correlation coefficients between indirect and direct effect sizes. Even in the absence of a correlation between indirect and direct effect sizes, the polygenic score based on standard GWAS outperforms the polygenic score based on sib-GWAS.

To understand this behavior and dependency of the $\frac{R_{sib}^2}{R_{ur}^2}$ ratio on other parameters, we divide *Equation 23* by *Equation 22* and obtain

$$E[\frac{R_{sib}^2}{R_{ur}^2}] \approx \frac{E[R_{sib}^2]}{E[R_{ur}^2]} \approx \left(1+\rho\frac{\sigma_\eta}{\sigma_\beta}\right)^2 \alpha \frac{1+c}{1+c/\alpha}.$$

Noting further that

$$\left(1+\rho\frac{\sigma_\eta}{\sigma_\beta}\right)^2 \alpha = \left(\frac{\sigma_\beta+\rho\sigma_\eta}{\sigma_\beta}\right)^2 \frac{\sigma_\beta^2}{\sigma_\beta^2+2\rho\sigma_\beta\sigma_\eta+\sigma_\eta^2} = 1-(1-\rho^2)\frac{\tau_\eta^2}{\tau^2},$$

we obtain

$$E[\frac{R_{sib}^2}{R_{ur}^2}] \approx [1-(1-\rho^2)\frac{\tau_\eta^2}{\tau^2}]\frac{1+c}{1+c\frac{\tau^2}{h_\beta^2}}. \tag{24}$$

A few conclusions emerge from *Equation 24* and the accompanying simulations. First, the sib-GWAS based polygenic score will outperform the standard GWAS-based polygenic score only if direct and indirect effects are strongly negatively correlated (see *Appendix 1—figure 9A-D* for illustration). Second, the term

$$\frac{1+c}{1+c\frac{\tau^2}{h_\beta^2}} = \frac{1+\frac{\sum_i^m Var[\hat{\beta}_i]Var[x_i]}{\tau^2}}{1+\frac{\sum_i^m Var[\hat{\beta}_i]Var[x_i]}{h_\beta^2}} \tag{25}$$

can be interpreted as the dependence on the noise-to-signal ratio (where the signals are the

proportions of phenotypic variance explained by direct and indirect effects of transmitted alleles). For a given sampling variance (matched across the two study designs), the extent of the signal will differ between standard GWAS and sib-GWAS. Importantly, the sampling variance influences the ratio of prediction accuracies. If indirect effects do not exist or make negligible contributions to the trait in question, then the ratio of prediction accuracies is expected to be close to one. In the presence of indirect effects, however, the magnitude of the deviation from one depends on the relationship between direct and indirect effects (and their covariance) as well as on the (matched) sampling variance. Simulations of several parameter combinations suggest that the overall effect of this dependence on the noise-to-signal ratio is a decrease in $R_{sib}^2/R_{ur}^2$ as noise increases; as more SNPs are included in the polygenic scores, the advantage of the standard GWAS-based polygenic score over that of the sib-GWAS grows larger (**Appendix 1—figure 9 E-H**). These considerations inform the interpretation of patterns observed in **Figure 3C–F** of the main text.

### 1.3.3 Simulations of indirect effects

For each set of simulated individuals (discovery, estimation and prediction sets), we first simulated mother-father pairs, assigning parental alleles from $Bernoulli(p_i)$, where $p_i$ denotes the allele frequency at site $i$. We then sampled the parental alleles at random to generate offspring (one offspring per each mother-father pair to simulate a sample of unrelated individuals and two offspring to generate sibling pairs). Phenotypes of the offspring were assigned under an additive model, sampling from a Normal distribution with mean

$$\sum_i^m \beta_i x_i + \eta_i(x_i^p + x_i^m)$$

(where $x_i^m$ and $x_i^p$ are the maternal and paternal effect allele counts, respectively) and variance $\sigma_e^2$, representing the total variance of environmental effects. When there is no correlation between direct and indirect effects, $\sigma_e^2 = 1 - h_\beta^2 - 2\tau_\eta^2$. Using this approach, we generated a set of sibling pairs and estimated SNP effect sizes from these simulated data using a sib-GWAS. We calculated $n^*$ as follows: we simulated sets of unrelated individuals with a range of sample sizes. In each set, we performed a simple linear regression of the phenotypic values on the genotypes. We then estimated a linear relationship between the inverse of the median standard error of effect size estimates (as a dependent variable) and the square root of the sample size. Using this linear relationship, we predicted the sample size for the unrelated set that gives a median standard error equal to the median standard error of sib-GWAS effect size estimates ($n^*$). Finally, we simulated a set of unrelated individuals with sample size $n^*$ and compared the prediction accuracy ($R^2$) of the polygenic score based on standard GWAS on this sample with the one obtained from sib-GWAS.

We additionally investigated the effect of the number of SNPs included in the polygenic scores. For this analysis, we sorted the SNPs based on the association p-value obtained in an independent simulated set of unrelated individuals.

In these simulations, we used the following parameter values:

- The ratio of the phenotypic variance accounted for by direct effects versus by indirect effects ($h_\beta^2/\tau_\eta^2$): 5
- The phenotypic variance explained by offspring and parental alleles, given no correlation between direct and indirect effects ($h_\beta^2 + 2\tau_\eta^2$): 0.25 or 0.5
- The ratio of the variance of direct effects to the variance of indirect effects ($\sigma_\beta^2/\sigma_\eta^2$): 5
- Allele frequencies, $p$, drawn from a truncated exponential distribution, truncated on the left such that the minimum allele frequency is 1%.
- The number of loci, assumed independent (i.e., in linkage equilibrium): 100 (all causal), or 10,000 (all causal) or 10,000 (20% causal)
- SNP effect sizes drawn as

$$\begin{pmatrix} \beta_i \\ \eta_i \end{pmatrix} \sim N\left( \begin{pmatrix} 0 \\ 0 \end{pmatrix}, \begin{pmatrix} \sigma_\beta^2 & \rho\sigma_\beta\sigma_\eta \\ \rho\sigma_\beta\sigma_\eta & \sigma_\eta^2 \end{pmatrix} \right),$$

where $\rho$ is the correlation between direct and indirect effect sizes. Effects sizes were then re-scaled to satisfy $\sum_i^m 2\beta_i^2 p_i(1-p_i) = h_\beta^2$ and $\sum_i^m 2\eta_i^2 p_i(1-p_i) = \tau_\eta^2$. Effects were set to 0 for non-causal loci.

- The number of sibling pairs for sib GWAS: 10,000
- The number of unrelated individuals for prediction: 10,000
- The number of unrelated individuals for discovery GWAS (i.e., to decide which SNPs to include): 20,000
- Number of iterations used to estimate $n^*$ and $R^2$ for a given set of parameters: 10

### 1.4 Assortative mating

We consider assortative mating with regard to a phenotype, whereby the parents of individuals were more likely to mate if they were similar with respect to that phenotype. This process generates a correlation between genetic variants that contribute to the phenotype (i.e., linkage disequilibrium). Consequently, in a standard GWAS, the effect sizes of causal SNPs will partially capture the effect of other causal SNPs as well. Estimated effect sizes are thus expected to be inflated under positive assortative mating (mating of similar individuals) and deflated under negative assortative mating (mating of dissimilar individuals). In turn, in a sib-GWAS, the estimates are in expectation unaffected by assortative mating, because genetic differences between siblings arise from random Mendelian segregation in the parents.

### 1.4.1 Simulations of assortative mating

We used simulations to examine the phenotypic prediction accuracies of polygenic scores based on sib- and standard GWAS under a model with assortative mating (assuming no indirect effects or population stratification beyond assortative mating); to this end, we considered a sample of unrelated individuals, varying the degree of correlation between parental phenotypes $\rho_a$. Similar to our simulations for indirect effects (Section 1.3.3), we first simulated the estimation procedure in a sibling-based and in a standard GWAS (with sample size $n^*$). We then computed the prediction accuracy $R^2$ in an independent sample of unrelated individuals (see 'Further simulation details' below).

We first considered the simple case of a single generation of assortative mating. In the presence of positive assortative mating ($\rho_a > 0$), polygenic scores based on standard GWAS outperform those based on sib-GWAS, whereas the opposite is true in the case of negative assortative mating ($\rho_a < 0$) (*Appendix 1—figure 10 A*). In simulations of two generations of assortative mating, the gap between the prediction accuracies of scores based on standard and sib-GWAS (*Appendix 1—figure 10 B*) widens, suggesting that our qualitative findings apply to scenarios of sustained assortative mating as well.

We further investigated prediction accuracy as a function of the number of SNPs included in the polygenic scores, by progressively increasing the p-value threshold, using p-values obtained from an independent GWAS in unrelated samples (similar to our analysis in *Figure 3*). We considered two genetic architectures scenarios: (i) in which all SNPs are causal and (ii) the case in which 20% of of SNPs are causal (leading polygenic scores to include non-causal SNPs). Under both scenarios, the gap in prediction accuracies between standard and sib-GWAS grows with the number of SNPs (*Appendix 1—figure 10 C-F*).

### Further simulation details

We simulated parental and offspring alleles as described for indirect effects in Section 1.3.3. To mimic assortative mating between parents, we first simulated i.i.d. genotypes (with effect allele counts $x_i$ at each SNP $i$) and randomly assigned 'mother' and 'father' labels to each individual. We then generated corresponding parental phenotypes under an additive model as

$$N(\sum_i^m \beta_i x_i, \sqrt{1 - h^2})$$

where $\beta_i$ is the effect size of SNP $i$, and $h^2$ is the heritability. The same model was used to generate offspring phenotypes.

To mimic the assortative mating process, we induced a given correlation between parental phenotypes, $\rho_a$, by paring mothers and fathers as follows: we first generated a random matrix

$$\begin{pmatrix} u_{m,i} \\ u_{p,i} \end{pmatrix} \sim N\left( \begin{pmatrix} \bar{y}_m \\ \bar{y}_p \end{pmatrix}, \begin{pmatrix} \sigma^2_{y_m} & \rho_a \sigma_{y_m} \sigma_{y_p} \\ \rho_a \sigma_{y_m} \sigma_{y_p} & \sigma^2_{y_p} \end{pmatrix} \right),$$

where $\bar{y}_m$ and $\bar{y}_p$ are the average phenotypes of mothers and fathers, respectively, $\sigma_{y_m}$ and $\sigma_{y_p}$ are the standard deviation of the phenotypes of mothers and fathers, respectively. We then sorted the mothers and fathers sets such that the ranks of values in $y_m$ and $y_p$ match the ranks of values in $u_m$ and $u_p$, respectively, to obtain $cor(y_m, y_p) \approx cor(u_m, u_p) = \rho_a$. In the case of two generations of assortative mating, we simulated the generation of the grandparents similarly. We compared the performance of polygenic scores based on standard and sib-GWAS as described in Section 1.3.3. In the simulations, we used the following parameter values:

- Heritability under random mating ($h^2$): 0.5
- The number of loci, assumed independent (i.e., in linkage equilibrium) under random mating: 10,000 (all causal) or 10,000 (20% causal)
- Allele frequencies, $p$, drawn from a truncated exponential distribution, truncated on the left such that the minimum allele frequency is 1%.
- SNP effect sizes set to 0 for non-causal loci and drawn as $\beta_i \sim N(0, \sigma^2)$, choosing $\sigma^2$ to satisfy $\sum_i^m 2\beta_i^2 p_i(1 - p_i) = h^2$ for causal loci.
- The number of sibling pairs for sib-GWAS: 10,000
- The number of unrelated individuals for prediction: 10,000
- The number of unrelated individuals for discovery GWAS (i.e., to decide which SNPs to include in the polygenic score): 20,000
- The number of iterations used to estimate $n^*$ and $R^2$ for a given set of parameters: 10

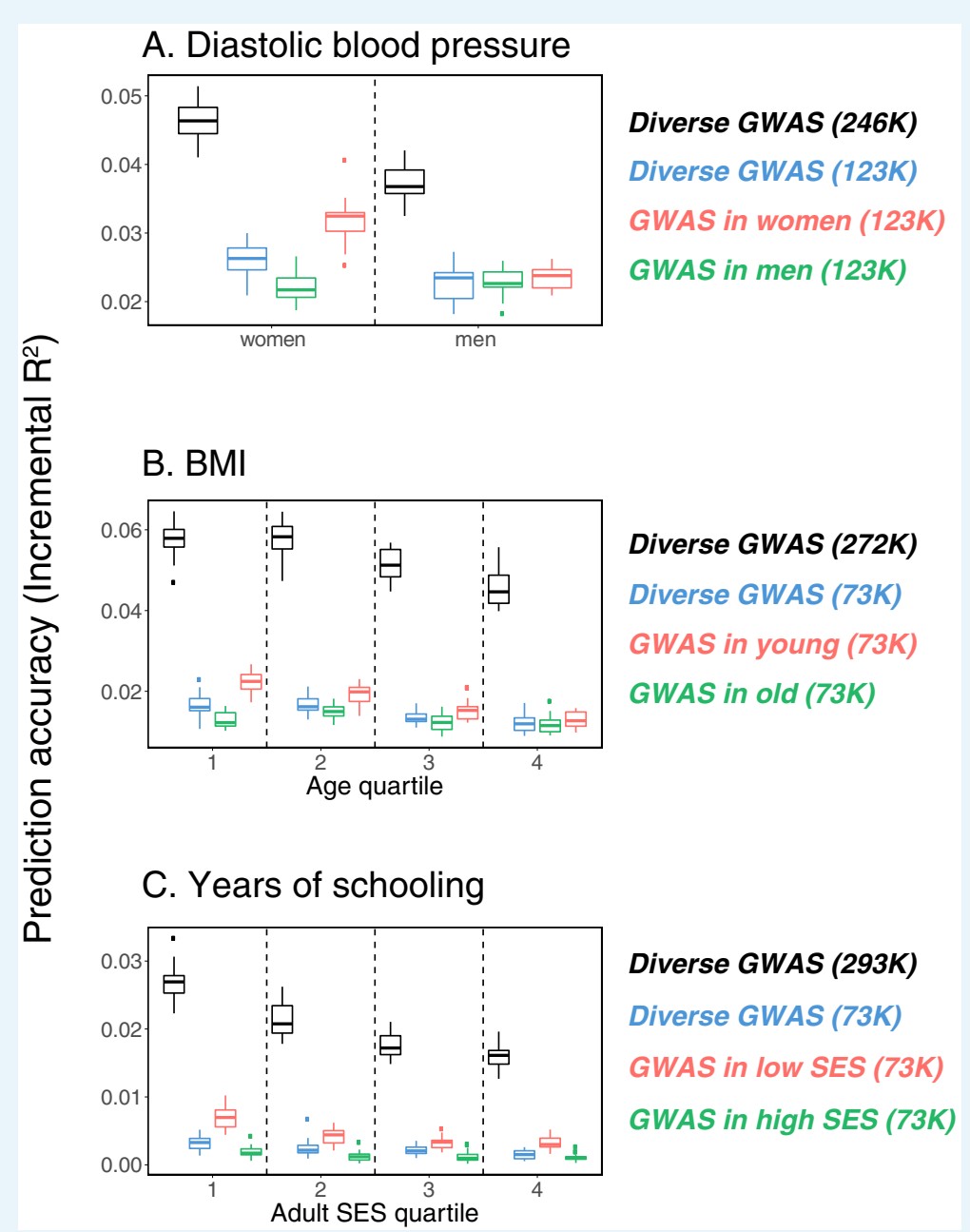

**Appendix 1—figure 1.** Variable prediction accuracy within an ancestry group. This figure extends *Figure 1* of the main text, showing prediction accuracies based on large-scale diverse GWAS that are the union of all strata matching the number of individuals in each stratum. The numbers in parentheses show GWAS sample sizes (see Materials and methods for details). Each box and whiskers plot was computed based on 20 iterations of resampling estimation and prediction sets. Thick horizontal lines denote the medians. The polygenic scores were estimated in samples of unrelated WB individuals. Phenotypes were then predicted in distinct samples of unrelated WB individuals, stratified by sex (**A**), age (**B**) or Townsend deprivation index, a measure of SES (**C**). In red and green cases, polygenic scores are based on a GWAS in a sample limited to one sex, age or SES group (a 'stratum'). In black, polygenic scores are based on a diverse GWAS in a pooled sample of all strata. In blue, polygenic scores are based

on a diverse GWAS in a pooled sample of all strata but downsampled to match the size of the stratified GWAS.

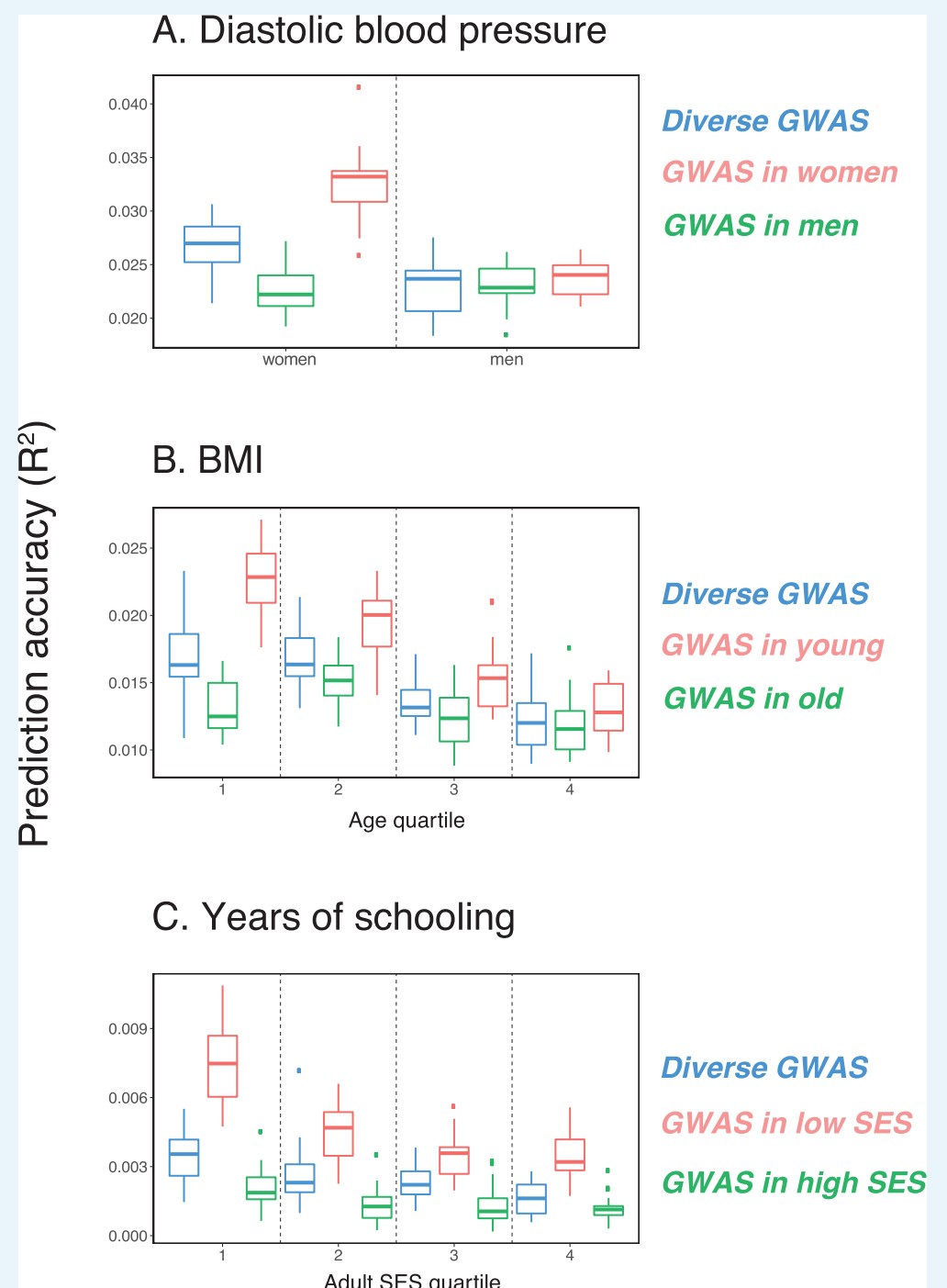

**Appendix 1—figure 2.** Variable prediction accuracy (measured as $R^2$) within an ancestry group. This figure mirrors *Figure 1* of the main text, except for the y-axis showing $R^2$ values (squared correlation between polygenic score and phenotype residualized on covariates), rather than incremental $R^2$. Each box and whiskers plot was computed based on 20 iterations of resampling estimation and prediction sets. Thick horizontal lines denote the medians. The polygenic scores were estimated in samples of unrelated WB individuals. Phenotypes were then predicted in distinct samples of unrelated WB individuals, stratified by sex (**A**), age (**B**) or Townsend deprivation index, a measure of SES (**C**). In red and green cases, polygenic scores

are based on a GWAS in a sample limited to one sex, age or SES group (a 'stratum'). In blue, polygenic scores are based on a GWAS in a diverse sample of all strata downsampled to match the size of the stratified GWAS.

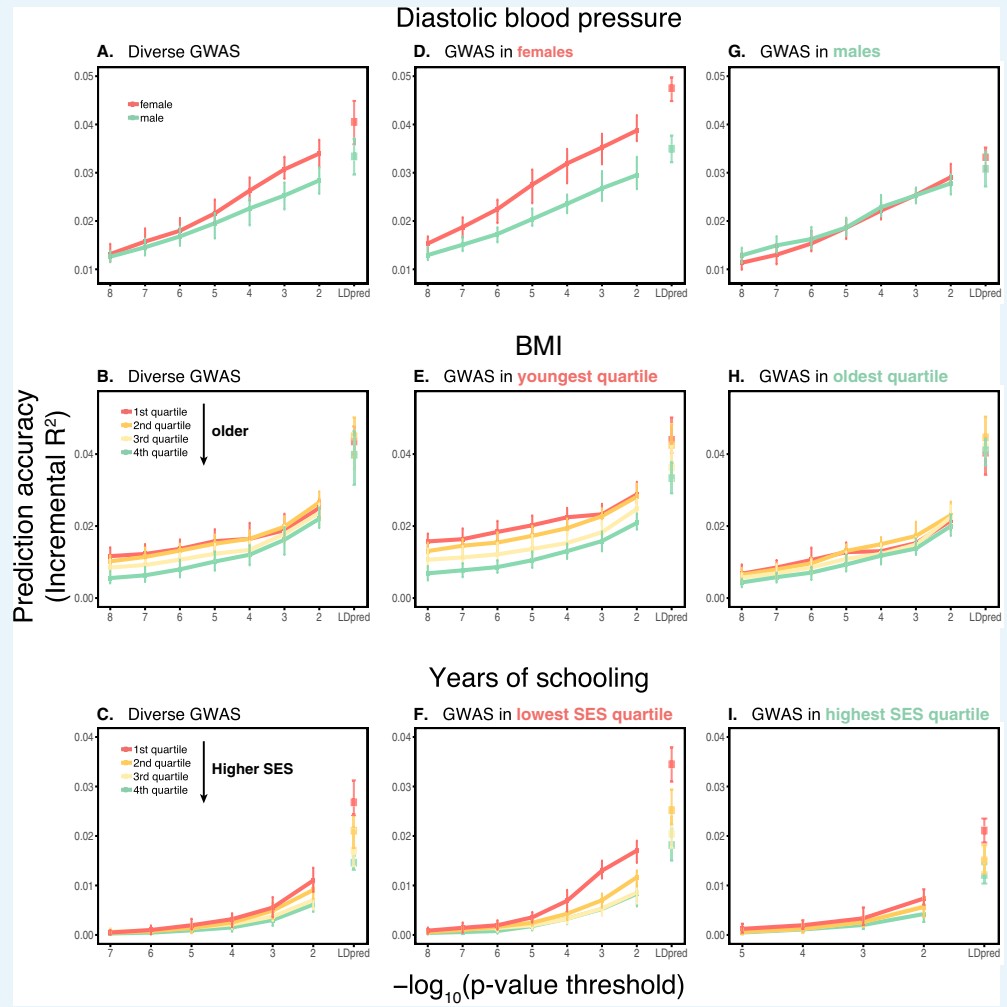

**Appendix 1—figure 3.** Dependence on the polygenic score model. This figure extends *Figure 1* of the main text, showing the prediction accuracies as a function of the p-value threshold for inclusion of a SNP in the polygenic score when based on a pruning and thresholding approach. The higher the p-value threshold is, the more SNPs are included. Last points on the x-axis correspond to a polygenic score model based on the LDpred approach (*Vilhjálmsson et al., 2015*) with a prior probability of 1 on loci being causal. Shown are incremental $R^2$ values in different prediction sets. Points and error bars are mean and central 80% range computed based on 20 iterations of resampling estimation and prediction sets. (**A– C**) The polygenic scores were estimated in samples of unrelated WB individuals. Phenotypes were then predicted in distinct samples of unrelated WB individuals, stratified by sex (**A**), age (**B**) or Townsend deprivation index, a measure of SES (**C**). (**D–I**) Same as in **A-C**, but here the polygenic scores are based on a GWAS in a sample limited to one sex, age or SES group. The trends shown in *Figure 1* of the main text are for p-value threshold of $10^{-4}$, and are qualitatively similar to the trends for other choices of the polygenic score model. For each trait, sample sizes are matched across all GWAS sets.

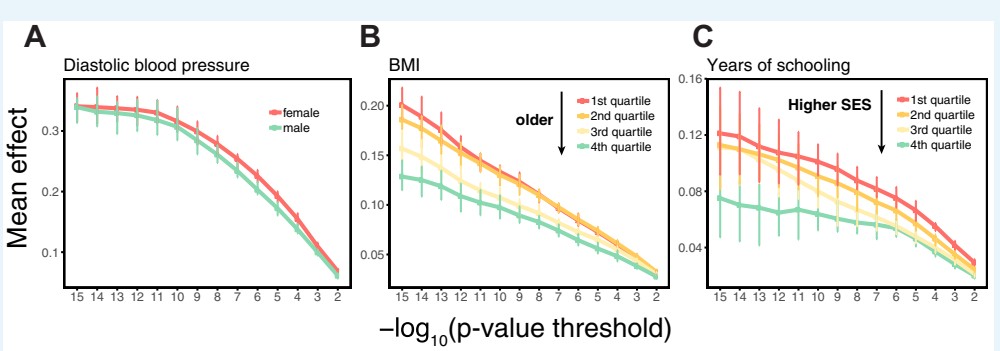

**Appendix 1—figure 4.** Estimating mean effect size across strata. SNPs were ascertained in large samples of unrelated WB individuals. The effects of trait-increasing alleles were then re-estimated in an independent set of unrelated WB individuals (that were excluded from the original GWAS) stratified by sex for diastolic blood pressure (**A**), by age for BMI (**B**) and by Townsend deprivation index, a measure of SES for years of schooling (**C**). Points and error bars are mean and central 80% range computed based on 20 iterations of resampling ascertainment and estimation sets, plotted as a function of the p-value threshold (for p-values obtained in the discovery GWAS).

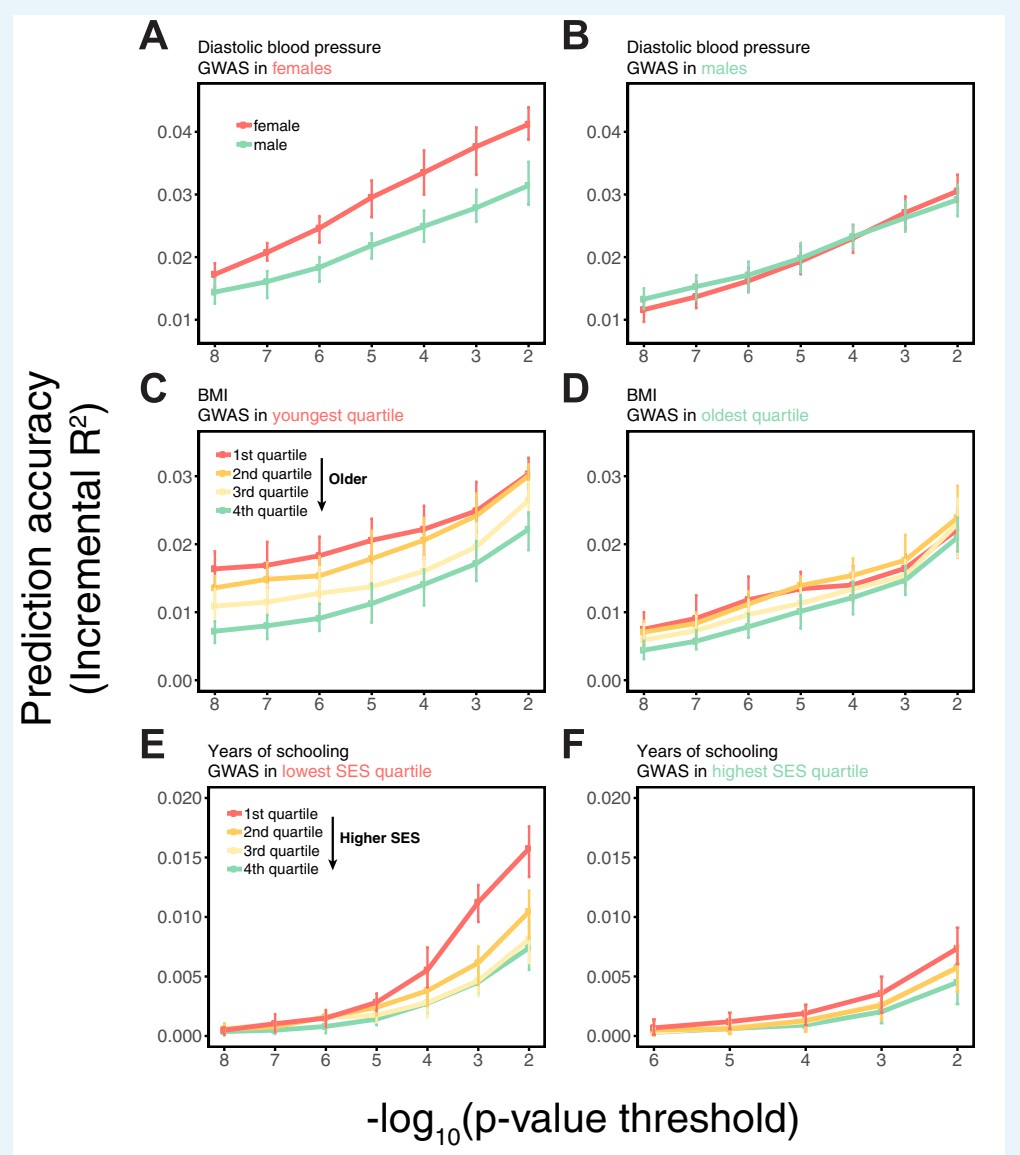

**Appendix 1—figure 5.** Variable prediction accuracy within an ancestry also seen using a linear mixed model. This figure mirrors the last two columns in *Appendix 1—figure 3*, except that here, the GWAS estimates were obtained from a linear mixed model (LMM) (*Loh et al., 2015*). Shown are the prediction accuracies, measured as incremental $R^2$, as a function of the p-value threshold for inclusion of a SNP in the polygenic score. Points and error bars are mean and central 80% range computed based on 20 iterations of resampling estimation and prediction sets. The polygenic scores are based on a GWAS in a sample limited to one sex, age or SES group. Phenotypes are then predicted in distinct samples of unrelated individuals, stratified by sex (**A,B**), age (**C,D**) or Townsend deprivation index, as a measure of SES (**E,F**). The qualitative trends are similar to those in *Appendix 1—figure 3*, which uses a standard linear regression with PCs (principal components of the genotype data) as a control for population structure when testing for an association between the phenotypes and genotypes. The similarity suggests that the observed differences in prediction accuracies across strata are not driven to a large degree by population structure confounding.

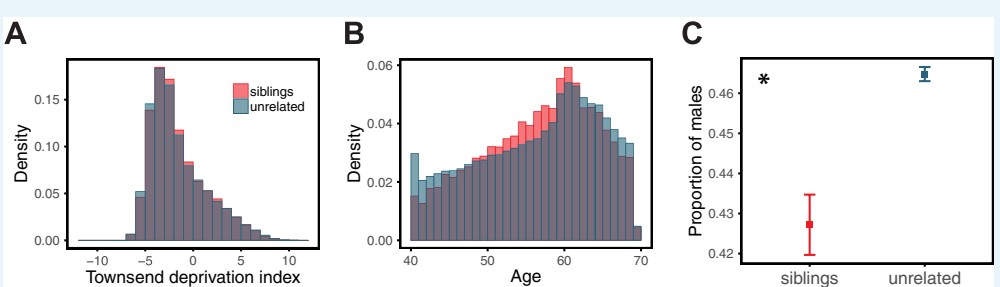

**Appendix 1—figure 6.** Comparison of siblings and unrelated individuals in the UK Biobank with respect to age, SES, and sex ratio. Panels show the distribution of Townsend deprivation index, a measure of SES (**A**), the age distribution (**B**), and the proportion of males (**C**) for the siblings and unrelated sets used in the analysis described for *Figure 3* of the main text. For each sibling pair, one sibling was randomly selected for these comparisons. The asterisk symbol marks a significant difference at the 1% level between siblings and unrelated individuals, as assessed by a Mann-Whitney test. SES and age distributions are quite similar in siblings and unrelated sets, whereas the proportion of males is significantly smaller in the siblings.

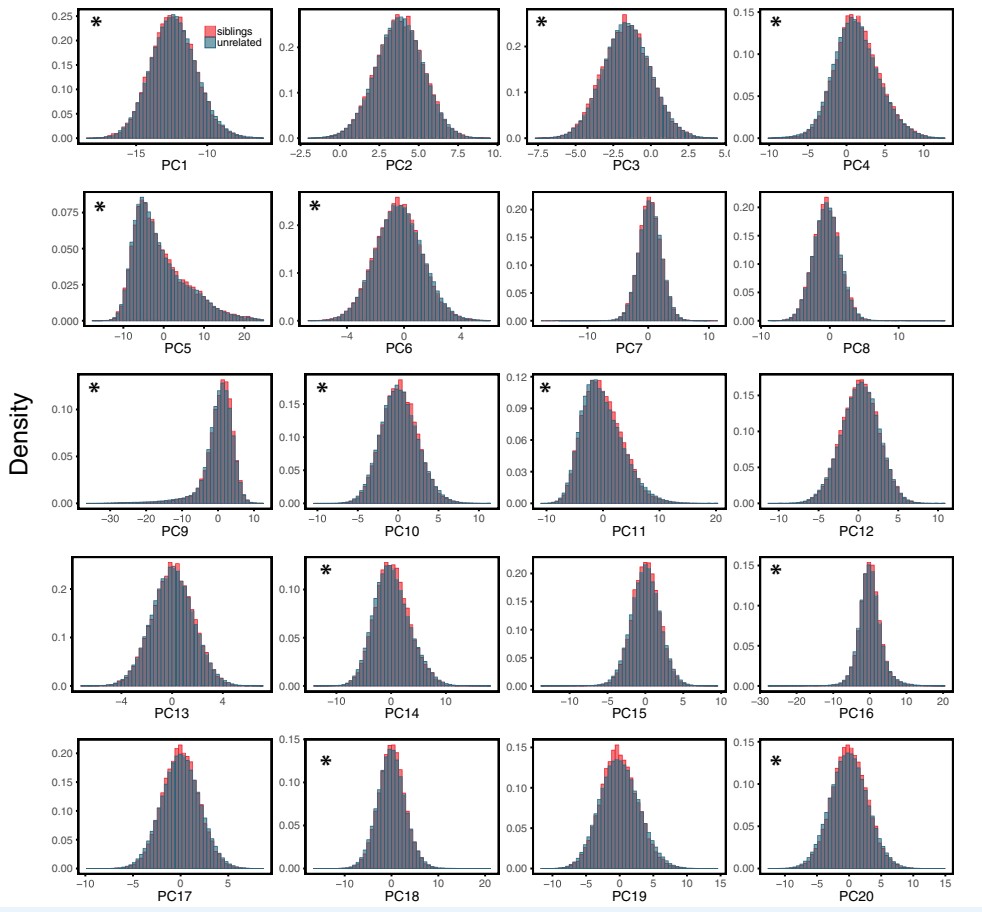

**Appendix 1—figure 7.** Comparison of siblings and unrelated individuals in the UK Biobank with respect to population structure. Panels show the distribution of PCs (principal components of the genotype data) for the siblings and unrelated sets used in the analysis described for *Figure 3* of the main text. For each sibling pair, one sibling was randomly selected for these comparisons. The asterisk symbol marks a significant difference at the 1% level between siblings and unrelated individuals, as assessed by a Mann-Whitney test. Despite slight but significant differences, siblings and unrelated sets are broadly similar with respect to their genetic ancestries.

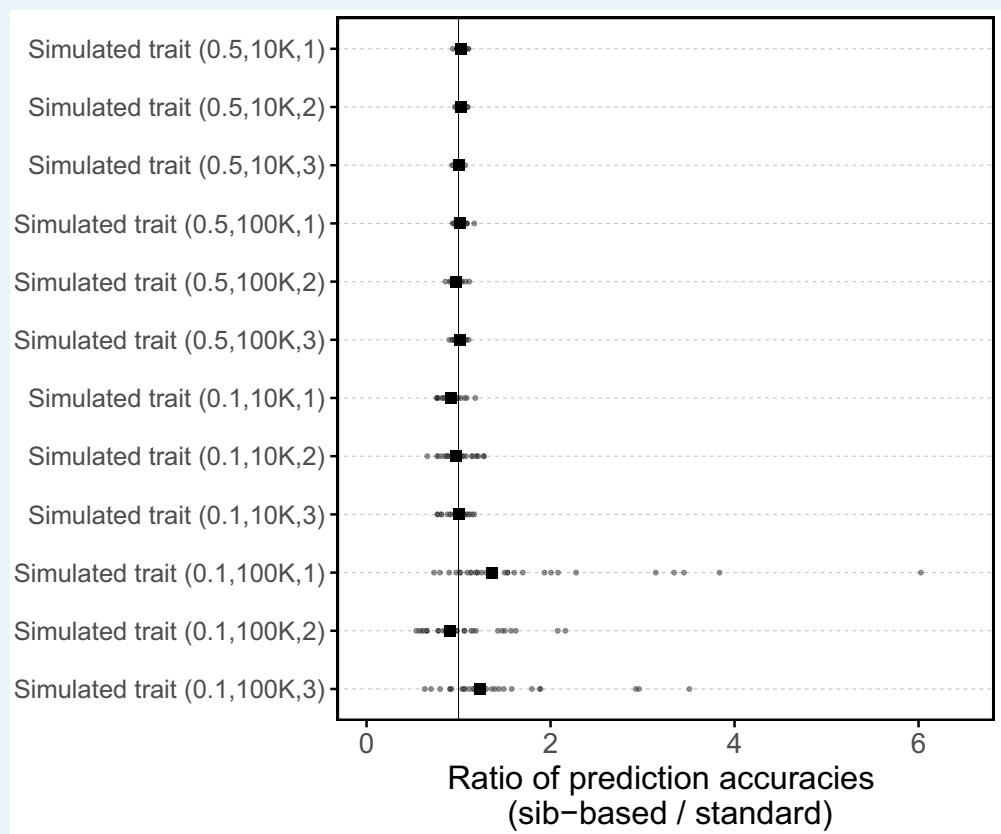

**Appendix 1—figure 8.** Comparison of prediction accuracies of polygenic scores based on standard and sib-GWAS for simulated traits. This figure mirrors *Figure 3B* of the main text, but here plotted for 12 simulated traits. The numbers in parentheses are the heritability, the number of causal loci considered, and the simulation replicate number, respectively. Three traits were simulated for each pair of heritability and number of causal loci parameters (see Materials and methods for simulation details). Small points show the ratio of the prediction accuracies in the two designs across 30 iterations; in each iteration, we resample sets of unrelated individuals to constitute three sets for discovery, estimation and prediction. Larger points show median values.

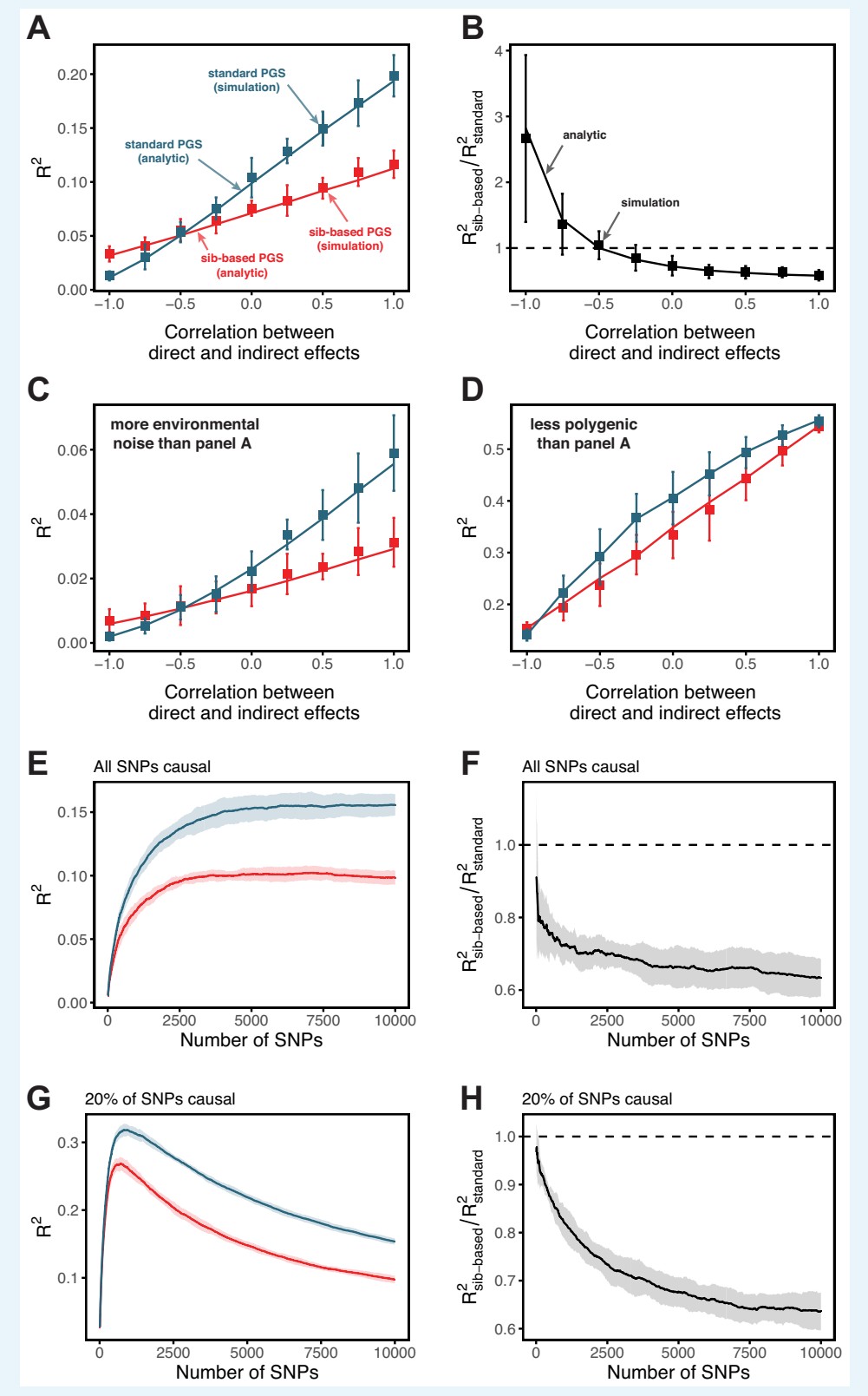

**Appendix 1—figure 9.** Simulation results for polygenic scores based on standard GWAS and sib-GWAS in the presence of indirect effects. (**A,B**) Simulation results as a function of the correlation between direct and indirect effects, $\rho$. Simulations were performed with $h_\beta^2 = 0.5$, $\tau_\eta^2 = 0.1$, and $\sigma_\beta^2/\sigma_\eta^2 = 5$. The size of the estimation set in the sib-GWAS is 10,000, and the size

of the estimation set in the standard GWAS is chosen to match sampling variances between the two study designs. The polygenic scores is based on 10,000 causal loci; its performance was evaluated in an independent set of 10,000 unrelated individuals. As long as direct and indirect effects are not strongly negatively correlated, the out of sample prediction accuracy is higher for the polygenic scores based on standard GWAS. (**C**) Same as (**A**) but with three-fold greater environmental noise. (**D**) Same as (**A**) but with 100 causal loci. In (**A–D**) points are mean ± 2 SD in 10 simulation iterations. Solid lines are values based on analytic expressions derived in Section 1.3.2. (**E–H**) Simulation results, with the same parameters as in (**A**) but $\rho = 0.5$, as a function of the number of SNPs included in the polygenic scores, with all loci being causal (**E,F**), or with 20% of loci being causal (**G,H**). SNPs are added in increasing order of their association p-value in an independent set of 20,000 unrelated individuals. In both cases, the ratio of prediction accuracies of polygenic scores based on sib- versus standard GWAS becomes smaller with the inclusion of more weakly associated SNPs, a behavior qualitatively similar to observations in *Figure 3* in the main text. Points are mean ± 2 SD in 10 simulations. See Section 1.3.3 for simulation details.

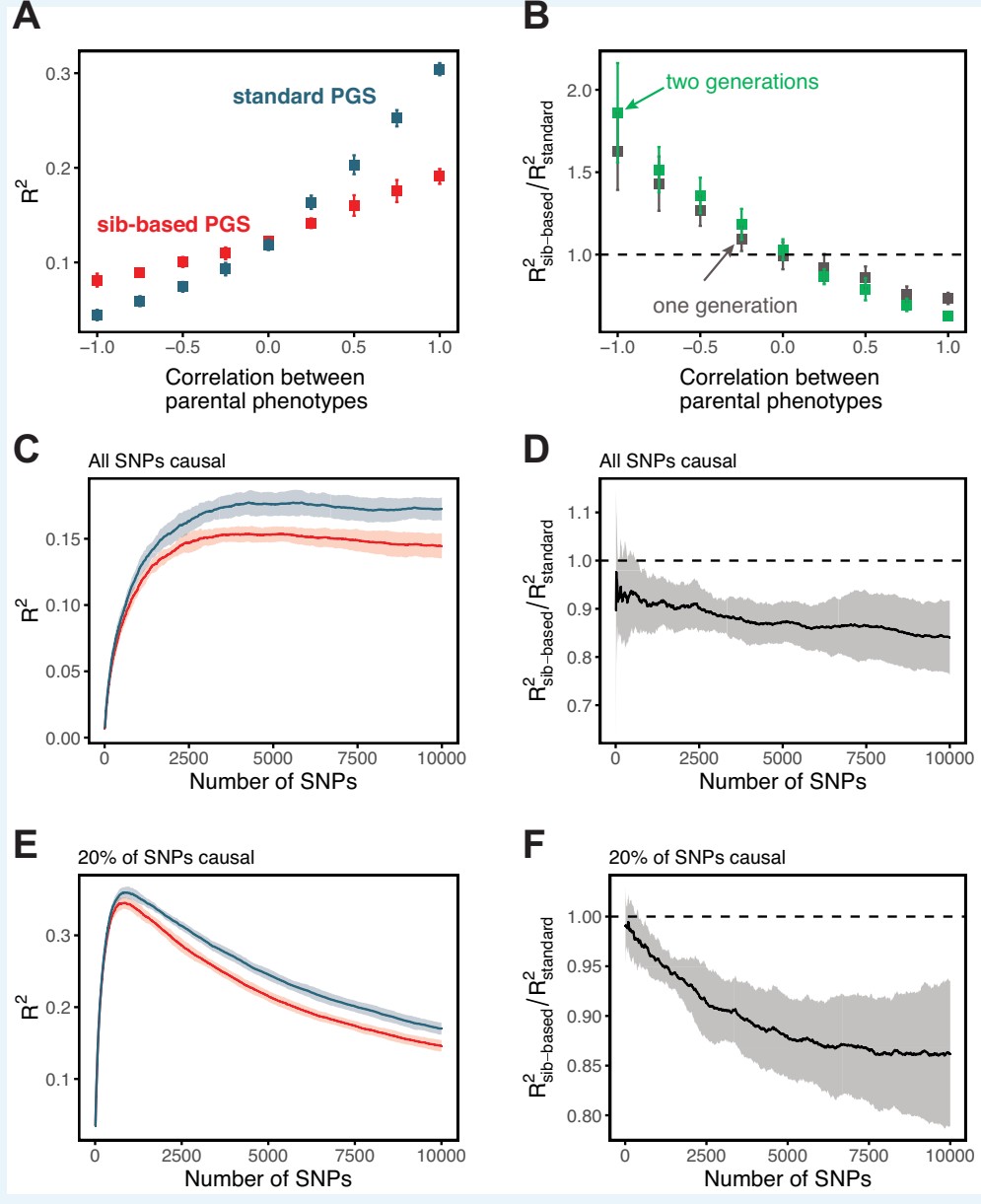

**Appendix 1—figure 10.** Simulation results for polygenic scores based on standard GWAS and

sib-GWAS in the presence of assortative mating. (**A**) Simulation results as a function of the approximate correlation between parental phenotypes, $\rho_a$. Simulations were performed with $h^2 = 0.5$ under random mating. The size of the estimation set in the sib-GWAS is 10,000, and the size of the estimation set in the standard GWAS is chosen to match sampling variances between the two study designs. The polygenic score is based on 10,000 causal loci; its performance was evaluated in an independent set of 10,000 unrelated individuals. Standard-GWAS based polygenic scores outperforms (underperforms) sib-GWAS based polygenic scores under positive (negative) assortative mating. (**B**) Ratio of prediction accuracies of the polygenic scores based on sib- versus standard GWAS, as a function of $\rho_a$, for two sets of simulations with one or two generations of assortative mating, with same parameters as in (**A**). (**C–F**) Simulation results, with the same parameters as in (**A**) but $\rho_a = 0.5$, as a function of the number of SNPs included in the polygenic score, with all loci being causal (**C,D**), or with 20% of loci being causal (**E,F**). SNPs are added in the order of their association p-value in an independent set of 20,000 unrelated individuals. In both cases, the ratio of prediction accuracies for scores based on sib-GWAS versus standard GWAS becomes smaller with the inclusion of more weakly associated SNPs, a behavior that is qualitatively similar to observations in **Figure 3** in the main text. Points are mean ± 2 SD in 10 simulation iterations. See Section 1.4.1 for simulation details.

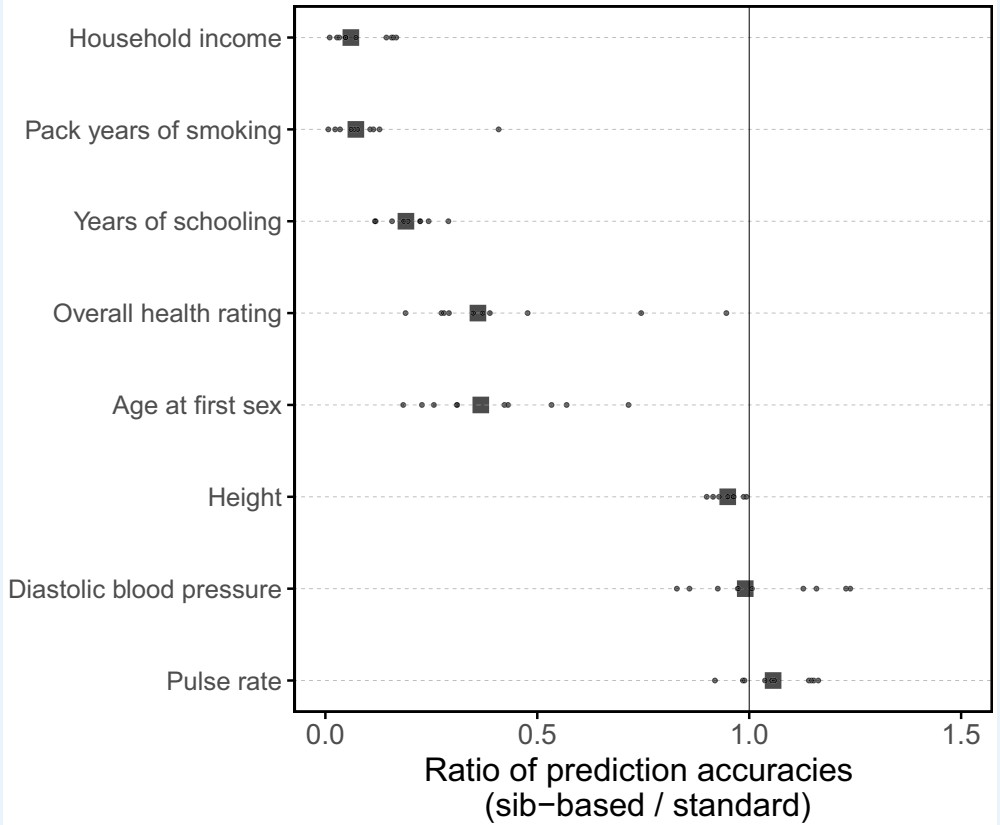

**Appendix 1—figure 11.** Comparison of prediction accuracies of polygenic scores based on standard and sib-GWAS matched for sex ratio. This figure mirrors **Figure 3B** of the main text, but here the samples of siblings and unrelated individuals used in the analysis are matched for their sex ratios. Results are shown for diastolic blood pressure, as the prediction accuracy differed between sexes (**Figure 1**); the related phenotype of pulse rate; and a subset of the traits for which the prediction accuracy varied by GWAS design (**Figure 3B**). Small points show the ratio of the prediction accuracies in the two designs across 10 iterations; in each iteration, we resample sets of unrelated individuals to constitute three sets for discovery, estimation and prediction. Larger points show median values. We note that pulse rate is now similarly predicted by the two GWAS approaches, suggesting that perhaps the slightly higher

prediction accuracy of the sib-GWAS shown in the main text *Figure 3B* are due to the sex ratio difference; for other traits, results are qualitatively unchanged.

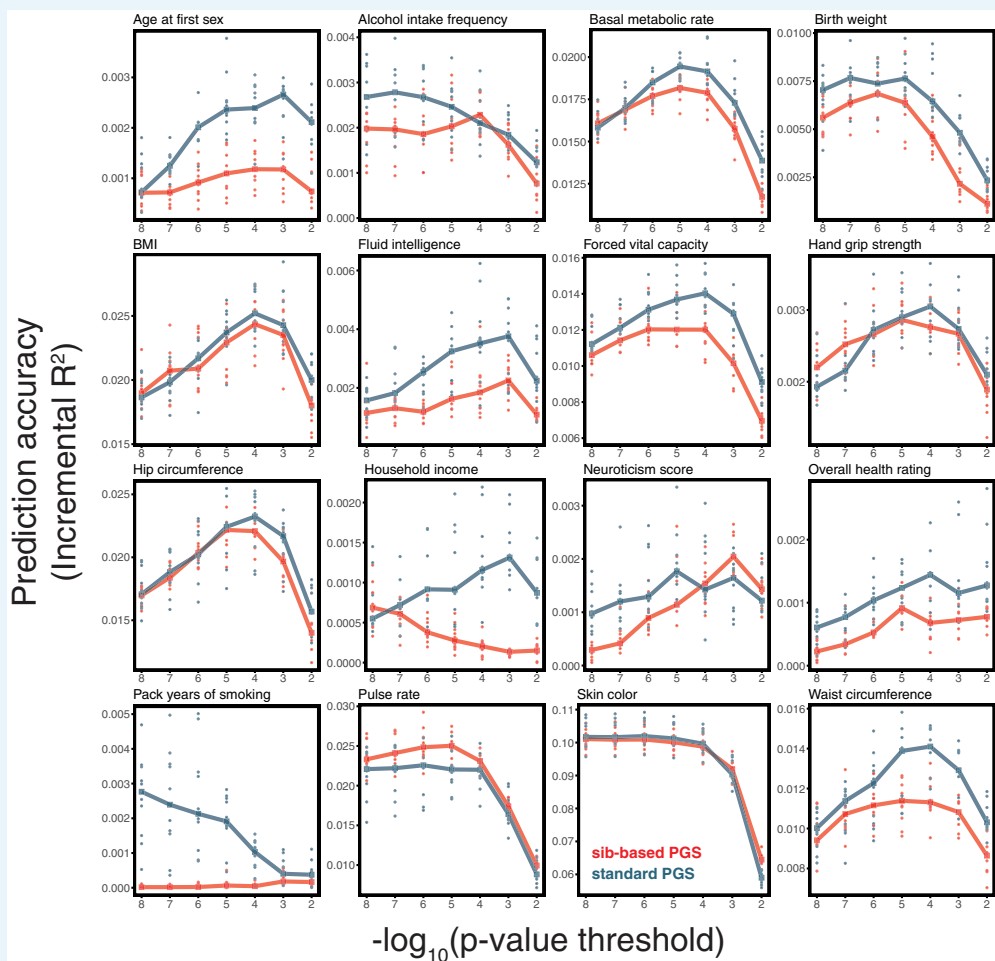

**Appendix 1—figure 12.** Prediction accuracy of polygenic scores based on sib-and standard GWAS, for a range of traits. This figure complements *Figure 3C–F* of the main text, showing the results of the study design depicted in *Figure 3A* for all traits presented in *Figure 3*. As described for *Figure 3*, we randomly divided unrelated individuals to constitute three non-overlapping sets for discovery, estimation and prediction. Small points correspond to 10 iterations of resmapling these three sets. The prediction accuracy is plotted as a function of the p-value threshold, where p-values come from the discovery GWAS. Lines show median values.

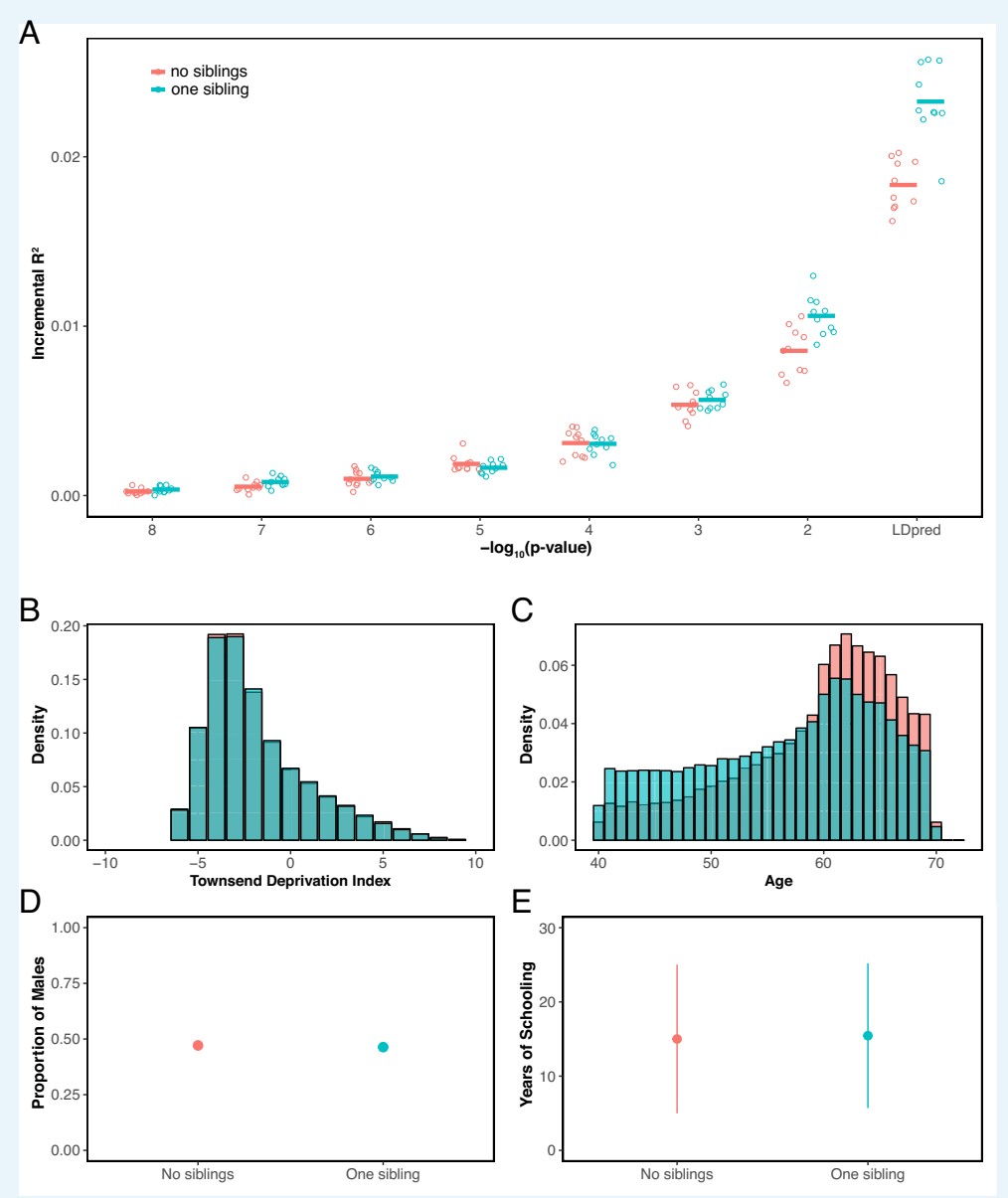

**Appendix 1—figure 13.** Prediction accuracy for years of schooling, for individuals with 0 or 1 full sibling. (**A**) The y-axis shows the prediction accuracy, measured as incremental $R^2$, in prediction sets stratified by participants' number of siblings, using a polygenic score for years of schooling based on a GWAS performed using individuals who reported to have exactly 1 sibling. The x-axis shows the p-value threshold for inclusion of a SNP in the polygenic score when based on a pruning and thresholding approach. Last points on the x-axis correspond to a polygenic score model based on the LDpred approach (*Vilhjálmsson et al., 2015*) with a prior probability of 1 on loci being causal. Points are values based on 10 iterations of resampling estimation and prediction sets. Thick horizontal lines denote the mean values. (**B**–**E**) Comparison of the distribution of Townsend deprivation index (**B**) the age distribution (**C**), the proportion of males (**D**), and mean years of schooling (± 2 SD) between individuals who reported having no sibling and those who reported having 1 sibling. The two sets have somewhat different distributions of ages (or possibly come from somewhat different birth cohorts), a feature that could contribute to the patterns seen in panel A, but are otherwise similar with respect to the other features considered.

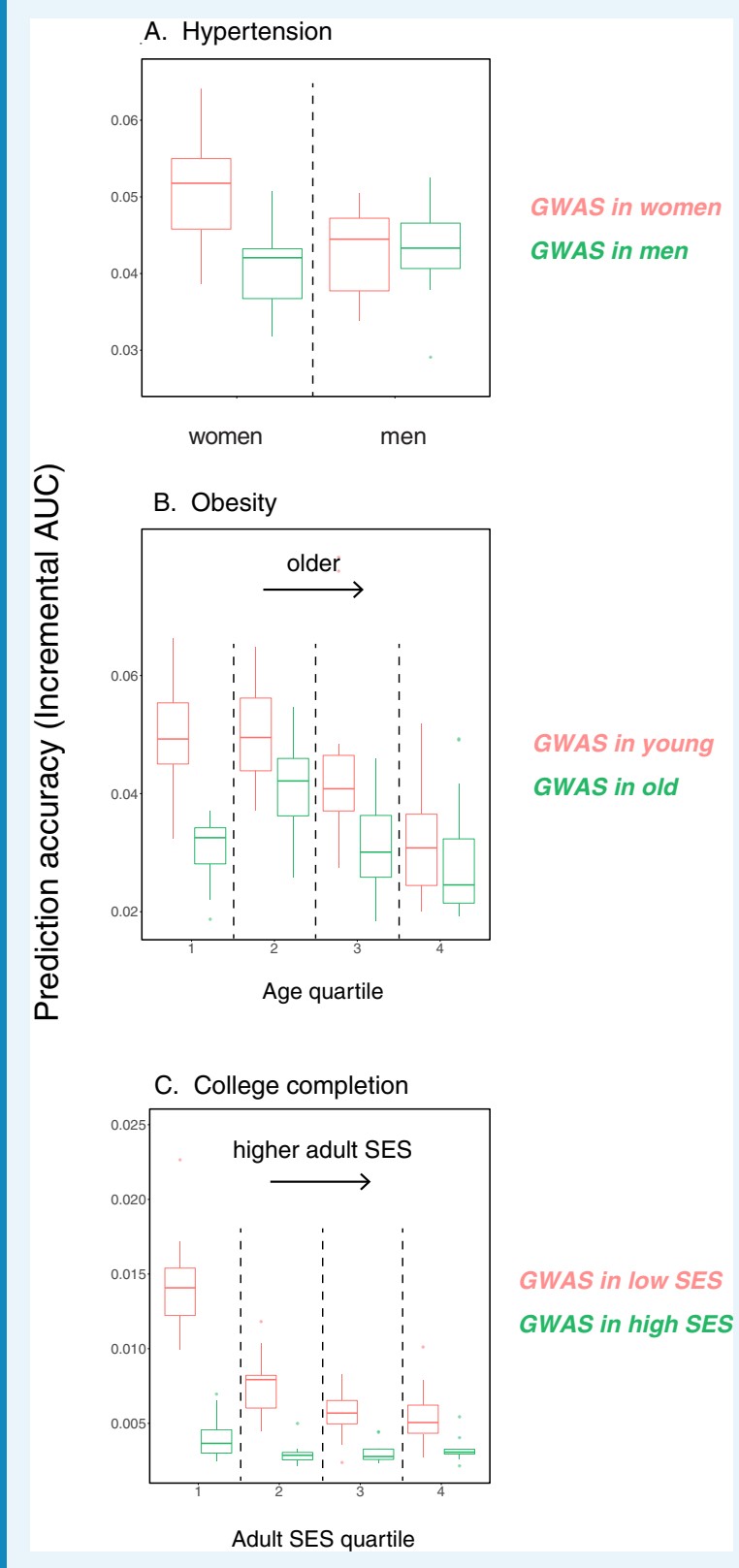

**Appendix 1—figure 14.** Variable prediction accuracy for binary traits, when measured as incremental AUC. This figure is analogous to the one shown in *Figure 1* of the main text, but considering dichotomized versions of the traits presented in *Figure 1* in the prediction sets, and with the y-axis showing incremental AUC values rather than incremental $R^2$. The polygenic

scores are based on GWAS using the quantitative trait values as in *Figure 1*. The traits are (**A**) diastolic blood pressure of over 110 mmHg, (**B**) BMI of over 35 Kg/m$^2$, and (**C**) completing a college or a university degree. Each box and whiskers plot was computed based on 20 iterations of resampling estimation and prediction sets. Thick horizontal lines denote the medians.

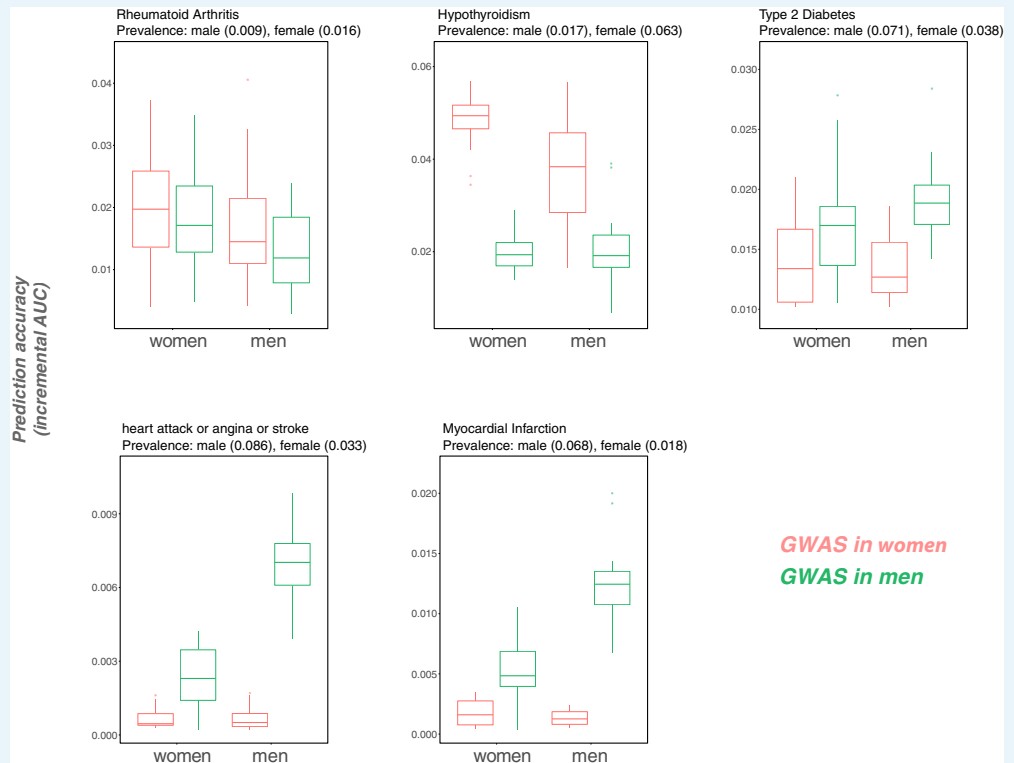

**Appendix 1—figure 15.** Variable prediction accuracy for binary disease phenotypes, measured as incremental AUC, in men versus women. This figure is analogous to the one shown in *Figure 1* of the main text, but looking at disease traits, and with the y-axis showing incremental AUC rather than incremental $R^2$. Each box and whiskers plot was computed based on 20 iterations of resampling estimation and prediction sets. Thick horizontal lines denote the medians. The variable prediction accuracy of PGS based on GWAS in men only versus women only could be driven in part by the differences in ratios of cases to controls (and hence by differences in the precision of the effect size estimates). However, we also observe that the prediction accuracy can vary depending on the sex composition of the prediction set (e.g., for cardiovascular outcomes), an observation that cannot be attributed to differences in case: control ratios of the GWAS.

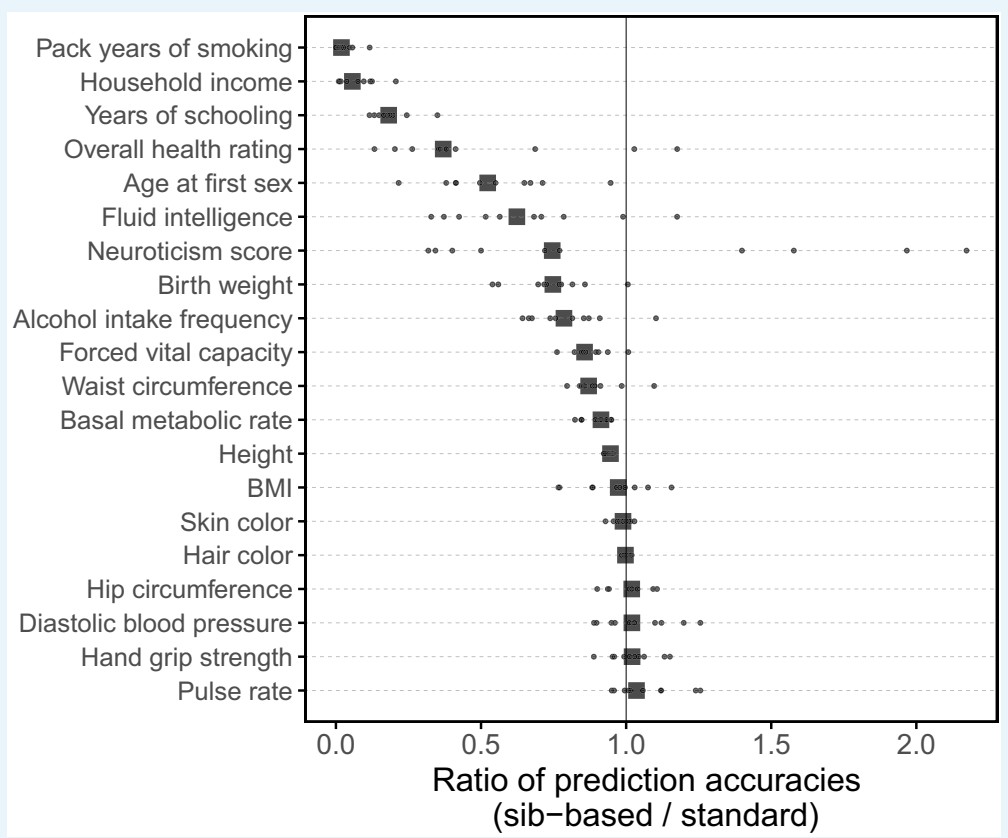

**Appendix 1—figure 16.** Comparison of prediction accuracies of polygenic scores (measured as $R^2$) based on standard and sib-GWAS. This figure mirrors **Figure 3B** of the main text, but here we first residualized the phenotypes on covariates, and then ran the same pipeline described as that used to generate **Figure 3B** on the residuals without further adjustment for covariates in the GWAS or prediction evaluation. Thus, this figure relates more directly to the analytical derivation in Section 1.2. However, the results in **Figure 3B** and here are qualitatively similar. Small points show the ratio of the prediction accuracies in the two designs across 10 iterations; in each iteration, we resample sets of unrelated individuals to constitute three sets for discovery, estimation and prediction. Larger points show median values.

**Appendix 1—table 1.** UK Biobank phenotype data used in this study and their corresponding data fields. In parentheses are the units of measurements.

| Trait | Description | UKB data field |
|---|---|---|
| Age | Age when attended assess-ment center (years) | 21003 |
| Age at first sex | Self-reported age at first sex-ual intercourse (years) | 2139 |
| Alcohol intake frequency | Self-reported category, en-coded as an integer: 1, 'Daily or almost daily'; 2, 'Three or four times a week'; 3, 'Once or twice a week'; 4, 'One to three times a month'; 5, 'Special occasions only'; 6, 'Never' | 1558 |
| Basal metabolic rate | Estimated from body compo-sition impedance measure-ments (KJ) | 23105 |

*Appendix 1—table 1 continued on next page*

*Appendix 1—table 1 continued*

| Trait | Description | UKB data field |
|---|---|---|
| Birth weight | Self-reported birth weight (Kg) | 20022 |
| Body mass index | Constructed from height and weight measurements (Kg/m$^2$) | 21001 |
| Diastolic blood pressure | Measured using automated devices (mmHg); values are adjusted for medicine use (see Materials and methods) | 4079, 6153, 6177 |
| Fluid intelligence | Unweighted sum of the number of correct answers given to 13 fluid intelligence questions | 20016 |
| Forced vital capacity | Calculated from breath spirometry (liters) | 3062 |
| Hair color | Self-reported category, encoded as an integer: 1, 'Blonde'; 2, 'Red'; 3, 'Light brown'; 4, 'Dark brown'; 5, 'Black'; none, 'Other' | 1747 |
| Hand grip strength | Measured right and left hand isometric grip strength (Kg) | 46, 47 |
| Height | Measured standing height (cm) | 50 |
| Hip circumference | Measured hip circumference (cm) | 49 |
| Hospital inpatient diagnoses | Diagnoses made during hospital inpatient admissions, coded according to the International Classification of Diseases (ICD-9 and ICD-10) | 41202, 41203, 41204, 41205, 41270, 41271 |
| Household income | Self-reported average total annual household income before tax category, encoded as an integer: 1, 'Less than £18,000'; 2, '£18,000 to £30,999'; 3, '£31,000 to £51,999'; 4, '£52,000 to £100,000'; 5, 'Greater than £100,000' | 738 |
| Myocardial infarction outcomes | Algorithmically-defined myocardial infarction outcomes obtained through combinations of UK Biobank's assessment data collection (e.g., self-reported conditions), and data from hospital admissions | 42000, 42001 |
| Neuroticism score | Derived summary score, based on participants' responses to 12 neurotic behaviour-related questions | 20127 |
| Number of full siblings | Sum of self-reported number of full brothers and full sisters | 1873, 1883 |
| Overall health rating | Self-reported category, encoded as an integer: 1, 'Excellent'; 2, 'Good'; 3, 'Fair'; 4, 'Poor' | 2178 |

*Appendix 1—table 1 continued on next page*

*Appendix 1—table 1 continued*

| Trait | Description | UKB data field |
|---|---|---|
| Pack years of smoking | Calculated for individuals who have ever smoked as the number of cigarettes smoked per day, divided by twenty, multiplied by the number of years of smoking (years) | 20161 |
| Pulse rate | Measured during the automated blood pressure readings (bpm) | 102 |
| Qualifications | Self-reported educational or professional qualifications, selected from: 'College or University degree', 'NVQ or HND or HNC or equivalent', 'Other professional qualifications eg: nursing, teaching', 'A levels/AS levels or equivalent', 'O levels/GCSEs or equivalent', 'CSEs or equivalent', or 'None of the above' | 6138 |
| Sex | Self-reported sex and as determined from genotyping analysis | 31, 22001 |
| Skin color | Self-reported category, encoded as an integer: 1, 'Very fair'; 2, 'Fair'; 3, 'Light olive'; 4, 'Dark olive'; 5, 'Brown'; 6, 'Black' | 1717 |
| Townsend deprivation index | Townsend deprivation index at recruitment | 189 |
| Vascular/heart problems | Self-reported vascular/heart problems diagnosed by doctor selected from the categories: 'Heart attack', 'Angina', 'Stroke', 'High blood pressure', and 'None of the above' | 6150 |
| Waist circumference | Measured waist circumference (cm) | 48 |

**Appendix 1—table 2.** Genetic correlations across samples that vary by a study characteristic. Numbers are genetic correlations estimated using LD score regression for BMI, years of schooling and diastolic blood pressure, across samples stratified by age, Townsend deprivation index (a measure of socioeconomic status, SES), and sex, respectively. 'Q' denotes quartile of age or SES.

| Trait/characteristic | Pair of strata | Genetic correlation (s.e.) |
|---|---|---|
| BMI/Age | (Q1,Q2) | 0.93 (0.036) |
| | (Q1,Q3) | 0.95 (0.035) |
| | (Q1,Q4) | 0.95 (0.038) |
| | (Q2,Q3) | 0.89 (0.032) |
| | (Q2,Q4) | 0.91 (0.036) |
| | (Q3,Q4) | 1.00 (0.040) |

*Appendix 1—table 2 continued on next page*

*Appendix 1—table 2 continued*

| Trait/characteristic | Pair of strata | Genetic correlation (s.e.) |
|---|---|---|
| Years of schooling/SES | (Q1,Q2) | 0.98 (0.054) |
|  | (Q1,Q3) | 1.00 (0.067) |
|  | (Q1,Q4) | 0.93 (0.068) |
|  | (Q2,Q3) | 0.97 (0.064) |
|  | (Q2,Q4) | 1.09 (0.074) |
|  | (Q3,Q4) | 1.04 (0.074) |
| Diastolic blood pressure/Sex | (male,female) | 0.93 (0.031) |

**Appendix 1—table 3.** Sample sizes used for siblings and unrelated sets. As described in *Figure 3A*, for the comparison of prediction accuracies of polygenic scores based on standard and sib-GWAS, we first ascertain SNPs in a large sample of unrelated individuals ('Unrelated-discovery') and then estimate the effect of these SNPs with a standard regression using unrelated individuals ('Unrelated-n*') and, independently, using sib-regression (in the 'Siblings' set). Finally, we used the polygenic scores for prediction in a third sample of unrelated individuals ('Unrelated-prediction'). This table shows sample sizes used for each set across the traits analyzed. For simulated traits, the numbers in parentheses are heritability, number of causal loci, and simulation replicate, respectively (three traits were simulated for each pair of heritability and number of causal loci parameters, see Materials and methods for simulation details).

| Trait | Siblings (pairs) | Unrelated-n* | Unrelated-discovery | Unrelated-prediction |
|---|---|---|---|---|
| Age at first sex | 13675 | 8746 | 244988 | 27220 |
| Alcohol intake frequency | 17282 | 10923 | 276885 | 30764 |
| Basal metabolic rate | 16802 | 13467 | 269750 | 29972 |
| Birth weight | 6750 | 5766 | 159074 | 17674 |
| BMI | 17217 | 12359 | 274868 | 30540 |
| Diastolic blood pressure | 14791 | 9514 | 253227 | 28136 |
| Fluid intelligence | 3889 | 2979 | 101016 | 11223 |
| Forced vital capacity | 14605 | 10009 | 252576 | 28064 |
| Hair color | 16859 | 11763 | 272209 | 30245 |
| Hand grip strength | 17070 | 10832 | 275117 | 30568 |
| Height | 17242 | 18147 | 269973 | 29997 |
| Hip circumference | 17254 | 11648 | 275930 | 30658 |
| Household income | 13240 | 8704 | 239326 | 26591 |
| Neuroticism score | 11756 | 6909 | 227010 | 25223 |
| Overall health rating | 17189 | 10378 | 276581 | 30731 |
| Pack years of smoking | 2307 | 1682 | 85544 | 9504 |
| Pulse rate | 14791 | 8812 | 253859 | 28206 |
| Skin color | 16903 | 10334 | 274159 | 30462 |
| Waist circumference | 17257 | 11749 | 275873 | 30652 |
| Years of schooling | 17037 | 11885 | 273553 | 30394 |
| Simulated trait (0.5,10K,1) | 17299 | 11685 | 276404 | 30711 |
| Simulated trait (0.5,10K,2) | 17299 | 11505 | 276566 | 30729 |
| Simulated trait (0.5,10K,3) | 17299 | 11422 | 276641 | 30737 |
| Simulated trait (0.5,100K,1) | 17299 | 11814 | 276288 | 30698 |

*Appendix 1—table 3 continued on next page*

*Appendix 1—table 3 continued*

| Trait | Siblings (pairs) | Unrelated-n* | Unrelated-discovery | Unrelated-prediction |
|---|---|---|---|---|
| Simulated trait (0.5,100K,2) | 17299 | 11833 | 276271 | 30696 |
| Simulated trait (0.5,100K,3) | 17299 | 11490 | 276579 | 30731 |
| Simulated trait (0.1,10K,1) | 17299 | 9072 | 278756 | 30972 |
| Simulated trait (0.1,10K,2) | 17299 | 9158 | 278678 | 30964 |
| Simulated trait (0.1,10K,3) | 17299 | 9111 | 278721 | 30968 |
| Simulated trait (0.1,100K,1) | 17299 | 9133 | 278701 | 30966 |
| Simulated trait (0.1,100K,2) | 17299 | 9069 | 278758 | 30973 |
| Simulated trait (0.1,100K,3) | 17299 | 9108 | 278723 | 30969 |

**Appendix 1—table 4.** Qualifications to years of schooling conversion table. Educational or professional qualifications were converted to years of schooling following *Okbay et al. (2016)*.

| Qualifications (UKB data field 6138) | Years of schooling |
|---|---|
| College or University degree | 20 |
| NVQ or HND or HNC or equivalent | 19 |
| Other professional qualifications eg: nursing, teaching | 15 |
| A levels/AS levels or equivalent | 13 |
| O levels/GCSEs or equivalent | 10 |
| CSEs or equivalent | 10 |
| None of the above | 7 |

