## [Decision Letter]

**Acceptance summary:**

Polygenic scores are being generated for an increasingly large number of traits, and are beginning to be used outside of a narrow research context. However many questions remain about how and when it is appropriate to use them. One oft-discussed concern is whether polygenic scores are "portable" between ancestry groups. But this study demonstrates that polygenic scores are often not even portable within ancestry groups when stratified on the basis of common demographic parameters. This and other results discussed in the paper raise important questions about the use of polygenic scores, and should be read and considered carefully by anyone designing, carrying out or applying the results of large-scale human genetic analyses.

**Decision letter after peer review:**

Thank you for submitting your article "Variable prediction accuracy of polygenic scores within an ancestry group" for consideration by *eLife*. Your article has been reviewed by three peer reviewers, one of whom is a member of our Board of Reviewing Editors, and the evaluation has been overseen by Mark McCarthy as the Senior Editor. The following individual involved in review of your submission has agreed to reveal their identity: Paul O'Reilly (Reviewer #3).

The reviewers have discussed the reviews with one another and the Reviewing Editor has drafted this decision to help you prepare a revised submission.

The manuscript in its current form would not, even with minor revision, be suitable for the journal. However, we would consider an extensively revised manuscript that addresses the following key concerns of the reviewers.

1) The paper will benefit from a clearer focus; while the authors aim was to simply establish the variable of portability between strata, they went quite further from this with the topics of assortative matting, sex balance, parents' effects, etc. Just achieving their main aim would make this paper very worthy as it points to issues that those already thinking in applying PRS in clinical practice right now need to urgently appreciate.

2) In focusing on the main aim, a deeper exploration of the causes that drive the lack of portability would increases the informativeness of the study. It was pointed out that the difference in portability across strata might be due to differences in power, sample size, heritability, phenotypic variance, phenotype distribution, influence of environment, insights in how these and potentially others affect portability would be helpful. This should also include a discussion on how is it that different strata can generate even higher predictive power than using the same stratum (in base and target) and why the PRS based on the full sample size, in the provided examples, always has a better "predictive ability" (highest R^2^), than PRS based on any of the strata.

3) Analysis of disease outcomes, here e.g. (extreme) obesity and hypertension, will be useful to translate findings to more clinical settings, including calculation of ROC-AUC, a standard easy-to-interpret predictive statistic.

Reviewer #1:

In this study, the authors examine the portability of polygenic scores within a homogeneous, generally healthy population of white British unrelated adults, living across the UK. They show that the PGS performance is influenced by age, sex, socio-economic status and study designs. The reduced accuracy of PGS is not (only) caused by environmental influences, the authors show that also the magnitude of the genetic effects among groups, and indirect effects of assortative mating affect the performance of the PGS.

While these findings raise important consideration with reference to the use of PGSs, it would have been interesting if they had gone a little deeper with their analyses; e.g. to also identify the features (distribution, prevalence of disease, effects of environment) of outcomes and covariates that impact the prediction performance the most.

It seems that for all traits, even though there are differences in R^2^ for sex-specific, age-specific and SES-specific PGSs, the highest R^2^ (still low) is achieved when the PGS is generated in the full population. Thus, isn't the conclusion then simply that stratification lowers the PGS performance and that a (blunt) PGS, based on the most comprehensive possible population performs best?

The authors only considered continuous outcomes for "prediction"; it would be informative to also include dichotomous/disease outcomes (related to the continuous risk factors), such as (extreme) obesity, hypertension, etc. That would also allow calculation prediction statistics such as AUC-ROC (sensitivity, specificity, PPV, NPV), which are easier to interpret in the context of clinical relevance.

The PGS was generated based on LD-based clumping, but it seems only one threshold was used; r^2^<0.1. From experience, it seems that higher LD threshold results in better performance of the PGS. Thus, it might be worth testing higher LD threshold for better performing PGSs.

Reviewer #2:

The manuscript of Mostafavi discusses a timely and important question: how accurate are polygenic risk scores (PRS) derived from GWAS studies and first validated in large, but single cohorts, when predicting risk in populations different to those used in their derivation. Recently GWAS studies have achieved sample sizes of a million subjects. In addition, the UK biobank has released rich datasets close to half a million people genotyped, extensively phenotyped, and followed so far for about 10 years. These resources have spurred the development of PRS for many traits ranging from complex disease onset, quantitative phenotypes, and even to sociological predictions. Furthermore, a number of direct to consumer companies have started to provide their customers PRS for medical traits. Therefore, the urgent question is how reliable are these scores and how well they port to other populations and people around the world.

Mostafavi work starts with the recent publications suggesting that PRS port poorly to populations other than European descent (the population object of most of the data available), in particular to African populations. There have been recent calls to expand GWAS and biobank style cohorts to include people from other ancestries around the world. However, Mostafavi analyzes now show that problems with portability are not just evident in different ancestries, but also to different strata from samples from European decent participants. If this is correct, this raises the urgency to hold off implementations of these scores in to healthcare settings until we truly understand how and when these scores can be applied across groups of people.

The authors provide a meticulous dissection of the situations when the PRS are not portable within people of the same Ancestry. They perform careful QC of the UK Biobank data, use reasonable methods to adjust the sample strata, perform the GWAS, and construct and test the PRS. The authors state that their main goal was to demonstrate how differences in the composition of the GWAS cohorts affect portability for different traits of interest within the same ancestry, and they achieve this. However, they do a good work exploring the possible factors involved in the reduced portability and leave us with important lessons on how our assumptions about GWAS data may be wrong at times. Their data and simulations add important information on how sex ratios, indirect effects, GxE, assortative mating and other factors confound the data and contribute to poor performance in subsets of the UK biobank subjects. The manuscript is well written and extensive details is provided on the methods and simulations. The main text is accompanied by extensive supplemental results of importance, which are referred at appropriate places in the text.

I have a number of questions that I would like the authors consider and when appropriate clarify in their final manuscript:

1) Throughout the manuscript the term "portability" is used and metrics that measure the performance of the PRS are used to judge whether portability is feasible or reduced. While this term has been used in prior literature w.r.t to the reduced ability of a PRS developed in cohorts of European descent people when applied to populations of different ancestry, this term is jargon. In statistics and machine learning the term that describes this poor performance is overfitting and the ability of a model, or in this case a PRS, to preform equally well in other samples, is termed generalization. Would it not be "overfitting" the term that more accurately describe the lack of "portability"?

2) The author's use R^2^ to measure the reduced generalization of the PRS to the different cohorts' strata, and they concede that is not clear what would be the best metric to use for this analysis. However, many publications use the AUC as a metric to decide the best model and threshold of SNPs to pick and to describe how the PRS augments other predictor bases on covariates or other factors. It is not intuitive how R^2^ correlates with AUC, and therefore it would be best if the authors try in at least one analysis to compare R^2^ to AUC.

3) Recent publications have used the LDPred method that uses as input summary statistics rather than genotypes to develop PRS based on UK Biobank data. Some of these have shown generalization to others cohorst. Can the author's comment whether LDpred features distinguished it from the methods used in their analysis conferring an advantage? PRS derived from LDpred also tend to include many more SNPs to achieve a greater AUC. The author's explore P-value cut-offs (and hence number of SNPS) in some analysis, but it was not clear whether this suggested more SNPs increases overfitting. Do the results from the author's suggest that including more SNPs increases overfitting?

Reviewer #3:

The generalisability of polygenic scores is an important issue given the possibility of them being incorporated in to healthcare soon. This paper makes excellent use of the UK Biobank resource to present two sets of interesting results relating to the predictive power of PRS. However, I'm not sure if the way in which the primary results are framed in relation to portability is quite right.

The first set of results appear to highlight how GWAS are better powered in certain subgroups of the population (eg. GWAS of DBP are better powered in women, BMI in those in middle age, years of schooling in lower SES groups). Years of schooling is better predicted in SES 4 by GWAS on SES 1 than by SES 4 itself – this shows hyper-portability, not limited portability as discussed throughout the article. This is also true for BMI and DBP (to a lesser extent DBP). The bivariate LD Score results showing genetic correlations close to 1 seems to support this being principally a power (rather than portability) issue, in which there are different relative contributions of genetics and the environment across the groups, making some better-powered to capture – largely the same – genetics. Based on the results (inc. those of Figure 2), I'd have thought that this is most likely due to the environment resulting in convergence and genetics resulting in divergence (at least in the case of BMI and education) – eg. BMI converges to low values in old age, irrespective of genetic predisposition (relatively), while education levels of those in the top SES are forced to be similar due to the qualification requirements of professional jobs. Thus, the highest phenotypic variance, heritability, and thus PRS predictive power for BMI is from the lower age group and schooling from the lowest SES. How specific this pattern of results is to these particular traits is unclear, and certainly there seems to be a somewhat different cause for the DBP results. However, in general I see these results as being analogous to eg. how GWAS on Alzheimer's is perhaps being best-powered to detect common genetic variants when applied to individuals in their 70s, GWAS on blood pressure best-powered when applied to individuals in their 30s, and lung cancer GWAS being optimised when applied to non-smokers.

While the downstream impact of variable GWAS power in different sub-groups on PRS prediction is extremely valuable to see here, I think the implications of these results may be missed unless discussed in the context of differential GWAS power in different sub-groups – eg. in relation to ancestry, the consensus is that portability will be optimised by expanding GWAS sampling to non-European populations, but the analogy is not true here – the consequence of these results (at least for these traits) is that portability will be optimised if GWAS sampling is restricted to those sub-groups for which heritability is highest (since, otherwise, the portability in fact seems high).

The second set of results expand on similar analyses by Kong et al., 2018 and Selzam et al., 2019, showing that the indirect effects of familial genetics, and potentially assortative mating, could contribute to standard PRS predictive power. Any impact on portability is not explicitly demonstrated here, but the point is made that differences in culture (family structures etc) could lead to significant differential portability given the strong influence of indirect effects on standard PRS for some traits. These are great results to see, and will add to the literature on the topic of indirect genetic effects nicely.

[Editors' note: further revisions were suggested prior to acceptance, as described below.]

Thank you for resubmitting your work entitled "Variable prediction accuracy of polygenic scores within an ancestry group" for further consideration by *eLife*. Your revised article has been evaluated by Mark McCarthy (Senior Editor) and a Reviewing Editor.

The manuscript has been improved and reviewers were generally satisfied with the changes made. However, while discussing the revisions some remaining issues were noted that need to be addressed before acceptance, as outlined below:

1) As sample size is an important determinant of the accuracy of the association effect sizes and thus also of the prediction accuracy, the sample sizes of each stratum needs to be clearly stated. Currently, the use of "unstratified" and "all" is misleading as it would suggest it include the data of strata combined. However, this seems not the case; the "unstratified" or "all" have now been ("artificially") reduced to match the sample size of the strata, which is not intuitive. After all, when data for multiple strata is available, one would be better off combining all data, rather than using the strata. We suggest to clearly report sample sizes, and to choose a more precise terminology to describe the "unstratified" or "all" group. Including an "all" group, that combined strata, as before, would seem informative to illustrate that sample size trumps population-specificity.

2) One of the reviewers noted difference between the original Figure 1 and the new Figure 1, and wondered whether the data changed (slightly) compared to their previous version because the results are slightly different (easiest to see by comparing median/boxes of young/old in BMI plot between original and new versions). It would be worth checking that this is just updated UKB data rather than an error. Furthermore, the R^2^ for DBP in the "mixed GWAS" is just less than half than for the "total GWAS" (again comparing original Vs new figures), while for education there's a 10-fold difference (ie. doubling GWAS N has led to 10x greater R^2^).. This can be the case ('inflection points' and all), but it would be worth checking that there's no error here because it seems a little surprising.

Reviewer #1:

The authors have addressed all concerns and the revised version is more focused and conveys a clearer message.

While they now present the "all" GWAS for the same sample size as for the stratified GWAS, this has not been presented in a very clear way. In the text, L179 they state they use the same sample size, but it would be good to report what this sample size is. Intuitively, one would not expect that an "all" GWAS had the same sample size as a stratified GWAS.

I suggest to clarify this in the legends of the figures, and also more clearly in the text.

Reviewer #2:

Motafavi et al. have updated their manuscript based on reviewer's feedback and answered all the questions and comments posed during the review process of the first submission.

In my opinion their rebuttal and responses are satisfactory and their new and improved manuscript reflect their responses to reviewers. The author's maintained the focus of their manuscript in demonstrating that within ancestry there are a number of possible confounders and study design considerations that impact the prediction accuracy of PGS and put in question its immediate implementation in the clinic as some propose. In addition, and without going overboard expanding hugely the manuscript to explore all possible avenues to explain these effects, their perform a number of analyses that reject the simple idea than environment variance among strata is solely responsible for these effects, but in fact more complex phenomena are at play, including indirect effects, assortative mating, etc. These studies would be the starting point for future studies by the authors and others in the field. I thank the authors to indulge us by adding several interesting analyses suggested by reviewers, such as the inclusion of results by LDpred, a method gaining traction, and showing that as more markers are included in the PGS portability may be more difficult. The comprehensiveness of the supplementary material and clarity of explanations are superb.

After reading the new material in its entirety (apologies that took some time for me to go over this again), I find it a significant contribution to the field, and I do not have any other significant comments to offer recommending its acceptance for publication.

Reviewer #3:

I am satisfied with the revisions that the authors have made and have no further comments to make on the manuscript. Nice work!

[Editors' note: further revisions were suggested prior to acceptance, as described below.]

Thank you for submitting your article "Variable prediction accuracy of polygenic scores within an ancestry group" for consideration by *eLife*. Your article has been discussed by Ruth Loos, the Reviewing Editor, and Mark McCarthy as the Senior Editor.

While you have been very responsive to the reviewers comments and concerns, we remain concerned that some of the presentation of findings remains counter-intuitive and could lead to mistaken inference. We would like you to consider one final set of revise your paper to include the following:

1) Include strata-combined analyses; i.e. a "full" analysis that combines the stratum-specific sample sizes into one large population. The current study shows that the stratum-specific PRSs perform better than strata-combined PRS, but only if the strata-combined (e.g. men+women) PRS is based on GWAS of the same size as the individual strata (e.g. men, women).

In the very first version of the paper, it was clear that the strata-combined PRS (based on combined stratum-specific data) performs much better than any of the stratum-specific PRS: it seems that the larger sample size overcomes any advantage of stratum-specific analyses (in these examples at least). In response to the reviewers' comments, the authors "reduced" the sample size of the strata-combined analyses to the size of that of the individual strata. However, reducing a strata-combined analyses to the same sample size as the stratum-specific sample sizes is not an intuitive comparison (i.e. a combined analyses indicated that samples are merged into a bigger sample), as stratum-specific implies that these are a subset of the strata-combined. The current version does not address this issue: the labeling of the (sub)groups has been changed, but that has the potential to mislead the reader to infer that stratum-specific analyses will provide the best prediction, whereas a strong predictor is also sample size.

2) Discuss the role of sample size in prediction in the Discussion section;

i.e. put the impact of stratum-specificity in prediction in comparison to the impact of sample size in the (this relates to the comment above). It is important that readers understand that both contribute to prediction, and that possibly sample size is even more important than stratum-specificity. It would be good if the authors can speculate on any circumstances in which the strata-combined analysis would not provide better overall prediction than the stratum-specific subsets.

[Editors' note: further revisions were suggested prior to acceptance, as described below.]

Thank you for resubmitting your work entitled "Variable prediction accuracy of polygenic scores within an ancestry group" for further consideration by *eLife*. Your revised article has been evaluated by Mark McCarthy (Senior Editor) and a Reviewing Editor.

The manuscript has been improved but there are some remaining issues that need to be addressed before we move to final acceptance. These relate to the discussion about the trade-off between sample size and stratum-specific effects on PRS performance that have been the subject of the recent revision requests. Thank you for adding additional information and figures that address this issue. However, as per the last round of revision requests, we do not feel that this question has been adequately addressed in the Discussion, and that some explicit text on this matter forms an essential part of the paper. As you will have seen from the to and fro over the revisions, this is an issue that has exercised several of the editors and reviewers, and I can’t imagine that it will be any less important to the readers of the paper. Put simply, the question is this: based on the data you have generated, are there any real world situations where researchers would be better off splitting a data set into its component strata (eg by gender) and proceeding on the basis of stratum specific risk scores, rather than using the whole data set? The data in the additional figure, suggest not (at least in the models and situations you tested), which is why this is important to place into context.

---

## [Author Response]

The manuscript in its current form would not, even with minor revision, be suitable for the journal. However, we would consider an extensively revised manuscript that addresses the following key concerns of the reviewers.1) The paper will benefit from a clearer focus; while the authors aim was to simply establish the variable of portability between strata, they went quite further from this with the topics of assortative matting, sex balance, parents' effects, etc. Just achieving their main aim would make this paper very worthy as it points to issues that those already thinking in applying PRS in clinical practice right now need to urgently appreciate.

We agree that establishing variable prediction accuracy--in clinical settings as well as in other areas in which polygenic scores are increasingly deployed--is the most important result in the paper. Assortative mating, indirect effects, GxE and stratification are examined only through the prism of portability. We have tried to emphasize our main focus in the revisions. In particular, to make the relevance to human diseases more explicit, we have added disease traits to our analysis, as well as considering three binary traits (Appendix—figure 13 and 14, and more detail in the response to comment #3).

2) In focusing on the main aim, a deeper exploration of the causes that drive the lack of portability would increases the informativeness of the study. It was pointed out that the difference in portability across strata might be due to differences in power, sample size, heritability, phenotypic variance, phenotype distribution, influence of environment, insights in how these and potentially others affect portability would be helpful. This should also include a discussion on how is it that different strata can generate even higher predictive power than using the same stratum (in base and target) and why the PRS based on the full sample size, in the provided examples, always has a better "predictive ability" (highest R^2^), than PRS based on any of the strata.

Following the helpful suggestions of reviewers, we realized that one of our analyses was misleading, in that the GWAS sample size in “all” was larger than that in the stratified examples, providing better power for the “all” GWAS set but in an unfair comparison. We now changed the “all” GWAS to have the same sample size as the GWAS conducted within each stratum. As we show, once sample sizes are matched, prediction accuracy for the “all”-based PGS is intermediate between the PGS based on the stratified samples (Figure 1).

In addition, we have expanded the Discussion of how a different stratum can generate even higher predictive power than using the same stratum in the GWAS and prediction sets. In short, the higher prediction accuracy results from the increased power of the GWAS in the stratum with the largest h^2^.

Beyond this discussion, we feel that an exploration of the precise factors driving variability in prediction accuracy for the traits we present is both beyond the scope of this paper and somewhat counterproductive to our main goal. In highlighting these particular examples, we wished to highlight the problem of generalizing polygenic scores from GWAS sets to prediction sets within a given ancestry group, rather than to explain the behavior of specific traits. In that regard, a detailed examination of the examples may draw attention away from our desired focus.

Having said that, we do discuss several factors that we believe are of general importance for portability across many traits, including for these examples. Notably, Figure 2A-C show that much of the trends in Figure 1 can be explained by differences in (SNP) heritabilities. In Figure 2D-F, we further show that these results cannot be explained by differences in the extent of environmental variance across strata alone. This conclusion stands in contrast to the common notion in the field (including ours, prior to performing this analysis) that h^2^ would differ across groups primarily because of differences in the extent of environmental variance. We appreciate that this point could have been easily missed by readers as originally framed and we have therefore elaborated upon it in our revision and hopefully clarified our thinking.

3) Analysis of disease outcomes, here e.g. (extreme) obesity and hypertension, will be useful to translate findings to more clinical settings, including calculation of ROC-AUC, a standard easy-to-interpret predictive statistic.

We have added analyses on extreme obesity and hypertension, using ROC-AUC as a measure for prediction accuracy (Appendix—figure 13). We find that the (incremental) AUC trends for obesity, for example, quantitatively mirror those observed for the prediction of BMI as a continuous trait. We have also added five examples of disease traits with relatively high heritability to the Appendix (Appendix—figure 14); in three of these cases, we again see differences in prediction accuracy depending on the GWAS set.

Reviewer #1:In this study, the authors examine the portability of polygenic scores within a homogeneous, generally healthy population of white British unrelated adults, living across the UK. They show that the PGS performance is influenced by age, sex, socio-economic status and study designs. The reduced accuracy of PGS is not (only) caused by environmental influences, the authors show that also the magnitude of the genetic effects among groups, and indirect effects of assortative mating affect the performance of the PGS.While these findings raise important consideration with reference to the use of PGSs, it would have been interesting if they had gone a little deeper with their analyses; e.g. to also identify the features (distribution, prevalence of disease, effects of environment) of outcomes and covariates that impact the prediction performance the most.

We agree that a dissection of different factors that influence portability would be of great utility; at the same time such a breakdown is beyond our scope in this paper and might require a different approach. For instance, we use incremental R^2^ throughout the paper mainly to facilitate comparison with key papers published previously in this area (e.g., Lee et al., 2018; Martin et al., 2019). However, it is not an ideal measure for breaking down these factors because it confounds biases and estimation noise. Furthermore, as noted in our response to the editor, we worry that focusing on the breakdown for the three specific examples may draw attention away from the generality of the concerns that we wish to raise, as these contributions likely vary in nature and magnitude across traits.

We do however discuss several factors that are likely to be generally important for portability across traits, including these examples. Notably, Figure 2A-C show that much of the trends in Figure 1 can be explained by differences in the heritability across strata.

We had initially hypothesized that these differences in h^2^ are mostly due to differences in environmental variance across strata, but as we show in Figure 2D-F, a model with constant genetic variance and changing environmental variance provides a poor fit to the data. As noted in response to the editor, we now clarify this point in revisions of the main text. Given these findings, we hypothesize that there is an interaction between genetic effects and sample characteristics, such that genetic effects are systematically larger in the groups with higher prediction accuracy, even though the effect sizes are highly correlated among strata (Appendix—table 2, Appendix—figure 3) (i.e., that we are seeing what is sometimes called “genetic amplification” in some strata).

It seems that for all traits, even though there are differences in R^2^ for sex-specific, age-specific and SES-specific PGSs, the highest R^2^ (still low) is achieved when the PGS is generated in the full population. Thus, isn't the conclusion then simply that stratification lowers the PGS performance and that a (blunt) PGS, based on the most comprehensive possible population performs best?

We apologize for the previous presentation, which led to confusion. As we detail in the response to the editor, we have modified Figure 1 to address this point (see above).

The authors only considered continuous outcomes for "prediction"; it would be informative to also include dichotomous/disease outcomes (related to the continuous risk factors), such as (extreme) obesity, hypertension, etc. That would also allow calculation prediction statistics such as AUC-ROC (sensitivity, specificity, PPV, NPV), which are easier to interpret in the context of clinical relevance.

We have now done so, both by dichotomizing continuous traits such as BMI and blood pressure and considering them as binary, and by adding five additional disease traits. Following the reviewers’ suggestions, we have used (incremental) AUC to measure prediction accuracy in these binary traits. As can be seen (Appendix—figures 13 and 14), the qualitative conclusions remain.

The PGS was generated based on LD-based clumping, but it seems only one threshold was used; r^2^<0.1. From experience, it seems that higher LD threshold results in better performance of the PGS. Thus, it might be worth testing higher LD threshold for better performing PGSs.

Our goal here was not to maximize prediction accuracy, but to show that at a given threshold, it is variable across groups of the same ancestry. Nonetheless, to evaluate the sensitivity of our analysis to the choice of the PGS model, we added an analysis repeating the three examples in Figure 1 with LDpred. We find that LDpred generally outperforms the prediction accuracy of clumping approaches. Most importantly for this paper, we find that the trends across strata remain qualitatively unchanged and are often accentuated when LDpred is used instead of clumping (Appendix—figure 2).

Reviewer #2:The manuscript of Mostafavi discusses a timely and important question: how accurate are polygenic risk scores (PRS) derived from GWAS studies and first validated in large, but single cohorts, when predicting risk in populations different to those used in their derivation. Recently GWAS studies have achieved sample sizes of a million subjects. In addition, the UK biobank has released rich datasets close to half a million people genotyped, extensively phenotyped, and followed so far for about 10 years. These resources have spurred the development of PRS for many traits ranging from complex disease onset, quantitative phenotypes, and even to sociological predictions. Furthermore, a number of direct to consumer companies have started to provide their customers PRS for medical traits. Therefore, the urgent question is how reliable are these scores and how well they port to other populations and people around the world.Mostafavi work starts with the recent publications suggesting that PRS port poorly to populations other than European descent (the population object of most of the data available), in particular to African populations. There have been recent calls to expand GWAS and biobank style cohorts to include people from other ancestries around the world. However, Mostafavi analyzes now show that problems with portability are not just evident in different ancestries, but also to different strata from samples from European decent participants. If this is correct, this raises the urgency to hold off implementations of these scores in to healthcare settings until we truly understand how and when these scores can be applied across groups of people.The authors provide a meticulous dissection of the situations when the PRS are not portable within people of the same Ancestry. They perform careful QC of the UK Biobank data, use reasonable methods to adjust the sample strata, perform the GWAS, and construct and test the PRS. The authors state that their main goal was to demonstrate how differences in the composition of the GWAS cohorts affect portability for different traits of interest within the same ancestry, and they achieve this. However, they do a good work exploring the possible factors involved in the reduced portability and leave us with important lessons on how our assumptions about GWAS data may be wrong at times. Their data and simulations add important information on how sex ratios, indirect effects, GxE, assortative mating and other factors confound the data and contribute to poor performance in subsets of the UK biobank subjects. The manuscript is well written and extensive details is provided on the methods and simulations. The main text is accompanied by extensive supplemental results of importance, which are referred at appropriate places in the text.I have a number of questions that I would like the authors consider and when appropriate clarify in their final manuscript:1) Throughout the manuscript the term "portability" is used and metrics that measure the performance of the PRS are used to judge whether portability is feasible or reduced. While this term has been used in prior literature w.r.t to the reduced ability of a PRS developed in cohorts of European descent people when applied to populations of different ancestry, this term is jargon. In statistics and machine learning the term that describes this poor performance is overfitting and the ability of a model, or in this case a PRS, to preform equally well in other samples, is termed generalization. Would it not be "overfitting" the term that more accurately describe the lack of "portability"?

We agree that the terminology is important here, but precisely because the problem of portability has to date largely been reduced to a question of population genetics alone (i.e., discussed in terms of allele frequencies and LD differences), we think it is a good idea to use the same terminology in highlighting the numerous other factors that may affect prediction in samples that differ from the GWAS sample. As we discuss in the text, these factors (e.g., SES composition, indirect effects, assortative mating patterns, stratification) differ among continental groups or different genetic ancestry groups in the same country. Therefore, our within-ancestry group analyses pertain to cross-ancestry portability as well.

2) The author's use R^2^ to measure the reduced generalization of the PRS to the different cohorts' strata, and they concede that is not clear what would be the best metric to use for this analysis. However, many publications use the AUC as a metric to decide the best model and threshold of SNPs to pick and to describe how the PRS augments other predictor bases on covariates or other factors. It is not intuitive how R^2^ correlates with AUC, and therefore it would be best if the authors try in at least one analysis to compare R^2^ to AUC.

As the reviewer notes, we use R^2^ not because we believe it is in any sense an optimal measure, but to allow for more ready comparisons to previous work. Following the reviewer’s suggestion, we have now added analyses with AUC instead of R^2^ (Appendix—figures 13 and 14) for a total of eight traits, including five disease traits. The qualitative conclusions are the same; a more in depth examination of how AUC relates to R^2^ is, in our opinion, beyond the scope of this paper.

3) Recent publications have used the LDPred method that uses as input summary statistics rather than genotypes to develop PRS based on UK Biobank data. Some of these have shown generalization to others cohorst. Can the author's comment whether LDpred features distinguished it from the methods used in their analysis conferring an advantage? PRS derived from LDpred also tend to include many more SNPs to achieve a greater AUC. The author's explore P-value cut-offs (and hence number of SNPS) in some analysis, but it was not clear whether this suggested more SNPs increases overfitting. Do the results from the author's suggest that including more SNPs increases overfitting?

As noted in our response to the reviewer #1, we added an analysis repeating the three examples in Figure 1 with LDpred. We find that LDpred (with a prior probability of 1 on the loci being causal) generally outperforms the prediction accuracy of clumping methods. Most importantly for this paper, we find that the trends across strata remain qualitatively unchanged and are often accentuated when LDpred is used instead of clumping (Appendix—figure 2).

Reviewer #3:The generalisability of polygenic scores is an important issue given the possibility of them being incorporated in to healthcare soon. This paper makes excellent use of the UK Biobank resource to present two sets of interesting results relating to the predictive power of PRS. However, I'm not sure if the way in which the primary results are framed in relation to portability is quite right.The first set of results appear to highlight how GWAS are better powered in certain subgroups of the population (eg. GWAS of DBP are better powered in women, BMI in those in middle age, years of schooling in lower SES groups). Years of schooling is better predicted in SES 4 by GWAS on SES 1 than by SES 4 itself – this shows hyper-portability, not limited portability as discussed throughout the article. This is also true for BMI and DBP (to a lesser extent DBP). The bivariate LD Score results showing genetic correlations close to 1 seems to support this being principally a power (rather than portability) issue, in which there are different relative contributions of genetics and the environment across the groups, making some better-powered to capture – largely the same – genetics. Based on the results (inc. those of Figure 2), I'd have thought that this is most likely due to the environment resulting in convergence and genetics resulting in divergence (at least in the case of BMI and education) – eg. BMI converges to low values in old age, irrespective of genetic predisposition (relatively), while education levels of those in the top SES are forced to be similar due to the qualification requirements of professional jobs. Thus, the highest phenotypic variance, heritability, and thus PRS predictive power for BMI is from the lower age group and schooling from the lowest SES. How specific this pattern of results is to these particular traits is unclear, and certainly there seems to be a somewhat different cause for the DBP results. However, in general I see these results as being analogous to eg. how GWAS on Alzheimer's is perhaps being best-powered to detect common genetic variants when applied to individuals in their 70s, GWAS on blood pressure best-powered when applied to individuals in their 30s, and lung cancer GWAS being optimised when applied to non-smokers.

We agree that a central factor driving the trends of Figure 1 is likely the differential power (and the precision of the estimates) across GWAS strata. We now further clarify this point in the text. We note further that for the three examples in Figure 1 and Figure 2, the data are poorly explained by assuming that the environmental variance differs across strata but the genetic variance remains the same (see Figure 2D-F). We therefore hypothesize that genetic variance is also substantially variable across strata; in other words, that there is an interaction between the genetics and the strata characteristics (Figure 2, Appendix—figure 3).

While the downstream impact of variable GWAS power in different sub-groups on PRS prediction is extremely valuable to see here, I think the implications of these results may be missed unless discussed in the context of differential GWAS power in different sub-groups – eg. in relation to ancestry, the consensus is that portability will be optimised by expanding GWAS sampling to non-European populations, but the analogy is not true here – the consequence of these results (at least for these traits) is that portability will be optimised if GWAS sampling is restricted to those sub-groups for which heritability is highest (since, otherwise, the portability in fact seems high).

The reviewer brings up a very interesting point. We have expanded the Discussion of how a different stratum can generate higher predictive power than using the same stratum in the GWAS and prediction sets. As the reviewer notes, the higher prediction accuracy is largely a result of the increased power of the GWAS in the stratum with the largest (SNP) h^2^ estimates.

The implications for how best to conduct a GWAS are not obvious to us, for a number of reasons: one, the three examples we present were “cherry-picked” in the sense that we had prior information about what sample characteristics may matter for the trait (e.g., sex for diastolic blood pressure). We used them to show that such characteristics *can* matter, and matter just as much as factors like ancestry. But these are just some characteristics about which we had prior knowledge, and it seems likely that there exist many more, unknown characteristics that affect heritability for these traits--and that for some other traits, it will be hard to guess a priori what may matter. We now show an additional example in the Appendix: when we focus on a trait of interest in the social sciences (educational attainment) and follow the study design used in most studies, the prediction accuracy of educational attainment is somewhat higher for individuals with a (single) sibling than individuals without a sibling, when the GWAS is done in individuals with one sibling (see Appendix—figure 12).

Second, as we discuss, factors such as GxE, assortative mating and indirect effects are soaked up into the GWAS estimates--and critically also into the SNP heritability estimates. Thus, the choice of a GWAS sample is about more than detection power; it is implicitly making a choice about all sorts of sample characteristics than may or may not hold true of the prediction set.

Given these considerations, it is not obvious to us how to choose an optimal GWAS sample a priori. Notably, as mentioned by the reviewer, the most “diverse” set (with respect to the sample characteristic) is not necessarily the optimal set in which to conduct the GWAS: in the revised version of Figure 1, we now show that after matching the sample size, the performance of a GWAS done on an unstratified sample (“all”) is intermediate between the performances of GWAS on stratified samples (Figure 1).

[Editors' note: further revisions were suggested prior to acceptance, as described below.]

The manuscript has been improved and reviewers were generally satisfied with the changes made. However, while discussing the revisions some remaining issues were noted that need to be addressed before acceptance, as outlined below:1) As sample size is an important determinant of the accuracy of the association effect sizes and thus also of the prediction accuracy, the sample sizes of each stratum needs to be clearly stated. Currently, the use of "unstratified" and "all" is misleading as it would suggest it include the data of strata combined. However, this seems not the case; the "unstratified" or "all" have now been ("artificially") reduced to match the sample size of the strata, which is not intuitive. After all, when data for multiple strata is available, one would be better off combining all data, rather than using the strata. We suggest to clearly report sample sizes, and to choose a more precise terminology to describe the "unstratified" or "all" group. Including an "all" group, that combined strata, as before, would seem informative to illustrate that sample size trumps population-specificity.

We agree that the term “all” is unclear, and thank the editors and reviewer #1 for their suggestion. We now use the term “diverse” to refer to the sample that consists of individuals from all strata. We also report the sample sizes used for GWAS in Figure 1 in the main text and the figure caption, in addition to the Materials and methods.

2) One of the reviewers noted difference between the original Figure 1 and the new figure 1, and wondered whether the data changed (slightly) compared to their previous version because the results are slightly different (easiest to see by comparing median/boxes of young/old in BMI plot between original and new versions). It would be worth checking that this is just updated UKB data rather than an error. Furthermore, the R^2^ for DBP in the "mixed GWAS" is just less than half than for the "total GWAS" (again comparing original Vs new figures), while for education there's a 10-fold difference (ie. doubling GWAS N has led to 10x greater R^2^).. This can be the case ('inflection points' and all), but it would be worth checking that there's no error here because it seems a little surprising.

For our analysis of DBP by sex we downsampled the “diverse” GWAS by a factor of ~2 (given two sex strata), whereas for BMI by age and years of schooling by SES we downsampled by a factor of ~4 (given four BMI and SES strata). As a result, the drop in R^2^ between our initial submission (without downsampling) and our resubmission (with downsampling) is more pronounced for BMI and years of schooling compared to DBP.

In addition, since our initial submission we removed 60 UKB participants that withdrew from the UKB, and repeated our analyses. Therefore, slight differences are expected due to random sampling.

[Editors' note: further revisions were suggested prior to acceptance, as described below.]

While you have been very responsive to the reviewers comments and concerns, we remain concerned that some of the presentation of findings remains counter-intuitive and could lead to mistaken inference. We would like you to consider one final set of revise your paper to include the following:1) Include strata-combined analyses; i.e. a "full" analysis that combines the stratum-specific sample sizes into one large population. The current study shows that the stratum-specific PRSs perform better than strata-combined PRS, but only if the strata-combined (e.g. men+women) PRS is based on GWAS of the same size as the individual strata (e.g. men, women).In the very first version of the paper, it was clear that the strata-combined PRS (based on combined stratum-specific data) performs much better than any of the stratum-specific PRS: it seems that the larger sample size overcomes any advantage of stratum-specific analyses (in these examples at least). In response to the reviewers' comments, the authors "reduced" the sample size of the strata-combined analyses to the size of that of the individual strata. However, reducing a strata-combined analyses to the same sample size as the stratum-specific sample sizes is not an intuitive comparison (i.e. a combined analyses indicated that samples are merged into a bigger sample), as stratum-specific implies that these are a subset of the strata-combined. The current version does not address this issue: the labeling of the (sub)groups has been changed, but that has the potential to mislead the reader to infer that stratum-specific analyses will provide the best prediction, whereas a strong predictor is also sample size.2) Discuss the role of sample size in prediction in the Discussion section;i.e. put the impact of stratum-specificity in prediction in comparison to the impact of sample size in the (this relates to the comment above). It is important that readers understand that both contribute to prediction, and that possibly sample size is even more important than stratum-specificity. It would be good if the authors can speculate on any circumstances in which the strata-combined analysis would not provide better overall prediction than the stratum-specific subsets.

We thank the editors for their consideration and input. We have added a Figure (Appendix—figure 1) to reflect the point raised by the editors. We have also added to the main text explicit comments on the higher prediction accuracy for the much-larger sample size of a GWAS combining all strata (i.e., without matching sample sizes).

Having said that, we agree with previous reviewers’ comments (and revised our manuscript accordingly) on the potential for misinterpretation in including a GWAS with a larger sample size in a main-text figure. Further, in matching sample sizes we mimic the methodology in previous research on polygenic score portability, to highlight the factors other than allele frequencies and linkage disequilibrium that affect polygenic prediction. Not matching would confound the effect of these factors with the effect of sample size.

We agree with the editors that the interplay of prediction accuracy with sample size is very important, but see it as orthogonal to the points discussed in the paper. In fact, our analyses explicitly control for sample size effect throughout. We therefore suspect that adding such a discussion might only act to confuse the reader.

[Editors' note: further revisions were suggested prior to acceptance, as described below.]

The manuscript has been improved but there are some remaining issues that need to be addressed before we move to final acceptance. These relate to the discussion about the trade-off between sample size and stratum-specific effects on PRS performance that have been the subject of the recent revision requests. Thank you for adding additional information and figures that address this issue. However, as per the last round of revision requests, we do not feel that this question has been adequately addressed in the Discussion, and that some explicit text on this matter forms an essential part of the paper. As you will have seen from the to and fro over the revisions, this is an issue that has exercised several of the editors and reviewers, and I can’t imagine that it will be any less important to the readers of the paper. Put simply, the question is this: based on the data you have generated, are there any real world situations where researchers would be better off splitting a data set into its component strata (eg by gender) and proceeding on the basis of stratum specific risk scores, rather than using the whole data set? The data in the additional figure, suggest not (at least in the models and situations you tested), which is why this is important to place into context.

We have added a paragraph to the Discussion section discussing the implications of our findings for PGS construction, notably on the choice between GWAS samples limited to one stratum versus the union of all strata, and removed the previous discussion of this point from the Results section.